# Demystifying the Optimal Fair Classifier in Multi-Class Classification

**Li Zhang**[1] **Yuyuan Li**[2,*] **XiaoHua Feng**[1] **Jiaming Zhang**[1] **Fengyuan Yu**[1] **Chaochao Chen**[1]

## Abstract

Ensuring fair and equitable treatment across diverse groups, particularly in multi-class classification tasks, poses a significant challenge due to the persistent biases inherent in machine learning models. Most existing bias mitigation techniques are tailored to binary settings, and the presence of multi-dimensional outputs and complex fairness mechanisms makes their extension to multi-class scenarios neither straightforward nor effective. In this paper, we investigate two fundamental, unresolved challenges in fair classification: (i) *characterizing the optimal accuracy-fairness frontier in multi-class settings*, and (ii) *designing practical algorithms that attain this optimum in different training phases*. To tackle these challenges, we first specify an analytically tractable probabilistic formulation of the optimal classifier under fairness constraints. Building upon this, we propose two attribute-blind algorithms to enforce fairness requirements in practice: an in-processing approach for fairness intervention during training via the reduction approach, and a post-processing approach for fine-tuning output probabilities with plug-in estimation. Theoretical analysis reveals that both methods converge to the optimal accuracy-fairness Pareto frontier. Experiments conducted on multiple datasets demonstrate the superior performance of our methods in balancing accuracy and fairness.

## 1. Introduction

In recent decades, machine learning models have become extensively integrated into decision-making applications in high-stakes domains such as healthcare (Liu et al., 2025; Zhang et al., 2025; Hu et al., 2024), financial services (Chouldechova, 2017; Hurlin et al., 2024; Du et al., 2024), and criminal justice (Berk et al., 2021; Barocas et al., 2023; Zeng et al., 2025b). This trend has motivated a growing body of research focused on developing and assessing fairness to mitigate discrimination in models. In this context, *group fairness* metrics are frequently employed in practice due to their consistency with the intuition that predictions should not systematically discriminate against any group within the population. Widely used group fairness metrics, such as demographic parity (DP; Dwork et al. (2012)), equal opportunity (EOP: Hardt et al. (2016)), and Equalized Odds (EO; Hardt et al. (2016)), vary in their definition of group-wise disparity.

Achieving high predictive performance while simultaneously enforcing group-fairness constraints poses a fundamental problem in multi-class fair learning. Most prior studies demonstrate an inherent trade-off between fairness and accuracy, whereby improving fairness typically comes at the cost of reduced accuracy (Chen et al., 2024; Xian et al., 2023; Xu & Strohmer, 2023; Zheng et al., 2022). However, despite these empirical and theoretical advances, *an explicit characterization of the optimal accuracy-fairness frontier in multi-class settings remains elusive*. This difficulty stems from the intrinsically non-decomposable and non-differentiable nature of most group-fairness criteria (Lohaus et al., 2020; Yao et al., 2024), which becomes further exacerbated by the multi-dimensional output structure (Alghamdi et al., 2022). Consequently, it is unclear whether the performance degradation arises from intrinsic algorithmic limitations or incidental artifacts, and the Pareto boundary between fairness and accuracy has not been established.

In practice, fairness calibration is commonly deployed either during training by modifying the model training objective (*in-processing*; Agarwal et al. (2018); Yazdani-Jahromi et al. (2024)) or after training by adjusting the pre-trained model's outputs (*post-processing*; Xian et al. (2023); Chen et al. (2024); Hu et al. (2025)). The current challenge in multi-class fairness calibration lies in *the lack of a systematic and consistent framework for approximating the optimal accuracy-fairness trade-off across different training phases*. Existing work typically develops in-processing and post-processing methods in isolation, with most approaches focusing on one paradigm exclusively. Moreover, the in-processing methods predominantly rely on surrogate fairness metrics to design objectives or reweighting meth-

---

[1]College of Computer Science and Technology, Zhejiang University [2]School of Communication Engineering, Hangzhou Dianzi University. Correspondence to: Yuyuan Li <y2li@hdu.edu.cn>.

*Proceedings of the 43rd International Conference on Machine Learning*, Seoul, South Korea. PMLR 306, 2026. Copyright 2026 by the author(s).

ods (Roh et al., 2021; Jung et al., 2023; Yazdani-Jahromi et al., 2024), while the inevitable surrogate gap (Yao et al., 2024; Lohaus et al., 2020) produces suboptimal performance and undermines convergence stability. For multi-class post-processing, certain advanced methods focus on a single fairness criterion (Xian et al., 2023; Denis et al., 2024), while others lack a precise characterization of the optimal fair classifier, leaving performance gains unexplored (Zhang et al., 2024; Tifrea et al., 2024). Hence, it remains an open problem to design algorithms that are applicable to multiple fairness criteria and controllable in different training phases.

To address these challenges, we theoretically investigate the Pareto frontier in multi-class fair classification and propose a novel group-fairness calibration framework named `OptFair`, under which we separately develop an in-processing method and a post-processing method to approximate the optimal accuracy-fairness trade-off. We specify the Bayes-optimal classifier under group-fairness constraints in the multi-class setting and introduce an entropic regularizer to render it analytically tractable. Building on this characterization, we develop optimization strategies that achieve the optimal accuracy-fairness trade-off for any specified level of fairness: the in-processing task is reduced to a sequence of cost-sensitive classification problems, while the post-processing problem is reformulated as a convex optimization problem via a plug-in estimator. We further analyze the generalization risk and show that both methods are statistically consistent with the optimal accuracy-fairness equilibrium. Experiments on multiple real-world datasets demonstrate that `OptFair` offers a more controllable accuracy-fairness balance and competitive classification performance compared to existing methods.

Our main contributions are summarized as follows:

- We present an analytically tractable formulation of the optimal classifier that achieves the optimal accuracy-fairness frontier in the multi-class setting.

- In-processing and post-processing algorithms are developed to approximate optimal fair classifiers, using a reduction-based method for in-processing and plug-in estimation for post-processing.

- Theoretically, generalization bounds are provided to demonstrate that both methods are statistically consistent with the optimal fair classifier.

- Extensive experiments on various datasets demonstrate that `OptFair` surpasses existing methods in balancing accuracy and fairness, with the flexibility to adjust the accuracy-fairness trade-off.

## 2. Related Work

**Multi-class group-fairness.** Multi-class fairness extends classical notions such as demographic parity and equal-ized odds by imposing distributional constraints on vector-valued prediction outcomes across sensitive groups (Alghamdi et al., 2022; Xian et al., 2023; Denis et al., 2024; Jung et al., 2023). Unlike the binary case, where fairness constraints reduce to scalar rate matching, the multi-class setting induces coupled constraints over the probability simplex, resulting in more complex feasibility regions.

**Accuracy-fairness trade-off.** A growing body of work has investigated the trade-off between predictive utility and fairness through constrained risk minimization and Pareto-optimal frontiers, particularly in the binary classification setting (Menon & Williamson, 2018; Zeng et al., 2024; Chen et al., 2024). However, theoretical understanding of the multi-class accuracy-fairness trade-off remains limited, with many existing characterizations deriving optimal fair classifiers only under restrictive structural assumptions (Denis et al., 2024; Xian & Zhao, 2024). Consequently, general multi-class settings still lack analytically tractable characterizations of Bayes-optimal fair classifiers, leaving the induced optimal accuracy-fairness frontier under-specified.

**In-processing.** Training-time fairness interventions typically enforce constraints through regularization (Kim et al., 2022; Baharlouei et al., 2024; Cho et al., 2020), dual or bilevel optimization (Roh et al., 2021; Yazdani-Jahromi et al., 2024), adversarial objectives (Zhong & Tandon, 2023; Zhang et al., 2018), or reduction-based formulations (Agarwal et al., 2018). While effective in practice, extending these methods to multi-class tasks is nontrivial due to coupled, multi-dimensional constraints, and existing theories seldom imply convergence to the accuracy-fairness frontier.

**Post-processing.** Post-processing approaches aim to correct model outputs without retraining, e.g., by thresholding scores (Chen et al., 2024; Zeng et al., 2024), mapping predictive probabilities (Tifrea et al., 2024; Alghamdi et al., 2022), or calibrating group-conditioned distribution (Xian et al., 2023; Xian & Zhao, 2024; Denis et al., 2024). Despite their computational appeal, many existing post-processing techniques are either designed for binary settings or depend on attribute-aware adjustments, making them less suitable when sensitive attributes are unavailable at inference time.

## 3. Preliminaries

**Notation.** The $p$-norm of the vector $\mathbf{a}$ is denoted as $\|\mathbf{a}\|_p$, while $\mathbf{1}_{m \times m}$ represents the $m \times m$ all-ones matrix. Write the probability simplex as $\Delta_m = \{ \mathbf{q} \in [0, 1]^m : \|\mathbf{q}\|_1 = 1 \}$, and let $\mathbf{e}_i$ be the $i$-th standard basis vector. $[\cdot]^\top$ represents the transpose of the vector or matrix. $\mathbb{I}_{\{\cdot\}}$ is the indicator function, which equals 1 if the event occurs and 0 otherwise. For two equally sized matrices $\mathbf{A}$ and $\mathbf{B}$, their Frobenius inner product is $\langle \mathbf{A}, \mathbf{B} \rangle = \sum_{i,j} a_{ij} b_{ij}$. For positive integer $n$, $[n] := \{1, \ldots, n\}$.

*Table 1.* Recovering Commonly Used Fairness Constraints from Group-Specific Confusion Matrices.

| Fairness Criteria | Fairness Constraints | Confusion Matrices Notation |
|---|---|---|
| Demographic parity | $\left\|\mathbb{P}(\widehat{Y}=y \mid A=a) - \mathbb{P}(\widehat{Y}=y)\right\| \leq \xi$ $\forall a \in \mathcal{A}, \forall y \in \mathcal{Y}$ | $\left\| \sum_{a' \in \mathcal{A}} \sum_{i \in \mathcal{Y}} [\omega_{a'} - \mathbb{I}_{a'=a}] \mathbf{C}_{i,y}^{a'} \right\| \leq \xi$ |
| Equal Opportunity | $\left\|\mathbb{P}(\widehat{Y}=y \mid A=a, Y=y) - \mathbb{P}(\widehat{Y}=y \mid Y=y)\right\| \leq \xi$ $\forall a \in \mathcal{A}, \forall y \in \mathcal{Y}$ | $\left\| \sum_{a' \in \mathcal{A}} \omega_{a'} \left[ \frac{1}{\mathbb{P}(Y=y)} - \frac{\mathbb{I}_{a'=a}}{\mathbb{P}(Y=y, A=a)} \right] \mathbf{C}_{y,y}^{a'} \right\| \leq \xi$ |
| Equal Odds | $\left\|\mathbb{P}(\widehat{Y}=y \mid A=a, Y=y') - \mathbb{P}(\widehat{Y}=y \mid Y=y')\right\| \leq \xi$ $\forall a \in \mathcal{A}, \forall y, y' \in \mathcal{Y}$ | $\left\| \sum_{a' \in \mathcal{A}} \omega_{a'} \left[ \frac{1}{\mathbb{P}(Y=y')} - \frac{\mathbb{I}_{a'=a}}{\mathbb{P}(Y=y', A=a)} \right] \mathbf{C}_{y',y}^{a'} \right\| \leq \xi$ |

**Multi-class fair classification.** Let $\mathcal{P} \sim (X, A, Y)$ be a random tuple, where $X \in \mathcal{X}$ for some feature space $\mathcal{X} \subseteq \mathbb{R}^d$, labels $Y \in \mathcal{Y} = [m]$, and the discrete sensitive attribute $A \in \mathcal{A}$. Given the attribute-blind randomized classifiers $h \in \mathcal{H} : \mathcal{X} \to \Delta_m$, the prediction $\widehat{Y}$ is associated with the random outputs of $h$ defined by $\mathbb{P}(\widehat{Y}=y \mid \mathbf{x}) = h_y(\mathbf{x})$. A randomized function is said to be Bayes-optimal, denoted by $\eta$, if it computes the true class probabilities exactly. Specifically, for $y \in [m]$, we denote the ground-truth conditional probability by

$$\eta_y(x) := \mathbb{P}(Y=y \mid X=x),$$
$$\eta_y(x,a) := \mathbb{P}(Y=y \mid X=x, A=a).$$

Denote by $\mu$ the marginal distribution of the data $X$, and by $\omega_a := \mathbb{P}(A=a) > 0$ the group weight. **Group-fairness constraints.** Let $A$ represent a group attribute (e.g. race and/or gender), where $A \in \mathcal{A}$. In this work, we generally focus on three commonly used group-fairness criteria, Demographic Parity (DP; Dwork et al. (2012)), Equalized Opportunity (EOP; Hardt et al. (2016)) and Equalized Odds (EO; Hardt et al. (2016)). Table 1 provides their definitions in the multi-class setting with multiple sensitive attributes, following prior work (Alghamdi et al., 2022; Denis et al., 2024; Xian et al., 2023; Xian & Zhao, 2024).

**Confusion matrices.** As fundamental analytical tools, confusion matrices encapsulate the information needed to derive diverse performance metrics and to assess group-fairness constraints in classification tasks (Yang et al., 2020; Narasimhan et al., 2024; Kim et al., 2020; Xu et al., 2026). Denote the population confusion matrix by $\mathbf{C} \in [0,1]^{m \times m}$, with elements defined for $i, j \in [m]$ as $\mathbf{C}_{i,j} = \mathbb{P}(Y=i, \widehat{Y}=j)$. To represent group-fairness constraints, we focus on the group-specific confusion matrices $\mathbf{C}^a$, $a \in \mathcal{A}$, where $\mathbf{C}_{i,j}^a := \mathbb{P}(Y=i, \widehat{Y}=j \mid A=a)$.

**Performance and Fairness metrics.** For performance metrics, we consider a risk metric expressed as a linear function of the population confusion matrix, namely $\mathcal{R}(h) = \langle \mathbf{R}, \mathbf{C}(h) \rangle = \sum_{a \in \mathcal{A}} \omega_a \langle \mathbf{R}, \mathbf{C}^a(h) \rangle$. In this paper, we primarily focus on standard classification error rate $\mathbb{P}(\widehat{Y} \neq Y)$ to set $\mathbf{R} = \mathbf{1}_{m \times m} - \mathbf{I}$. For fairness metrics, as presented in Table 1, the fairness constraints can typically

be expressed by the following formulation:

$$|\mathscr{D}_k(h)| = \left| \sum_{a \in \mathcal{A}} \langle \mathbf{D}^{a,k}, \mathbf{C}^a(h) \rangle \right| \leq \xi, k \in [K],$$

where $K$ represents the number of constraints required to implement the fairness criteria. For proofs, additional fairness criteria, and further discussions, see Appendix A.

## 4. Optimal Fairness-Constrained Classifier

This section develops a parametric characterization of the Bayes-optimal fair classifier under mild assumptions. The proofs in this section can be found in the Appendix B.

We study optimal classifiers by formulating accuracy maximization under fairness constraints. Given a specified fairness level $\xi > 0$, the primal problem is defined as:

$$\min_{h \in \mathcal{H}} \mathcal{R}(h) = \sum_{a \in \mathcal{A}} \omega_a \langle \mathbf{R}, \mathbf{C}^a(h) \rangle,$$

$$\text{s.t. } |\mathscr{D}_k(h)| = \left| \sum_{a \in \mathcal{A}} \langle \mathbf{D}^{a,k}, \mathbf{C}^a(h) \rangle \right| \leq \xi, \ k \in [K]. \quad (1)$$

To tackle this problem, prior studies typically impose the continuity assumption on the output distribution (Chen et al., 2024; Denis et al., 2024; Xian & Zhao, 2024), which may fail in the presence of discrete distributions. In contrast, we adopt an assumption that entails negligible cost.

**Assumption 4.1.** (Feasibility). There exists a classifier $h$ that is feasible for (1) when $\xi = 0$.

This assumption is primarily imposed to guarantee the existence of the solution, ensuring that the feasible set is nonempty for any $\xi > 0$. The fairness criteria in Table 1 obviously meet this assumption since there exist naive classifiers (e.g., $h(x) = \mathbf{e}_y, y \in [m], \forall x \in \mathcal{X}$) that satisfy the fairness constraint when $\xi = 0$. This assumption also applies to other fairness criteria, as discussed in the Appendix B.1. The next theorem presents a formulation of the Bayes-optimal fair classifier.

**Theorem 4.2.** *Under the assumption 4.1, for any $\xi > 0$, there exists an optimal solution to the problem (1), which*

*can be realized through the following form,*

$$h^*(x) \in \text{conv} \left\{ e_y : y \in \underset{j \in [m]}{\arg\max} \, \beta_j^{\lambda^*}(x) \right\}, \quad (2)$$

$$\beta^\lambda(x) = \sum_{a \in \mathcal{A}} p_a(x)[\mathbf{M}(a, \lambda)]^\top \eta(x, a), \quad (3)$$

$$\mathbf{M}(a, \lambda) := \mathbf{I} - \frac{1}{\omega_a} \sum_{k=1}^K \lambda_k \mathbf{D}^{a,k} \quad (4)$$

*where $p_a(x) := \mathbb{P}(A = a | X = x)$, the dual parameter $\lambda \in \mathbb{R}^K$, and $\text{conv}\{\cdot\}$ denotes the convex hull. Denoting $\Lambda := \{\lambda \in \mathbb{R}^K : \|\lambda\|_1 \leq B_\Lambda\}$, the parameter $\lambda^*$ is the solution of the dual problem:*

$$\min_{\lambda \in \Lambda} H(\lambda) = \mathbb{E}_X \left[ \max_{j \in [m]} \beta_j^\lambda(X) \right] + \xi \|\lambda\|_1. \quad (5)$$

**Proof Sketch.** We begin by deriving the formulation of the dual problem. Considering the Lagrangian $\mathcal{L}(h, \lambda^{(1)}, \lambda^{(2)})$,

$$\mathcal{R}(h) + (\lambda^{(1)} - \lambda^{(2)})^\top \mathscr{D}(h) - \xi \|\lambda^{(1)} + \lambda^{(2)}\|_1, \quad (6)$$

where $\lambda^{(1)}, \lambda^{(2)} \in \mathbb{R}_{\geq 0}^K$ are the dual parameters, and $\mathscr{D}(h) := \{\mathscr{D}_k(h)\}_{k \in [K]}$. Denoting $\lambda := \lambda^{(1)} - \lambda^{(2)}$, the dual problem can be proven to be equivalent to

$$\max_{\lambda \in \mathbb{R}^K} \min_{h \in \mathcal{H}} \mathcal{L}(h, \lambda) = \mathcal{R}(h) + \lambda^\top \mathscr{D}(h) - \xi \|\lambda\|_1. \quad (7)$$

We derive that the inner minimization has a solution formulated in Eq. (2), with which the dual optimization objective is deduced as Eq. (5). Then, to verify that the primal-dual gap is zero, we further prove the boundedness, feasibility, and optimality of the solution presented above. □

Although a formal characterization of the optimal fair classifier is given in Theorem 4.2, learning this classifier in finite-sample settings is challenging due to its analytical intractability. To remedy this, we introduce a relaxed version of the primal problem. Motivated by the entropic optimal transport (Cuturi, 2013; Genevay et al., 2016), we incorporate a tailored entropic regularizer into primal objective. For $h \in \mathcal{H} : \mathcal{X} \to \Delta_m$, the differential entropy of the probabilistic classifier $h$ is given by

$$\mathcal{E}(h) := -\mathbb{E}_X \left[ \sum_{i=1}^m h_i(X) \log h_i(X) \right]. \quad (8)$$

Since $h_i(x) \log h_i(x) \to 0$ as $h_i(x) \to 0$ for $i \in [m]$, we define $h_i(x) \log h_i(x) := 0$ when $h_i(x) = 0$, thereby ensuring that the differential entropy is well-defined on $\Delta_m$. The fair classification with entropic regularization is formulated as

$$\min_{h \in \mathcal{H}} \mathcal{R}(h) - \tau \mathcal{E}(h), \quad \text{s.t.} \ |\mathscr{D}_k(h)| \leq \xi, \ k \in [K]. \quad (9)$$

In this case, $\tau \geq 0$ is the hyperparameter that controls the impact of the entropy regularization term. The regularized problem has the following closed-form solution.

**Theorem 4.3.** *Under the assumption 4.1, for any $\xi > 0$, there exists an optimal solution defined by (9), and its closed-form expression is given as follows,*

$$h_i^{\lambda^*}(x) = \frac{\exp\left(\beta_i^{\lambda^*}(x)/\tau\right)}{\sum_{j=1}^m \exp\left(\beta_j^{\lambda^*}(x)/\tau\right)}, \quad i \in [m]. \quad (10)$$

*where $\beta_i^\lambda(x)$ denotes the $i$-th element of $\beta^\lambda(x)$ defined in (3), and the parameter $\lambda^* \in \mathbb{R}^K$ is the solution to the dual problem:*

$$\min_{\lambda \in \Lambda} H(\lambda) = \mathbb{E}_X \left[ \tau \log \sum_{j=1}^m \exp\left(\beta_j^\lambda(X)/\tau\right) \right] + \xi \|\lambda\|_1.$$

The following example illustrates the form of the optimal classifier under the DP constraint, and the solutions for other fairness constraints can be derived in a similar manner.

**Example 4.4.** (DP.) *Using the constraints for demographic parity as described in Table 1, denoting the dual parameter as $\lambda \in \mathbb{R}^{m \times |\mathcal{A}|}$, its optimal fair classifier is determined by the decision vector $\beta^\lambda(x) \in \mathbb{R}^m$, where*

$$\beta_i^\lambda(x) = \eta_i(x) + \sum_{a \in \mathcal{A}} \lambda_{i,a} \left( 1 - \frac{p_a(x)}{\omega_a} \right), \ i \in [m]. \quad (11)$$

*Plugging (11) into Theorem 4.2 and Theorem 4.3 can obtain the optimal fair classifier for the original problem (1) and the entropy-regularized problem (9).*

By introducing an entropic regularizer, the optimal classification problem under fairness constraints admits an explicit expression and becomes smoother and analytically tractable, thereby providing a theoretical foundation for the development of efficient debiasing algorithms that accommodate multiple fairness criteria across different training phases.

## 5. Methodology

In Theorem 4.2 and 4.3, we derive the explicit form of the ground-truth Bayes-optimal classifier under fairness constraints, given knowledge of the distribution $\mathcal{D} = (X, Y, A)$. However, in practice, we only have access to finite data $\mathcal{S} = \{(x_i, y_i, a_i)\}_{i=1}^N$ sampling from $\mathcal{D}$, and it remains necessary to develop feasible algorithms for achieving the optimal accuracy-fairness trade-off. Fairness calibration is typically performed through intervention during the training phase (in-processing) or by adjusting the outputs of the pre-trained model (post-processing), both of which will be detailed subsequently. The proofs in this section can be found in the Appendix C.

## 5.1. Reductions Approach (In-Processing)

In-processing approach aims to directly learn the model that realizes the fairness constraint, where the classifiers $h : \mathcal{X} \to \Delta_m$ are parameterized by a function class $\mathcal{F}$ of model $f_\theta : \mathcal{X} \to \mathbb{R}^m, \theta \in \Theta$. A direct approach to solving (1) is to formulate an equivalent convex-concave problem in terms of its Lagrangian. Given that $h$ is a randomized classifier, the primal problem can be equivalently written as the following saddle-point optimization,

$$\max_{\lambda \in \Lambda} \min_{h \in \mathcal{H}} \mathcal{L}(h, \lambda) = \min_{h \in \mathcal{H}} \max_{\lambda \in \Lambda} \mathcal{L}(h, \lambda). \quad (12)$$

From the game-theoretic perspective, randomized classifiers act as mixed strategies for the stochastic game, with the presence of equilibrium ensuring min-max interchangeability and the existence of the optimal solution (Chatterjee et al., 2004; Agarwal et al., 2018).

For the inner optimization $\min_{h \in \mathcal{H}} \mathcal{L}(h, \lambda)$, the optimal solution has been provided in (2) for the original problem, which admits a simplified and sufficient form $h^*(x; \beta^\lambda) \in \arg\max_j \beta_j^\lambda(x)$. However, the saddle-point solution remains computationally intractable due to the agnostic nature of the distribution $\mathbb{P}(A \mid X)$ and the Bayes-optimal score $\eta$. To address this issue, we reduce the task of solving $h^*$ given $\lambda$ into a cost-sensitive learning objective.

**Theorem 5.1.** *Let the cost-sensitive loss be defined by*

$$\ell_{\text{cal}}(y, f(x; \theta), a, \lambda)$$
$$= - \sum_{i=1}^m \left[ \mathbf{M}'(a, \lambda) \right]_{y,i} \log \frac{\exp([f(x;\theta)]_i)}{\sum_{j=1}^m \exp([f(x;\theta)]_j)}, \quad (13)$$

*where $f(x; \theta)$ is the model to train, and $\mathbf{M}'(a, \lambda) = \mathbf{M}(a, \lambda) + \kappa \mathbf{1}_{m \times m}$ with $\kappa$ chosen to ensure all matrix entries are strictly positive. Given $\lambda \in \Lambda$, the optimal score function $f^* \in \arg\min_f \mathbb{E}_{\mathcal{D}}[\ell_{\text{cal}}(Y, f, A, \lambda)]$ ensures that the corresponding classifier $h^*(x; f)$ is equivalent to $h^*(x; \beta^\lambda)$. The loss $\ell_{\text{cal}}$ is referred to as calibrated for the objective $\min_{h \in \mathcal{H}} \mathcal{L}(h, \lambda)$.*

With calibrated loss, since the problem is convex-concave with the convex domain, we seek the saddle point using the primal-dual optimization scheme, which has proven effective in the binary case (Agarwal et al., 2018; Cotter et al., 2019), as detailed in Algorithm 1.

**Theorem 5.2.** *Suppose that $\ell_{\text{cal}}$ receives a $\nu^t$-approximate optimal parametric response $\theta^{t+1}$ at each iteration t, i.e., $\widehat{\mathcal{L}}(h^{t+1}, \lambda^t) \le \min_{h \in \mathcal{H}} \widehat{\mathcal{L}}(h, \lambda^t) + \nu^t$. Denoting $u := \max_{h \in \mathcal{H}} \|\widehat{\mathscr{D}}(h)\|_\infty + \xi$, and $\bar{\nu}^T := \sum_{t=1}^T \nu^t / T$, for $\eta_\lambda = \frac{B_\Lambda}{u\sqrt{KT}}$, the output $\bar{h}^T = \sum_{t=1}^T h^t/T, \bar{\lambda}^T = \sum_{t=1}^T \lambda^t/T$ comprises an approximate mixed Nash equilibrium:*

$$\max_{\lambda \in \Lambda} \widehat{\mathcal{L}}(\bar{h}^T, \lambda) - \min_{h \in \mathcal{H}} \widehat{\mathcal{L}}(h, \bar{\lambda}^T) \le \rho^T = \bar{\nu}^T + u B_\Lambda \sqrt{\frac{K}{T}}.$$

---

**Algorithm 1** OptFair (In-processing)

**Input:** data $\mathcal{S} = \{x_i, y_i, a_i\}_{i=1}^N$; bound $B_\Lambda$;
  fairness level $\xi$; learning rate $\eta_\theta, \eta_\lambda$;
Compute the empirical statistics $\widehat{\mathbf{D}}^{a,k}$, weight $\widehat{\omega}_a$;
Initialize model $\theta$, parameter $\lambda$;
**for** $t = 0$ **to** $T - 1$ **do**
  Set $\widehat{\mathbf{M}}'(a, \lambda^t) = \widehat{\mathbf{M}}(a, \lambda^t) + \kappa \mathbf{1}_{m \times m}, a \in \mathcal{A}$;
  Performing $R$ steps of gradient update on $\theta^t$ using $\ell_{\text{cal}}$
  (Eq. (13)), yielding $\theta^{t+1}$;
  Update corresponding classifier $h^{t+1}$ with $\theta^{t+1}$;
  Calculate the update of $\lambda^{t+\frac{1}{2}} = \lambda^t + \eta_\lambda \mathscr{D}(h^{t+1})$;
  Performing proximal operator
    $$\lambda^{t+1} = \text{prox}_{\eta_\lambda(\xi\|\lambda\|_1 + \mathbb{I}_{\lambda \in \Lambda})}(\lambda^{t+\frac{1}{2}});$$
**end for**
**Output:** $\{(h^t, \lambda^t)\}_{t=1}^T$;

---

For the fairness criteria in Table 1, the constant $u \le 1 + \xi$. As the number of iterations $T$ grows, this bound decreases to optimization risk $\bar{\nu}^T$, which is also convergent with respect to the inner iteration. Therefore, Theorem 5.2 shows that the parameter solution converges to the empirical equilibrium of accuracy and fairness.

Next, we provide a non-asymptotic analysis of the risk with respect to accuracy and fairness. For $\mathcal{H}$ determined by model $\theta \in \Theta$, we let $\mathcal{H}_y = \{h_y(\cdot) : h \in \mathcal{H}\}$ and assume that the pseudo-dimension (Mohri et al., 2018) of $\mathcal{H}_y$ is finite and satisfies $\max_{y \in [m]} P\dim(\mathcal{H}_y) \le d$.

**Theorem 5.3.** *If $(\bar{h}, \bar{\lambda})$ comprises a $\rho$-approximate Nash equilibrium, denoting the sample number in group $a \in \mathcal{A}$ by $N_a$, $\Omega_k^N = \max_a \|\mathbf{D}^{a,k} - \widehat{\mathbf{D}}^{a,k}\|_\infty$, $u_k = \max_a \|\mathbf{D}^{a,k}\|_1$, $k \in [K]$, and $\gamma_d(n, \frac{1}{\delta}) = \sqrt{\frac{d \log(en/d) + \log(1/\delta)}{n}}$, with probability at least $1 - \delta$,*

$$\mathcal{R}(\bar{h}) \le \mathcal{R}(h^*) + 2\rho + \mathcal{O}\left(\gamma_d(N, \frac{m^2}{\delta})\right),$$

$$|\mathscr{D}_k(\bar{h})| \le \xi + \frac{1 + 2\rho}{B_\Lambda} + \mathcal{O}\left(u_k \sum_{a \in \mathcal{A}} \gamma_d(N_a, \frac{m^2}{\delta}) + \Omega_k^N\right),$$

*where $h^* \in \mathcal{H}$ is an optimal solution for fairness-constrained classification.*

Theorem 5.3 clarifies that the risk of Algorithm 1 originates from two primary sources: the optimization error, represented by the term dependent on $\rho$, and the generalization error, which is related to the sample size $n$ expressed by $\gamma$. Furthermore, it also indicates that as the sample size grows and the optimization error decreases, the model will converge to the accuracy-fairness Pareto frontier.

## 5.2. Plug-in Estimation (Post-Processing)

Post-processing modifies the output of a pre-trained model to ensure fairness by mitigating disparities across different groups, typically without altering the black-box model itself.

Given the dataset $\mathcal{S} = \{x_i, y_i, a_i\}_{i=1}^N$, the plug-in estimator requires the ground-truth conditional probabilities $\mathbb{P}(A, Y \mid X)$. Fortunately, the powerful capabilities of modern neural networks enable us to approximate these agnostic function by training probabilistic classifiers. Previous research (Chen et al., 2024; Xian & Zhao, 2024; Denis et al., 2024; Hu et al., 2025) has shown that such auxiliary models can be flexibly constructed in accordance with the form of the pre-trained model. Therefore, we define the empirical fair classification problem with an entropic regularizer as follows:

$$\min_{h \in \mathcal{H}} \widehat{\mathcal{R}}(h) - \tau \widehat{\mathcal{E}}(h), \quad \text{s.t.} \; |\widehat{\mathscr{D}}_k(h)| \leq \xi, \; k \in [K], \quad (14)$$

where $\widehat{\mathcal{R}}, \widehat{\mathcal{E}}, \widehat{\mathscr{D}}$ are the corresponding empirical functions defined on the empirical distribution $\mu_{\mathcal{S}} \sim \mathcal{P}^N$. Since no assumption is imposed on the sample distribution or the output distribution, the empirical solution can be derived by plugging the estimators into Theorem 4.3.

**Corollary 5.4.** *Assume that $\exists h \in \mathcal{H}$, such that $\widehat{\mathscr{D}}_k(h) = 0$, $k \in [K]$. Denoting $\mathbf{q}^a(x) := [\mathbb{P}(A = a \mid X = x, Y = y)]_{y \in [m]}$, estimated by auxiliary model $\widehat{\mathbf{q}}^a(x) \approx \mathbf{q}^a(x)$, the corresponding classifier*

$$\widehat{h}_i^\lambda(x) := \frac{\exp\left(\widehat{\beta}_i^\lambda(x)/\tau\right)}{\sum_{j=1}^m \exp\left(\widehat{\beta}_j^\lambda(x)/\tau\right)}, \quad i \in [m], \quad (15)$$

$$\widehat{\beta}^\lambda(x) := \left[\sum_{a \in \mathcal{A}} \text{Diag}(\widehat{\mathbf{q}}^a(x))\widehat{\mathbf{M}}(a, \lambda)\right]^\top \widehat{\eta}(x), \quad (16)$$

*define an optimal solution for the empirical regularized fair problem when $\lambda = \widehat{\lambda}^*$, and the parameter $\widehat{\lambda}^*$ is the solution to the empirical dual problem $\min_{\lambda \in \Lambda} \widehat{H}(\lambda)$,*

$$\widehat{H}(\lambda) := \frac{\tau}{N} \sum_{i=1}^N \log\left(\sum_{j=1}^m \exp\left(\widehat{\beta}_j^\lambda(x_i)/\tau\right)\right) + \xi \|\lambda\|_1.$$

**Proposition 5.5.** *Given pre-trained $\widehat{\eta}(x)$ and auxiliary model $\widehat{\mathbb{P}}(A \mid X, Y)$, the empirical objective can be decomposed as $\widehat{H}(\lambda) = \widehat{f}(\lambda) + \xi \|\lambda\|_1$, where $\widehat{f}(\lambda)$ is convex and $L$-smooth with respect to $\lambda$.*

This property of the objective function motivates us to directly apply proximal gradient descent to solve for the optimal value of $\lambda$, which ensures fast convergence for convex and smooth functions (Parikh et al., 2014), as outlined in Algorithm 2. We now proceed with a deeper analysis of the algorithm's generalization risk.

---

**Algorithm 2** OptFair (Post-Processing)

**Input:** data $\mathcal{S} = \{x_i, y_i, a_i\}_{i=1}^N$; bound $B_\Lambda$;
        fairness level $\xi$; learning rate $\eta_\lambda$;
Compute the empirical statistics $\widehat{\mathbf{D}}^{a,k}$, weight $\widehat{\omega}_a$;
Training model $\widehat{\eta}$ and auxiliary model $\widehat{\mathbf{q}}^a(\cdot)$;
Initialize parameter $\lambda$;
**for** $t = 0$ **to** $T - 1$ **do**
     Calculate the update of $\lambda^{t+\frac{1}{2}} = \lambda^t - \eta_\lambda \nabla \widehat{H}(\lambda^t)$;
     Performing proximal operator
$$\lambda^{t+1} = \text{prox}_{\eta_\lambda(\xi\|\lambda\|_1 + \mathbb{I}_{\lambda \in \Lambda})}(\lambda^{t+\frac{1}{2}});$$
**end for**
Obtain fair classifier $\widehat{h}^{\lambda^T}$ from Eq. (15);
**Output:** $\lambda^T, \widehat{h}^{\lambda^T}$;

---

**Theorem 5.6.** (Informal). *Under mild assumptions, with operator norm $\|f\|_1 := \int_{\mathcal{X}} |f| d\mu$, denoting the risk resulting from the auxiliary model, finite samples, and empirical statistics respectively by*

$$\epsilon_1 := |\mathcal{A}|\|\eta - \widehat{\eta}\|_1 + \sum_{a \in \mathcal{A}} \|\mathbf{q}^a - \widehat{\mathbf{q}}^a\|_1, \; \epsilon_2 := \sqrt{\frac{\log(1/\delta)}{N}},$$

$$\epsilon_3 := \sum_{k=1}^K \Omega_k^N, \quad \Omega_k^N = \max_a \|\mathbf{D}^{a,k} - \widehat{\mathbf{D}}^{a,k}\|_\infty,$$

*there exists an integer $N_1 > 0$, such that if the sample number $N \geq N_1$, the empirically optimal fair classifier $\widehat{h}^{\widehat{\lambda}^*}$ determined by $\widehat{\lambda}^* \in \arg\max_{\lambda \in \Lambda} \widehat{H}(\lambda)$ satisfies that, for any $\delta \in (0, 1)$, with a probability of at least $1 - \delta$,*

$$\mathcal{R}(\widehat{h}^{\widehat{\lambda}^*}) \leq \mathcal{R}(h^{\lambda^*}) + \tau \log m + \mathcal{O}\left(\left(\frac{1}{\tau} + 1\right) \sum_{i=1}^3 \epsilon_i\right),$$

$$|\mathscr{D}_k(\widehat{h}^{\widehat{\lambda}^*})| \leq \xi + \mathcal{O}\left(\epsilon_1 + \frac{1}{\tau}\epsilon_2 + \Omega_k^N\right),$$

*and by adjusting $\tau$, the risk of accuracy can achieve*

$$\mathcal{R}(\widehat{h}^{\widehat{\lambda}^*}) \leq \mathcal{R}(h^{\lambda^*}) + \mathcal{O}\left(\psi(\epsilon_1) + \psi(\epsilon_2) + \psi(\epsilon_3)\right),$$

*where $\psi(\epsilon) := \epsilon + \epsilon^{\frac{1}{2}}$ and classifier $h^{\lambda^*} \in \mathcal{H}$ is determined by the optimal solution $\lambda^*$ characterized in Theorem 4.3.*

The complete theorem with its proof is provided in the Appendix C.4. Theorem 5.6 highlights the sources of generalization error in the post-processing algorithm: the gap between the auxiliary model and the true conditional distribution, the generalization error from finite samples, and the discrepancy between the frequency estimator and true statistics.

### 5.3. From Random to Deterministic Classifier

Certain practical scenarios may require the algorithm to return a deterministic classifier, even at the cost of a potential accuracy drop or a mild violation of the fairness

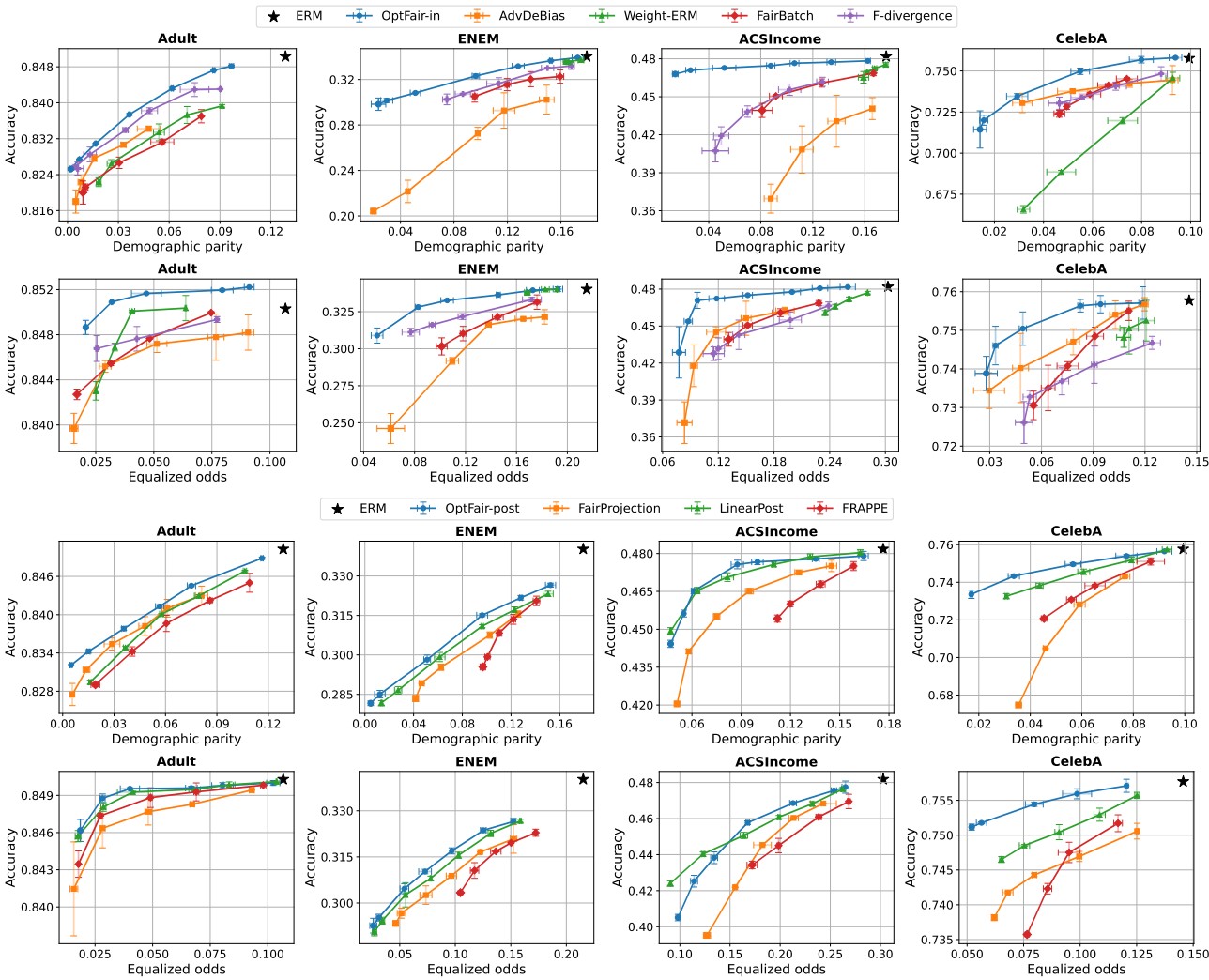

*Figure 1.* Multi-Class Fair Classification Results. The top row shows comparisons between in-processing methods and baselines, whereas the bottom row reports results for post-processing methods. The curve closer to the upper left corner indicates a better trade-off between accuracy and fairness. The error bars for each point represent the 90% confidence interval.

constraints. For the in-processing approach, the learned pair $(\bar{h}, \bar{\lambda})$ can be regarded as an approximate saddle-point solution. Thus, to obtain a deterministic classifier, we fix $\lambda = \bar{\lambda}$ and minimize the cost-sensitive loss in Eq. (13), yielding the classifier closest to the equilibrium solution. For the post-processing approach, a deterministic prediction rule can be directly obtained by taking $\arg\max h(x)$. In practice, $\tau$ is chosen to be small to mitigate the effect of the regularization term on the accuracy-fairness trade-off, as characterized in Theorem 5.6. Consequently, the randomized classifier in Eq. (15) is already nearly deterministic.

# 6. Experiments

We now present the empirical results demonstrating that OptFair achieves competitive performance on accuracy-

fairness trade-off compared to benchmark methods. The detailed information in this section is provided in **Appendix E**, with additional experiments in **Appendix F**.

## 6.1. Datasets and Experimental Settings

**Datasets.** The experiments are conducted on four real-world datasets, including **Adult** (Asuncion et al., 2007), **ENEM** (INEP, 2018), **ACSIncome** (Ding et al., 2021) and **CelebA** (Zhang et al., 2020), which are well established for assessing bias mitigation techniques. Logistic regression is used for Adult, MLPs for ENEM and ACSIncome, and ResNet for CelebA. Except for Adult, which is a binary classification dataset, the tasks constructed on all other datasets involve four or more classes.

**Baselines.** We compare the performance of OptFair with

in- and post-processing baselines, along with classic Empirical Risk Minimization method (ERM): (i) In-processing: **AdvDebias** (Zhang et al., 2018), **Weight-ERM** (Yang et al., 2020), **FairBatch** (Roh et al., 2021), and F-divergence (Zhong & Tandon, 2023); (ii) Post-processing: **FairProjection** (Alghamdi et al., 2022), **LinearPost** (Xian & Zhao, 2024), and **FRAPPÉ** (Tifrea et al., 2024). We select baselines that support multi-class fairness calibration and multiple fairness notions in the attribute-blind setting; thus, binary fair learning and attribute-aware approaches are not included (e.g. Agarwal et al. (2018); Cho et al. (2020); Xian et al. (2023); Denis et al. (2024); Li et al. (2025)).

**Evaluation.** *(i) Firstly*, we partition each dataset into a 60% training set and the remaining 40% as the test set. *(ii) Secondly*, to ensure fair comparison, in-processing and post-processing methods are evaluated separately, since their effectiveness varies across datasets, with our model denoted as `OptFair`-in and `OptFair`-post *(iii) Thirdly*, we evaluate the methods with the Accuracy (Acc) and fairness metric ($\mathscr{D}$), with smaller values of fairness metrics indicating a fairer model.

## 6.2. Overall Comparison

We compare `OptFair` with benchmarks tailored to multi-class classification in terms of the in-processing and post-processing case on the four datasets. The detailed Pareto frontier is reported in Figure 1 to evaluate the ability of `OptFair` to strike the optimal accuracy-fairness balance.

**Comparison results.** Overall, when compared to existing SOTA methods, `OptFair` demonstrates superior performance in achieving a balanced trade-off between accuracy and fairness. We observe that, particularly in the in-processing setting, our method exhibits substantial advantages and offers greater controllability compared to competing methods. This benefit stems from the fact that our approach theoretically approximates the optimal accuracy-fairness trade-off.

**Intrinsic trade-off.** The experiments reveal a fundamental trade-off between fairness and accuracy; imposing fairness as a constraint typically reduces accuracy, consistent with our theoretical results. However, in some cases, such as EO on Adult, accuracy improves, especially with in-processing methods. This is due to the reduction of inherent bias, which enhances model performance, as seen in previous work (Agarwal et al., 2018).

## 6.3. Ablation Study

This subsection presents ablation experiments that combine in-processing and post-processing methods. Since explicit sensitive attributes are difficult to integrate into image

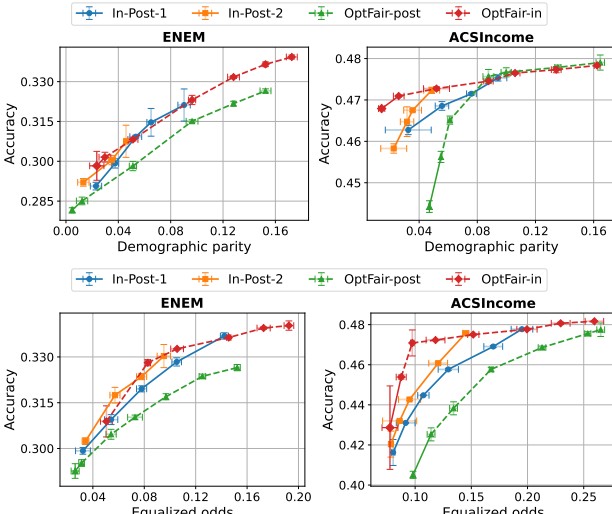

*Figure 2.* Ablation study. Top row: the post-processing module is activated when the DP disparity falls below 0.1 for In-Post-1 and 0.05 for In-Post-2 on both ENEM and ACSIncome. Bottom row: the module is activated when the EO disparity falls below 0.15/0.20 for In-Post-1 and 0.10/0.15 for In-Post-2 on ENEM/ACSIncome, respectively.

datasets, preventing the joint application of both approaches, we focus on tabular datasets and consider Equalized Odds as a representative fairness criterion on ENEM and ACSIncome datasets. Our approach proceeds in two stages: we first train the model using the in-processing method until a predefined fairness threshold is reached, and then apply post-processing to further calibrate fairness. The experimental result is shown in Figure 2.

**In- & post-processing.** The results indicate that, although slight improvements may be observed in some cases, combining the two debiasing methods generally yields performance that lies between the two individual approaches. The in-processing method targets representation-level bias during training at a higher computational cost, whereas post-processing calibrates the model's output distribution without correcting representation-level bias. Therefore, the combination of both methods often fails to yield further improvements in debiasing effectiveness, as demonstrated by the ablation experiments here.

## 7. Discussion

**Relation to previous work.** The optimal fair classifier proposed in this paper extends prior work on generalized fairness definitions and multi-class scenarios. In the binary case, the optimal fair classifier in Theorem 4.2 degenerates into threshold retrieval on $[0, 1]$, aligning with the results from previous binary fairness studies (Menon & Williamson, 2018; Chen et al., 2024). The in-processing method resembles the original reduction approach in the binary case (Agar-

wal et al., 2018), with the primary distinction being that the loss function is formulated as cross-entropy. Furthermore, existing literature (Xian & Zhao, 2024; Denis et al., 2024) establishes that optimal fair classifiers can be obtained by modifying the Bayes-optimal scores (cf. Theorem 4.2 and Example 4.4), and these frameworks typically assume continuity of the output distribution and are confined to post-processing.

**Attribute-aware classifier.** The intervention technique for attribute-aware classifiers also can be derived following the proof technique of Theorem 4.2. In Appendix D, we provide expressions for optimal classifiers under the corresponding fairness constraints, along with the corresponding example. Notably, the attribute-aware Bayes-optimal fair classifier for the DP constraint shares a similar structure with the formulation proposed in Denis et al. (2024).

## 8. Conclusion

This paper studies multi-class fair classification and proposes a group-fairness calibration framework, `OptFair`. We derive an analytically tractable probabilistic characterization of the Bayes-optimal fair classifier, which makes the multi-class accuracy-fairness Pareto frontier explicit and actionable for algorithm design. Based on this characterization, we develop two training-phase compatible *attribute-blind* methods: an in-processing reduction to cost-sensitive classification and a post-processing convex calibration of output probabilities. We provide generalization guarantees showing that both approaches are statistically consistent and converge to the optimal Pareto frontier. Experiments on multiple real-world datasets demonstrate that the proposed method consistently achieves a more controllable accuracy-fairness balance than existing baselines across several group-fairness criteria. Finally, `OptFair` offers a systematic pathway for multi-class fairness calibration and motivates extensions to richer fairness notions and broader deployment settings.

## Reproducibility Statement

Details for the experimental setting are provided in the beginning of Section 6 and Appendix E, and the code can be found at https://github.com/liizhang/OptFair-Multi-Class-Fairness.

## Acknowledgments

This work was supported in part by the National Natural Science Foundation of China (No. 62522217, No. 62402148).

## Impact Statement

This work advances fair machine learning for multi-class classification by providing principled methods to characterize and approximate the optimal accuracy-fairness trade-off. Our framework may benefit high-stakes applications such as healthcare, finance, education, and criminal justice by helping reduce group-level disparities while maintaining predictive utility. It also offers flexibility under different deployment settings, and attribute-blind inference can be useful when sensitive attributes are unavailable or restricted at test time. However, the considered fairness criteria capture only specific group-level notions and may not address individual fairness, intersectional harms, or domain-specific ethical concerns. Therefore, deployment should involve careful validation, monitoring, and transparent reporting of the chosen fairness criteria and limitations.

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

# A. Fairness Criteria and Discussions

## A.1. Group Fairness Criteria and Confusion-Matrix Representations

We clarify several fairness criteria and their representations through confusion matrices, including those in Table 1 (DP, EOP, EO), and additional metrics like Overall Accuracy Equality (OAE) and Mean Equalized Odds (Alghamdi et al., 2022).

For random tuple $(X, Y, A)$, we recall the definition of attribute-blind prediction $\mathbb{P}(\widehat{Y} = y \mid \mathbf{x}) = h_y(\mathbf{x})$, and group-specific confusion matrices $\mathbf{C}^a$, where $a \in \mathcal{A}$, with $\mathbf{C}^a_{i,j} := \mathbb{P}(Y = i, \widehat{Y} = j \mid A = a)$. Moreover, we use $p_\delta$ to denote the probability of event $\delta$ occurring. For example, $p_a := \mathbb{P}(A = a)$, $p_y = \mathbb{P}(Y = y)$, $p_{a,y} := \mathbb{P}(A = a, Y = y)$, $p_{y|a} := \mathbb{P}(Y = y \mid A = a)$, $p_{a|y} := \mathbb{P}(A = a \mid Y = y)$.

**Example A.1.** *For the multi-class DP criterion (Xian et al., 2023; Denis et al., 2024),*

$$\mathscr{D}_{DP} = \max_{y \in [m]} \max_{a \in \mathcal{A}} \left| \mathbb{P}(\widehat{Y} = y \mid A = a) - \mathbb{P}(\widehat{Y} = y) \right|,$$

*where* $\mathbb{P}(Y = y \mid A = a) = \sum_{i \in [m]} \mathbb{P}(\widehat{Y} = y, Y = i \mid A = a) = \sum_{i \in [m]} \mathbf{C}^a_{i,y}$ *and* $\mathbb{P}(\widehat{Y} = y) = \sum_{a' \in \mathcal{A}} \mathbb{P}(A = a') \sum_{i \in [m]} \mathbb{P}(\widehat{Y} = y, Y = i \mid A = a') = \sum_{a' \in \mathcal{A}} \sum_{i \in [m]} \mathbb{P}(A = a') \mathbf{C}^{a'}_{i,y}.$
*Hence, we have*

$$\mathscr{D}_{DP} = \max_{y \in [m]} \max_{a \in \mathcal{A}} \left| \sum_{a' \in \mathcal{A}} \sum_{i \in [m]} \left( \mathbb{I}[a' = a] - \mathbb{P}(A = a') \right) \mathbf{C}^{a'}_{i,y} \right| = \max_{\substack{k = (y,a) \in \mathcal{K} \\ \mathcal{K} = [m] \times \mathcal{A}}} \left| \sum_{a' \in \mathcal{A}} \langle \mathbf{D}^{a',k}, \mathbf{C}^{a'} \rangle \right|,$$

*where* $\mathbf{D}^{a',k} \in \mathbb{R}^{m \times m}$, *and when* $k = (y,a)$, *the $y$-th column elements of $\mathbf{D}^{a',k}$ are $\mathbb{I}[a' = a] - \mathbb{P}(A = a')$ with all other elements set to $0$.*

**Example A.2.** *For the multi-class EOP criterion (Xian & Zhao, 2024),*

$$\mathscr{D}_{EOP} = \max_{y \in [m]} \max_{a \in \mathcal{A}} \left| \mathbb{P}(\widehat{Y} = y \mid A = a, Y = y) - \mathbb{P}(\widehat{Y} = y \mid Y = y) \right|,$$

*where* $\mathbb{P}(\widehat{Y} = y \mid A = a, Y = y) = \frac{\omega_a}{\mathbb{P}(Y = y, A = a)} \mathbf{C}^a_{y,y}$ *and* $\mathbb{P}(\widehat{Y} = y \mid Y = y) = \sum_{a' \in \mathcal{A}} \frac{\mathbb{P}(A = a')}{\mathbb{P}(Y = y)} \mathbf{C}^{a'}_{y,y}.$
*Hence, we have*

$$\mathscr{D}_{EOP} = \max_{y \in [m]} \max_{a \in \mathcal{A}} \left| \sum_{a' \in \mathcal{A}} \omega_{a'} \left( \frac{\mathbb{I}[a' = a]}{\mathbb{P}(Y = y, A = a)} - \frac{1}{\mathbb{P}(Y = y)} \right) \mathbf{C}^{a'}_{y,y} \right| = \max_{\substack{k = (y,a) \in \mathcal{K} \\ \mathcal{K} = [m] \times \mathcal{A}}} \left| \sum_{a' \in \mathcal{A}} \langle \mathbf{D}^{a',k}, \mathbf{C}^{a'} \rangle \right|,$$

*where* $\mathbf{D}^{a',k} \in \mathbb{R}^{m \times m}$, *and when* $k = (y,a)$, *the entry in the $y$-th row and $y$-th column is $\omega_{a'} \left( \frac{\mathbb{I}[a' = a]}{\mathbb{P}(Y = y, A = a)} - \frac{1}{\mathbb{P}(Y = y)} \right)$ with all other elements set to $0$.*

**Example A.3.** *For the multi-class EO criterion (Xian & Zhao, 2024),*

$$\mathscr{D}_{EO} = \max_{y, y' \in [m]} \max_{a \in \mathcal{A}} \left| \mathbb{P}(\widehat{Y} = y \mid A = a, Y = y') - \mathbb{P}(\widehat{Y} = y \mid Y = y') \right|,$$

*where* $\mathbb{P}(\widehat{Y} = y \mid A = a, Y = y') = \frac{\omega_a}{\mathbb{P}(Y = y', A = a)} \mathbf{C}^a_{y',y}$ *and* $\mathbb{P}(\widehat{Y} = y \mid Y = y') = \sum_{a' \in \mathcal{A}} \frac{\mathbb{P}(A = a')}{\mathbb{P}(Y = y')} \mathbf{C}^{a'}_{y',y}.$
*Hence, we have*

$$\mathscr{D}_{EO} = \max_{y, y' \in [m]} \max_{a \in \mathcal{A}} \left| \sum_{a' \in \mathcal{A}} \omega_{a'} \left( \frac{\mathbb{I}[a' = a]}{\mathbb{P}(Y = y', A = a)} - \frac{1}{\mathbb{P}(Y = y')} \right) \mathbf{C}^{a'}_{y',y} \right| = \max_{\substack{k = (y,y',a) \in \mathcal{K} \\ \mathcal{K} = [m] \times [m] \times \mathcal{A}}} \left| \sum_{a' \in \mathcal{A}} \langle \mathbf{D}^{a',k}, \mathbf{C}^{a'} \rangle \right|,$$

*where* $\mathbf{D}^{a',k} \in \mathbb{R}^{m \times m}$, *and when* $k = (y, y', a)$, *the entry in the $y'$-th row and $y$-th column is $\omega_{a'} \left( \frac{\mathbb{I}[a' = a]}{\mathbb{P}(Y = y', A = a)} - \frac{1}{\mathbb{P}(Y = y')} \right)$ with all other elements set to $0$.*

**Example A.4.** *We follow ([Alghamdi et al., 2022](#)) to introduce another natural extension of Equalized Odds in multi-class setting, the Mean Equalized Odds (MEO), and consider its constraint representation:*

$$\mathscr{D}_{MEO} = \max_{y \in [m]} \max_{a \in \mathcal{A}} \frac{1}{2}(|\text{TPR}_y(a) - \text{TPR}_y| + |\text{FPR}_y(a) - \text{FPR}_y|),$$

*where* $\text{TPR}_y(a) = \mathbb{P}(\widehat{Y} = y \mid Y = y, A = a), \text{TPR}_y = \mathbb{P}(\widehat{Y} = y \mid Y = y)$ *and* $\text{FPR}_y(a) = \mathbb{P}(\widehat{Y} = y \mid Y \neq y, A = a), \text{FPR}_y = \mathbb{P}(\widehat{Y} = y \mid Y \neq y)$. *We can obtain that*

$$\mathbb{P}(\widehat{Y} = y \mid Y \neq y, A = a) = \frac{\mathbb{P}(\widehat{Y} = y, Y \neq y \mid A = a)}{\mathbb{P}(Y \neq y \mid A = a)} = \frac{\sum_{i \neq y} \mathbf{C}_{i,y}^a}{\sum_{j \neq y} \mathbb{P}(Y = j \mid A = a)}$$

$$\mathbb{P}(\widehat{Y} = y \mid Y \neq y) = \frac{\mathbb{P}(\widehat{Y} = y, Y \neq y)}{\mathbb{P}(Y \neq y)} = \frac{\sum_{a' \in \mathcal{A}} \omega_{a'} \sum_{i \neq y} \mathbf{C}_{i,y}^{a'}}{\sum_{j \neq y} \mathbb{P}(Y = j)}.$$

*It shows that*

$$\frac{1}{2}(|\text{TPR}_y(a) - \text{TPR}_y| + |\text{FPR}_y(a) - \text{FPR}_y|)$$

$$= \frac{1}{2} \left| \sum_{a' \in \mathcal{A}} \omega_{a'} \left( \frac{1}{\mathbb{P}(Y = y, A = a)} \mathbb{I}[a' = a] - \frac{1}{\mathbb{P}(Y = y)} \right) \mathbf{C}_{y,y}^{a'} \right|$$

$$+ \left| \sum_{a' \in \mathcal{A}} \sum_{i \neq y} \omega_{a'} \left( \frac{1}{\sum_{j \neq y} \mathbb{P}(Y = j, A = a)} \mathbb{I}[a = a'] - \frac{1}{\sum_{j \neq y} \mathbb{P}(Y = j)} \right) \mathbf{C}_{j,y}^{a'} \right|,$$

*where the entry in the $y$-th row and $y$-th column is $\frac{p_{a'}}{p_{a',y}}\mathbb{I}[a = a'] - \frac{p_a}{p_y}$ with all other elements set to $0$ for $\mathbf{D}_{a',y}^{a,0} \in \mathbb{R}^{m \times m}$, and the entry in the $y$-th column is $\frac{p_{a'}}{\sum_{y_j \neq y} p_{a',y_j}}\mathbb{I}[a = a'] - \frac{p_a}{\sum_{y_j \neq y} p_{y_j}}$ except for the $y$-th row with all other elements set to $0$ for $\mathbf{D}_{a',y}^{a,1} \in \mathbb{R}^{m \times m}$.*

**Example A.5.** *For the multi-class fairness criterion Overall Accuracy Equality (OAE),*

$$\mathscr{D}_{OAE} = \max_{a \in \mathcal{A}} \left| \mathbb{P}(\widehat{Y} = Y \mid A = a) - \mathbb{P}(\widehat{Y} = Y) \right|,$$

*where $\mathbb{P}(\widehat{Y} = Y \mid A = a) = \sum_{y \in [m]} \mathbf{C}_{y,y}^a$ and $\mathbb{P}(\widehat{Y} = Y) = \sum_{a' \in \mathcal{A}} \omega_{a'} \sum_{y \in [m]} \mathbf{C}_{y,y}^{a'}$. Hence, we have*

$$\mathscr{D}_{OAE} = \max_{a \in \mathcal{A}} \left| \sum_{a' \in \mathcal{A}} \sum_{y \in [m]} \left( \mathbb{I}(a' = a) - \omega_{a'} \right) \mathbf{C}_{y,y}^{a'} \right| = \max_{k = a \in \mathcal{A}} \left| \sum_{a' \in \mathcal{A}} \langle \mathbf{D}^{a',k}, \mathbf{C}^{a'} \rangle \right|,$$

*where $\mathbf{D}^{a',k} \in \mathbb{R}^{m \times m}$, and when $k = a$, the diagonal entries of the matrix are set to $\mathbb{I}(a' = a) - \omega_{a'}$, with all other elements set to $0$.*

# B. Proofs and Discussions for Section 4

### B.1. Discussion on Assumption 4.1

(**Existence**). Assumption 4.1 requires the existence of a classifier that satisfies the fairness constraint $|\mathscr{D}_k| = 0$. We now discuss the feasibility of this condition for the five fairness metrics mentioned above. First, for the DP, EO, EOP, and MEO criteria, as described in the main text, it is clear that there exist naive classifiers $h(x) = \mathbf{e}_y$, for $y \in [m]$ and $\forall x \in \mathcal{X}$, that satisfy the fairness constraint $|\mathscr{D}_k| = 0$. For the OAE criterion, a clear solution satisfying the constraint is the random classifier $h(x) = \left( \frac{1}{m}, \cdots, \frac{1}{m} \right) \in \Delta_m$ for all $x \in \mathcal{X}$.

## B.2. Proof of Theorem 4.2

### B.2.1. VARIATIONAL FORMULATION AND USEFUL LEMMAS

For clarity of exposition, we first reformulate the Eq. 1 in the variational form. Let $\eta_y(x, a) := \mathbb{P}(Y = y \mid X = x, A = a)$ be the attribute-aware Bayes-optimal score.

For $h \in \mathcal{H} : \mathcal{X} \to \Delta_m$,

$$\mathcal{R}(h) = 1 - \sum_{a \in \mathcal{A}} \omega_a \langle \mathbf{I}, \mathbf{C}^a(h) \rangle = 1 - \sum_{a \in \mathcal{A}} \omega_a \mathbb{E}_{X|A=a}\big[[\eta(X, a)]^\top h(X)\big],$$

$$\mathscr{D}_k(h) = \sum_{a \in \mathcal{A}} \langle \mathbf{D}^{a,k}, \mathbf{C}^a(h) \rangle = \sum_{a \in \mathcal{A}} \mathbb{E}_{X|A=a}\big[[\eta(X, a)]^\top \mathbf{D}^{a,k} h(X)\big],$$

where $\mathbb{E}_{X|A=a}[f(x)] := \int_{\mathcal{X}} f(x) d\mu_a(x)$.

The Bayes-optimal fair classification problem in (1) can be reformulated as

$$
\begin{aligned}
\min_{h \in \mathcal{H}} \quad & 1 - \sum_{a \in \mathcal{A}} \omega_a \mathbb{E}_{X|A=a}\big[[\eta(X, a)]^\top h(X)\big], \\
\text{s.t.} \quad & \left| \sum_{a \in \mathcal{A}} \mathbb{E}_{X|A=a}\big[[\eta(X, a)]^\top \mathbf{D}^{a,k} h(X)\big] \right| \leq \xi, \ k \in \mathcal{K}.
\end{aligned}
\tag{17}
$$

To prove Theorem 4.2, we first introduce some useful definitions and lemmas.

**Definition B.1.** Let $V$ be a real vector space and let $A, B \subseteq V$. The sum of $A$ and $B$ is defined by

$$A + B := \{ a + b \mid a \in A, \ b \in B\}.$$

**Definition B.2.** *(Convex hull.)* For a set $C \subset \mathbb{R}^n$, the convex hull $\mathrm{conv}\{C\}$ consists of all the convex combinations of elements of $C$:

$$\mathrm{conv}\{C\} = \left\{ \sum_{i=0}^p \alpha_i x_i \mid x_i \in C, \alpha_i \geq 0, \sum_{i=0}^p \alpha_i = 1, p \geq 0 \right\}$$

**Lemma B.3.** *(Interchange rule for subdifferentials, see (Bertsekas, 1973).) The subdifferential of the function $F(x) = \mathbb{E}\{f(x, \omega)\}$ at a point $x$ is given by*

$$\partial F(x) = \mathbb{E}\{\partial f(x, \omega)\}$$

*where $f(\cdot, \omega)$ is a real-value convex function and the set $\mathbb{E}\{\partial f(x, \omega)\}$ is defined as*

$$
\begin{aligned}
\mathbb{E}\{\partial f(x, \omega)\} &:= \int_\Omega \partial f(x, \omega) d\mathcal{P}(\omega) \\
&= \left\{ \theta^* \in \mathbb{R}^n : \theta^* = \int_\Omega \theta^*(\omega) d\mathcal{P}(\omega), \theta^*(\cdot) : measurable, \theta^*(\omega) \in \partial f(x, \omega) \ a.e. \right\}.
\end{aligned}
$$

**Lemma B.4.** *(Danskin's theorem, see (Rockafellar & Wets, 2009).) Let $f_1, \ldots, f_m : \mathbb{R}^n \to (-\infty, +\infty]$ be convex functions. Define $f(x) = \max\{f_1(x), \ldots, f_m(x)\}, \quad \forall x \in \mathbb{R}^n$. For $\theta \in \bigcap_{i=1}^m dom f_i$, define $I(\theta) = \{i \mid f_i(\theta) = f(\theta)\}$. Then,*

$$\partial f(\theta) = \mathrm{conv}\{\partial f_i(\theta) : i \in I(\theta)\}.$$

*where $\mathrm{conv}\{\cdot\}$ indicates the convex hull operation.*

**Lemma B.5.** *(First-Order Optimality condition, see (Rockafellar & Wets, 2009).) Let $f : \mathbb{R}^d \to \mathbb{R}$ be a convex continuous function. We consider the minimizer $\theta^*$ of the function $f$ over the set $B$. Then for $\theta^*$ to be locally optimal it is necessary that*

$$\partial f(\theta^*) + \mathcal{N}_B(\theta^*) \ni 0,$$

*where $\mathcal{N}_B$ denotes the normal cone of set $B$.*

B.2.2. COMPLETE PROOF OF THEOREM 4.2

For the primal problem Eq. (17), it follows that the formal Lagrange function can be written as

$$
\mathcal{L}(h, \lambda^{(1)}, \lambda^{(2)})
$$

$$
= 1 - \sum_{a \in \mathcal{A}} \omega_a \mathbb{E}_{X|A=a} \left[ [\eta(X,a)]^\top h(X) \right] + \sum_{k=1}^{K} (\lambda_k^{(1)} - \lambda_k^{(2)}) \sum_{a \in \mathcal{A}} \mathbb{E}_{X|A=a} \left[ [\eta(X,a)]^\top \mathbf{D}^{a,k} h(X) \right]
$$

$$
- \xi \sum_{k=1}^{K} (\lambda_k^{(1)} + \lambda_k^{(2)})
$$

$$
= 1 + \sum_{a \in \mathcal{A}} \mathbb{E}_{X|A=a} \left[ [\eta(X,a)]^\top \left( \sum_{k=1}^{K} (\lambda_k^{(1)} - \lambda_k^{(2)}) \mathbf{D}^{a,k} - \omega_a \mathbf{I} \right) h(X) \right] - \xi \sum_{k=1}^{K} (\lambda_k^{(1)} + \lambda_k^{(2)}),
$$

where $\lambda^{(1)}, \lambda^{(2)} \in \mathbb{R}_{\geq 0}^K$ are the dual parameters. We first characterize the explicit form of the dual problem.

**(Duality).** The dual problem can be written as

$$
\max_{\lambda^{(1)}, \lambda^{(2)} \in \mathbb{R}_{\geq 0}^K} \min_{h \in \mathcal{H}} L(h, \lambda^{(1)}, \lambda^{(2)}).
$$

With $\mathbf{M}(a, \lambda) = \mathbf{I} - \frac{1}{\omega_a} \sum_{k=1}^{K} (\lambda_k^{(1)} - \lambda_k^{(2)}) \mathbf{D}^{a,k}$, after removing the terms unrelated to the classifier $h$, the inner problem $\min_{h \in \mathcal{H}} L(h, \lambda^{(1)}, \lambda^{(2)})$ turns to

$$
\max_{h \in \mathcal{H}} \sum_{a \in \mathcal{A}} \omega_a \mathbb{E}_{X|A=a} \left[ [\eta(X,a)]^\top \mathbf{M}(a, \lambda) h(X) \right].
$$

Leveraging the definition of conditional expectation and the law of total expectation, the inner problem can be converted into a solvable form:

$$
\sum_{a \in \mathcal{A}} \omega_a \mathbb{E}_{X|A=a} \left[ [\eta(X,a)]^\top \mathbf{M}(a, \lambda) h(X) \right]
$$

$$
= \sum_{a \in \mathcal{A}} P(A = a) \mathbb{E}_{X|A=a} \left[ [\eta(X,a)]^\top \mathbf{M}(a, \lambda) h(X) \right]
$$

$$
= \mathbb{E}_A \left[ \mathbb{E}_{X|A} \left[ [\eta(X,A)]^\top \mathbf{M}(A, \lambda) h(X) \right] \right)
$$

$$
= \mathbb{E}_{X,A} \left[ [\eta(X,A)]^\top \mathbf{M}(A, \lambda) h(X) \right]
$$

$$
= \mathbb{E}_X \left[ \mathbb{E}_{A|X} \left[ [\eta(X,A)]^\top \mathbf{M}(A, \lambda) h(X) \right] \right]
$$

$$
= \mathbb{E}_X \left[ \sum_{a \in \mathcal{A}} P(A = a|X) [\eta(X,a)]^\top \mathbf{M}(a, \lambda) h(X) \right]
$$

$$
= \mathbb{E}_X \left[ \left( \sum_{a \in \mathcal{A}} P(A = a|X) [\eta(X,a)]^\top \mathbf{M}(a, \lambda) \right) h(X) \right].
$$

Given $(\lambda^{(1)}, \lambda^{(2)})$, it is explicit that the inner problem has a solution

$$
h(x) \in \text{conv} \left\{ e_y : y \in \arg\max_{i \in [m]} \left( \sum_{a \in \mathcal{A}} P(A = a|X) [\eta(X,a)]^\top \mathbf{M}(a, \lambda) \right)_i \right\}
$$

The definition of convex hull (conv) follows Def. B.2.

Thus, we can derive the dual problem as $\min_{\lambda^{(1)}, \lambda^{(2)} \in \mathbb{R}_{\geq 0}^K} H(\lambda^{(1)}, \lambda^{(2)})$, and

$$
H(\lambda^{(1)}, \lambda^{(2)}) = \mathbb{E}_X \left[ \max_{i \in [m]} \left( \sum_{a \in \mathcal{A}} \mathbb{P}(A = a|X) [\eta(X,a)]^\top \mathbf{M}(\lambda, a) \right)_i \right] + \xi \| \lambda^{(1)} + \lambda^{(2)} \|_1.
$$

Since the dual problem minimizes the $H(\lambda^{(1)}, \lambda^{(2)})$ subject to $\lambda^{(1)}, \lambda^{(2)} \geq 0$ element-wise, it follows that the optimal dual parameter $(\lambda^{(1)}, \lambda^{(2)})$ satisfies $\lambda_k^{(1)} \lambda_k^{(2)} = 0$, for every $k \in \mathcal{K}$; that is, if $\lambda_k^{(1)} > 0$, then $\lambda_k^{(2)} = 0$, and if $\lambda_k^{(2)} > 0$, then $\lambda_k^{(1)} = 0$. This result can be derived via a proof by contradiction: If $\lambda_k^{(1)} > \lambda_k^{(2)} > 0$ in the optimal dual parameter $(\lambda^{(1)}, \lambda^{(2)})$, defining $\lambda_k^{(1)*} := \lambda_k^{(1)} - \lambda_k^{(2)}$, $\lambda_k^{(2)*} := 0$, it is explicit that, with other elements unchanged, the pair $(\lambda_k^{(1)*}, \lambda_k^{(2)*})$ yields a strictly smaller value of the dual function $H$ compared to $(\lambda_k^{(1)}, \lambda_k^{(2)})$, which implicates that the parameter $(\lambda^{(1)}, \lambda^{(2)})$ is not optimal.

Therefore, denoting $\lambda = \lambda^{(1)} - \lambda^{(2)} \in \mathbb{R}^K$, the dual problem can be reduced to $\min_{\lambda \in \mathbb{R}^K} H(\lambda)$, and

$$H(\lambda) = \mathbb{E}_X \left[ \max_{i \in [m]} \left( \sum_{a \in \mathcal{A}} \mathbb{P}(A = a | X)[\eta(X, a)]^\top \mathbf{M}(\lambda, a) \right)_i \right] + \xi \|\lambda\|_1,$$

where $\mathbf{M}(a, \lambda) = \omega_a \mathbf{I} - \sum_{k=1}^K \lambda_k \mathbf{D}^{a,k}$.

Now, we discuss whether the primal problem is equivalent to the dual problem, i.e., whether strong duality holds. The standard approach is to apply relaxed Slater's condition or the Fenchel-Rockafellar theorem, which will be discussed later. However, since our domain $\mathcal{H} : \mathcal{X} \to \Delta_m$ is not even a linear vector space, we cannot rely on those theorems to prove strong duality. Thanks to its closed-form expression, we directly show that there is no duality gap between the dual and primal problems.

**(Boundness.)** First let us show that any optimal $\lambda^*$ is bounded. Since there exists a classifier that is feasible for $\xi = 0$, the feasible set of the primal problem is non-empty for every $\xi > 0$. With a slight abuse of notation, considering one of the optimal classifiers $h_\xi^*$ under fairness constraint $\xi > 0$ and the classifier $h'$ that is feasible for $\xi = 0$, the duality gap between the primal and dual problem tells us that

$$\max_{\lambda \in \mathbb{R}^K} \min_{h \in \mathcal{H}} L(h, \lambda) = \max_{\lambda \in \mathbb{R}^K} 1 - H(\lambda) \leq \mathcal{R}(h_\xi^*) \leq \mathcal{R}(h'),$$

Note that given $\lambda$, the solution to the inner optimization does not depend on $\xi$. Hence, for $\xi > 0$,

$$H(\lambda) = \mathbb{E}_X \left[ \max_{i \in [m]} \left( \sum_{a \in \mathcal{A}} \mathbb{P}(A = a | X)[\eta(X, a)]^\top \mathbf{M}(\lambda, a) \right)_i \right] + \xi \|\lambda\|_1 \geq 1 - \mathcal{R}(h').$$

Taking the limit $\xi \to 0$, we obtain that $H(\lambda) \geq 1 + \xi \|\lambda\|_1 - \mathcal{R}(h')$. As $\|\lambda\|_1 \to \infty$, it follows that $H(\lambda) \to \infty$, which contradicts the objective of minimizing $H(\lambda)$. Therefore, we obtain that the parameter $\lambda$ must be $L_1$-bounded.

**(Feasibility.)** Next, it is necessary to prove that the solution induced by the dual problem is feasible for the primal problem. Note that we have denoted

$$\beta^\lambda(x) := \sum_{a \in \mathcal{A}} \mathbb{P}(A = a | X = x)[\eta(x, a)]^\top \mathbf{M}(\lambda, a) \in \mathbb{R}^m.$$

To present the proof more clearly, we consider the sub-differential of $H(\lambda)$ with respect to the $k$-th element of $\lambda$. Since $H(\lambda)$ is convex to parameters $\lambda$, the sub-differential is given by

$$\frac{\partial}{\partial \lambda_k} H(\lambda) = \frac{\partial}{\partial \lambda_k} \mathbb{E}_X \left[ \max_{i \in [m]} \left( \beta^\lambda(X) \right)_i \right] + \xi \partial |\lambda_k|$$

$$= \mathbb{E}_X \left[ \frac{\partial}{\partial \lambda_k} \max_{i \in [m]} \left( \beta^\lambda(X) \right)_i \right] + \xi \partial |\lambda_k|$$

$$= \left\{ \int_{\mathcal{X}} \text{conv} \left( -\sum_{a \in \mathcal{A}} \mathbb{P}(A = a | X = x)[\eta(x, a)]^\top \mathbf{D}^{a,k} e_i : i \in \arg\max_i(\beta^\lambda(x)) \right) d\mathbb{P}(x) \right\} + \xi \partial |\lambda_k|.$$

The second equation is due to Lemma B.3, and the third equation is by Lemma B.4. The addition here is defined by Def. B.1. Let the probabilistic classifier $h_\lambda(x) \in \text{conv}\{e_y : y \in \arg\max_{i \in [m]} \beta^\lambda(x)\}$. Due to the linearity, the sub-differential for

$\lambda_k$ can be formulated as

$$
\frac{\partial}{\partial \lambda_k} H(\lambda)
$$

$$
= \left\{ -\int_{\mathcal{X}} \sum_{a \in \mathcal{A}} \mathbb{P}(A=a|X=x)[\eta(x,a)]^{\top} \mathbf{D}^{a,k} h_\lambda(x) d\mathbb{P}(x) : h_\lambda(x) \in \mathrm{conv}\{e_y : y \in \arg\max_{i \in [m]} \beta^\lambda(x)\} \right\} + \xi \partial |\lambda_k|
$$

$$
= \left\{ -\mathbb{E}_X \left[ \sum_{a \in \mathcal{A}} \mathbb{P}(A=a|X)[\eta(x,a)]^{\top} \mathbf{D}^{a,k} h_\lambda(x) \right] : h_\lambda(x) \in \mathrm{conv}\{e_y : y \in \arg\max_{i \in [m]} \beta^\lambda(x)\} \right\} + \xi \partial |\lambda_k|
$$

$$
= \left\{ -\mathscr{D}_k(h_\lambda(x)) : h_\lambda(x) \in \mathrm{conv}\{e_y : y \in \arg\max_{i \in [m]} \beta^\lambda(x)\} \right\} + \xi \partial |\lambda_k|.
$$

Note that the choice of $h_\lambda$ does not depend on $k \in \mathcal{K}$. Hence, we obtain the sub-differential of $H$ with respect to $\lambda$,

$$
\frac{\partial}{\partial \lambda} H(\lambda) = \left\{ -\mathscr{D}_{1:K}(h_\lambda(x)) : h_\lambda(x) \in \mathrm{conv}\{e_y : y \in \arg\max_{i \in [m]} \beta^\lambda(x)\} \right\} + \xi \partial \|\lambda\|_1,
$$

where $\mathscr{D}_{1:K}$ denotes a vector of length $k$, with the $k$-th element given by $\mathscr{D}_k$. This formulation naturally clarifies that each sub-gradient in $\partial H$ can be characterized by dual parameter $\lambda$ and decision $h_\lambda$.

By Lemma B.5, the optimality condition for the dual problem implies that $0 \in \partial H(\lambda^*)$. In other words, there exists at least one decision $h_{\lambda^*}^*(x), x \in \mathcal{X}$ that meets the optimality condition. Given optimal $\lambda^*$ and corresponding $h_{\lambda^*}^*$, if $\lambda_k^* > 0$, the $k$-th sub-differential reduces to $-\mathscr{D}_k(h_{\lambda^*}^*) + \xi$, which implies $\mathscr{D}_k(h_{\lambda^*}^*) = \xi$. Similarly, if $\lambda_k^* < 0$, the $k$-th sub-differential reduces to $-\mathscr{D}_k(h_{\lambda^*}^*) - \xi$, which implies $\mathscr{D}_k(h_{\lambda^*}^*) = -\xi$. Besides, if $\lambda_k^* = 0$, it follows that $0 \in -\mathscr{D}_k(h_{\lambda^*}^*) + [-\xi, \xi]$, namely $\mathscr{D}_k(h_{\lambda^*}^*) \in [-\xi, \xi]$.

Overall, we have shown that for all $k \in \mathcal{K}$, $|\mathscr{D}_k(h_{\lambda^*}^*)| \leq \xi$, which implies the feasibility.

**(Optimality.)** The last step is to prove that the classifier $h_{\lambda^*}^*$ above is the optimal solution of the primal problem. Defining $(\lambda_k^{*(1)}, \lambda_k^{*(2)})$ by $\lambda_k^{*(1)} := |\lambda_k^*| \mathbb{I}(\lambda_k^* > 0)$ and $\lambda_k^{*(2)} := |\lambda_k^*| \mathbb{I}(\lambda_k^* < 0)$, it is obvious that the pair also satisfies the optimality conditions for the dual problem given $h_{\lambda^*}^*$. From the proof above, we can obtain that, for $\forall k \in \mathcal{K}$,

$$
(\lambda_k^{(1)} - \lambda_k^{(2)}) \mathscr{D}_k(h_{\lambda^*}^*) - (\lambda_k^{(1)} + \lambda_k^{(2)}) \xi = 0,
$$

which satisfies the optimality conditions for the dual solution of the constrained optimization problem. Consequently, the Lagrangian function equals the risk function when plugging in the optimal classifier, $\mathcal{L}(h_{\lambda^*}^*, \lambda^{*(1)}, \lambda^{*(2)}) = \mathcal{R}(h_{\lambda^*}^*)$.

For any other classifier $h' \in \mathcal{H}$ that satisfies the fairness constraints, denoting its corresponding dual parameter that maximizes the dual problem as $(\lambda^{'(1)}, \lambda^{'(2)})$, it can be deduced that

$$
\mathcal{L}(h_{\lambda^*}^*, \lambda^{*(1)}, \lambda^{*(2)}) \leq \mathcal{L}(h', \lambda^{'(1)}, \lambda^{'(2)}) \leq \mathcal{R}(h').
$$

Therefore, we arrive at

$$
\mathcal{R}(h_{\lambda^*}^*) = \mathcal{L}(h_{\lambda^*}^*, \lambda^{*(1)}, \lambda^{*(2)}) \leq \mathcal{R}(h').
$$

This completes the proof. $\qquad\square$

## B.3. Proof of theorem 4.3

### B.3.1. USEFUL LEMMA

**Lemma B.6.** *Let $\alpha \in \mathbb{R}^m$ be a vector that satisfies the constraints $\sum_{i=1}^m \alpha_i = 1$ and $\alpha_i \geq 0$. Consider the optimization problem:*

$$
\max_{\alpha \in \mathbb{R}^m} f(\alpha) = \sum_{i=1}^m \beta_i \alpha_i - \rho \alpha_i \log \alpha_i \quad s.t. \quad \sum_{i=1}^m \alpha_i = 1, \quad \alpha_i \geq 0,
$$

*where $\rho > 0$ is a constant. The solution to this problem is given by:*

$$\alpha_i^* = \frac{e^{\frac{\beta_i}{\rho}}}{\sum_{j=1}^m e^{\frac{\beta_j}{\rho}}}, \quad for \quad i = 1, \ldots, m.$$

*The maximum value of the objective function is:*

$$f(\alpha^*) = \rho \log \left( \sum_{j=1}^m e^{\frac{\beta_j}{\rho}} \right).$$

*Proof of Lemma B.6.* We begin by defining the Lagrangian of the problem. The Lagrangian for this constrained optimization problem is:

$$\mathcal{L}(\alpha, \lambda, \mu) = \sum_{i=1}^m \left( \beta_i \alpha_i - \rho \alpha_i \log \alpha_i \right) + \lambda \left( \sum_{i=1}^m \alpha_i - 1 \right) + \sum_{i=1}^m \mu_i \alpha_i,$$

where $\lambda$ and $\mu_i$ are Lagrange multipliers associated with the constraints $\sum_{i=1}^m \alpha_i = 1$ and $\alpha_i \geq 0$, respectively. To find the optimal solution, we compute the partial derivative of the Lagrangian with respect to $\alpha_i$ and set it equal to zero,

$$\frac{\partial \mathcal{L}}{\partial \alpha_i} = \beta_i - \rho(1 + \log \alpha_i) + \lambda = 0.$$

Solving for $\log \alpha_i$, we get

$$\log \alpha_i = \frac{\beta_i + \lambda - \rho}{\rho}.$$

Exponentiating both sides,

$$\alpha_i = e^{\frac{\beta_i + \lambda - \rho}{\rho}} = C e^{\frac{\beta_i}{\rho}},$$

where $C = e^{\frac{\lambda - \rho}{\rho}}$ is a constant that we will determine using the constraint $\sum_{i=1}^m \alpha_i = 1$. Next, we impose the constraint $\sum_{i=1}^m \alpha_i = 1$,

$$\sum_{i=1}^m \alpha_i = \sum_{i=1}^m C e^{\frac{\beta_i}{\rho}} = C \sum_{i=1}^m e^{\frac{\beta_i}{\rho}} = 1.$$

Solving for $C$, we get

$$C = \left( \sum_{i=1}^m e^{\frac{\beta_i}{\rho}} \right)^{-1}.$$

Thus, the optimal solution for $\alpha_i$ is:

$$\alpha_i^* = \frac{e^{\frac{\beta_i}{\rho}}}{\sum_{j=1}^m e^{\frac{\beta_j}{\rho}}}.$$

Finally, substituting this solution into the objective function, the maximum value is:

$$f(\alpha^*) = \rho \log \left( \sum_{j=1}^m e^{\frac{\beta_j}{\rho}} \right).$$

Thus, the proof is complete. $\square$

B.3.2. COMPLETE PROOF OF THEOREM 4.3

Denoting $\rho_a(x) := \mathbb{P}(A = a \mid X = x)$, as shown in the proof of Theorem 4.2, the optimization objective can be formulated as

$$\min_{h \in \mathcal{H}} \quad 1 - \mathbb{E}_X\left[\sum_{a \in \mathcal{A}} \rho_a(X)[\eta(X,a)]^\top h(X)\right] + \tau \mathbb{E}_X\left[\sum_{i=1}^m h_i(X)\log h_i(X)\right],$$

$$\text{s.t.} \quad \left|\mathbb{E}_X\left[\sum_{a \in \mathcal{A}} \frac{1}{\omega_a}\rho_a(X)[\eta(X,a)]^\top \mathbf{D}^{a,k}h(X)\right]\right| \le \xi, \ k \in \mathcal{K}. \tag{18}$$

It follows that the formal Lagrange function can be written as

$$L(h,\lambda^{(1)},\lambda^{(2)})$$

$$= 1 + \mathbb{E}_X\left[\sum_{a \in \mathcal{A}} \rho_a(X)[\eta(X,a)]^\top \left(\frac{1}{\omega_a}\sum_{k=1}^K (\lambda_k^{(1)} - \lambda_k^{(2)})\mathbf{D}^{a,k} - \mathbf{I}\right)h(X)\right]$$

$$+ \tau \mathbb{E}_X\left[\sum_{i=1}^m h_i(X)\log h_i(X)\right] - \xi \sum_{k=1}^K (\lambda_k^{(1)} + \lambda_k^{(2)})$$

$$= 1 - \mathbb{E}_X\left[\sum_{a \in \mathcal{A}} \rho_a(X)[\eta(X,a)]^\top \mathbf{M}(a,\lambda)h(X) - \tau\sum_{i=1}^m h_i(X)\log h_i(X)\right] - \xi\sum_{k=1}^K (\lambda_k^{(1)} + \lambda_k^{(2)}),$$

$$= 1 - \mathbb{E}_X\left[(\beta^\lambda(X))^\top h(X) - \tau\sum_{i=1}^m h_i(X)\log h_i(X)\right] - \xi\sum_{k=1}^K (\lambda_k^{(1)} + \lambda_k^{(2)}),$$

where $\mathbf{M}(a,\lambda) := \mathbf{I} - \frac{1}{\omega_a}\sum_{k=1}^K (\lambda_k^{(1)} - \lambda_k^{(2)})\mathbf{D}^{a,k}$, and $\lambda^{(1)}, \lambda^{(2)} \in \mathbb{R}_{\ge 0}^K$ are the dual parameters. We first characterize the explicit form of the dual problem.

**(Duality).** The dual problem can be written as

$$\max_{\lambda^{(1)},\lambda^{(2)}\in\mathbb{R}_{\ge 0}^K} \min_{h \in \mathcal{H}} L(h,\lambda^{(1)},\lambda^{(2)}).$$

Considering the inner optimization, after removing the terms unrelated to the classifier $h$, the inner problem $\min_{h \in \mathcal{H}} L(h,\lambda^{(1)},\lambda^{(2)})$ turns to

$$\max_{h \in \mathcal{H}} \quad \mathbb{E}_X\left[\sum_{i=1}^m \beta_i^\lambda(X)h_i(X) - \tau h_i(X)\log h_i(X)\right].$$

By Lemma B.6, it is explicit that there exists an optimal solution

$$h_i^\star(x) = \frac{\exp\left(\beta_i^\lambda(x)/\tau\right)}{\sum_{j=1}^m \exp\left(\beta_j^\lambda(x)/\tau\right)}, \quad i = 1,\ldots,m.$$

Plugged the $h^*$ into the inner optimization, the maximum value of the objective above is

$$\max_{h \in \mathcal{H}} \mathbb{E}_X\left[\sum_{i=1}^m \beta_i^\lambda(X)h_i(X) - \tau h_i(X)\log h_i(X)\right] = \mathbb{E}_X\left[\tau\log\sum_{j=1}^m \exp\left(\beta_j^\lambda(X)/\tau\right)\right].$$

This result can equivalently be obtained through the Gibbs variational principle (Alquier, 2024). Consequently, the dual problem can be derived as $\min_{\lambda^{(1)},\lambda^{(2)}\in\mathbb{R}_{\ge 0}^K} H(\lambda^{(1)},\lambda^{(2)})$, and

$$H(\lambda^{(1)},\lambda^{(2)}) = \mathbb{E}_X\left[\tau\log\sum_{j=1}^m \exp\left(\beta_j^\lambda(X)/\tau\right)\right] + \xi\|\lambda^{(1)} + \lambda^{(2)}\|_1.$$

Following the similar derivation in Theorem 4.2, we know that the optimal dual parameters $(\lambda^{(1)}, \lambda^{(2)})$ satisfy $\lambda_k^{(1)}\lambda_k^{(2)} = 0$, for every $k \in \mathcal{K}$; that is, if $\lambda_k^{(1)} > 0$, then $\lambda_k^{(2)} = 0$, and if $\lambda_k^{(2)} > 0$, then $\lambda_k^{(1)} = 0$. Therefore, denoting $\lambda = \lambda^{(1)} - \lambda^{(2)} \in \mathbb{R}^K$, the dual problem can be reduced to $\min_{\lambda \in \mathbb{R}^K} H(\lambda)$, and

$$H(\lambda) = \mathbb{E}_X\left[\tau \log \sum_{j=1}^m \exp\left(\beta_j^\lambda(X)/\tau\right)\right] + \xi\|\lambda\|_1.$$

We now follow the proof of Theorem 4.2 to show that the duality gap between the primal and dual problem is zero.

**(Boundness.)** It should be noted that entropy regularization is introduced only into the optimization objective, while the feasible domain of the original problem remains unchanged. Consequently, by adhering to the identical proof strategy as in Theorem 4.2, the boundedness property of the dual parameters follows directly.

**(Feasibility.)** Compared to the optimization problem without entropy regularization, a notable improvement here is that the dual problem is smoother. Considering the differential of $\beta^\lambda(x)$ with respect to the $k$-th element of $\lambda$, it is given by

$$\frac{\partial \beta^\lambda(x)}{\partial \lambda_k} = \sum_{a \in \mathcal{A}} \rho_a(x)[\eta(x,a)]^\top \frac{\partial \mathbf{M}(\lambda, a)}{\partial \lambda_k} = -\sum_{a \in \mathcal{A}} \frac{1}{\omega_a} \rho_a(x)[\eta(x,a)]^\top \mathbf{D}^{a,k}.$$

Thus, the sub-differential of $H(\lambda)$ with respect to $\lambda_k$ is given by

$$\begin{aligned}
\frac{\partial}{\partial \lambda_k} H(\lambda) &= \frac{\partial}{\partial \lambda_k} \mathbb{E}_X\left[\tau \log \sum_{j=1}^m \exp\left(\beta_j^\lambda(X)/\tau\right)\right] + \xi\partial|\lambda_k| \\
&= \mathbb{E}_X\left[\frac{\partial}{\partial \lambda_k} \tau \log \sum_{j=1}^m \exp\left(\beta_j^\lambda(X)/\tau\right)\right] + \xi\partial|\lambda_k| \\
&= \mathbb{E}_X\left[\tau \frac{\frac{1}{\tau}\sum_{i=1}^m \exp(\beta_i^\lambda(X)/\tau)\frac{\partial\beta_i^\lambda(X)}{\partial\lambda_k}}{\sum_{j=1}^m \exp(\beta_j^\lambda(X)/\tau)}\right] + \xi\partial|\lambda_k| \\
&= \mathbb{E}_X\left[\sum_{i=1}^m \frac{\exp(\beta_i^\lambda(X)/\tau)}{\sum_{j=1}^m \exp(\beta_j^\lambda(X)/\tau)}\left(-\sum_{a \in \mathcal{A}} \frac{1}{\omega_a}\rho_a(x)[\eta(X,a)]^\top \mathbf{D}^{a,k}\right)_i\right] + \xi\partial|\lambda_k| \\
&= \mathbb{E}_X\left[\sum_{i=1}^m [h_\lambda^*(X)]_i\left(-\sum_{a \in \mathcal{A}} \frac{1}{\omega_a}\rho_a(x)[\eta(X,a)]^\top \mathbf{D}^{a,k}\right)_i\right] + \xi\partial|\lambda_k| \\
&= -\mathbb{E}_X\left[\sum_{a \in \mathcal{A}} \frac{\rho_a(x)}{\omega_a}[\eta(X,a)]^\top \mathbf{D}^{a,k} h_\lambda^*(X)\right] + \xi\partial|\lambda_k| \\
&= -\sum_{a \in \mathcal{A}} \mathbb{E}_{X|A=a}\left[[\eta(X,a)]^\top \mathbf{D}^{a,k} h_\lambda^*(X)\right] + \xi\partial|\lambda_k| \\
&= -\mathscr{D}_k(h_\lambda^*) + \xi\partial|\lambda_k|.
\end{aligned}$$

The second equation is due to Lemma B.3, and the addition between real value and set here is also defined by Def. B.1. Hence, we obtain the sub-differential of $H$ with respect to $\lambda$,

$$\frac{\partial}{\partial \lambda} H(\lambda) = -\mathscr{D}_{1:K}(h_\lambda^*) + \xi\partial\|\lambda\|_1,$$

where $\mathscr{D}_{1:K}$ denotes a vector of length $k$, with the $k$-th element given by $\mathscr{D}_k$. This formulation indicates that the sub-gradient of the dual function $H(\lambda)$ depends solely on the value of $\lambda$.

By Lemma B.5, the optimality condition for the dual problem implies that $0 \in \partial H(\lambda^*)$. Given optimal $\lambda^*$ and corresponding $h_{\lambda^*}^*$, if $\lambda_k^* > 0$, the $k$-th sub-differential reduces to $-\mathscr{D}_k(h_{\lambda^*}^*) + \xi$, which implies $\mathscr{D}_k(h_{\lambda^*}^*) = \xi$. Similarly, if $\lambda_k^* < 0$, the $k$-th sub-differential reduces to $-\mathscr{D}_k(h_{\lambda^*}^*) - \xi$, which implies $\mathscr{D}_k(h_{\lambda^*}^*) = -\xi$. Besides, if $\lambda_k^* = 0$, it follows that $0 \in -\mathscr{D}_k(h_{\lambda^*}^*) + [-\xi, \xi]$, namely $\mathscr{D}_k(h_{\lambda^*}^*) \in [-\xi, \xi]$.

Overall, we have shown that for all $k \in \mathcal{K}$, $|\mathcal{D}_k(h^*_{\lambda^*})| \leq \xi$, which implies the feasibility.

**(Optimality.)** Let $h^*_{\lambda^*}$ denote the classifier induced by the optimal dual parameter $\lambda^*$. Defining $(\lambda_k^{*(1)}, \lambda_k^{*(2)})$ by $\lambda_k^{*(1)} :=$ $|\lambda_k^*|\mathbb{I}(\lambda_k^* > 0)$ and $\lambda_k^{*(2)} := |\lambda_k^*|\mathbb{I}(\lambda_k^* < 0)$, it is obvious that the pair ensures feasibility of both primal and dual problems. In particular, the complementary slackness condition implies that

$$(\lambda_k^{(1)} - \lambda_k^{(2)})\mathcal{D}_k(h^*_{\lambda^*}) - (\lambda_k^{(1)} + \lambda_k^{(2)})\xi = 0, \quad \forall k \in \mathcal{K},$$

so that the Lagrangian coincides with the primal objective at $h^*_{\lambda^*}$:

$$\mathcal{L}(h^*_{\lambda^*}, \lambda^{*(1)}, \lambda^{*(2)}) = \mathcal{R}(h^*_{\lambda^*}).$$

For any other feasible classifier $h' \in \mathcal{H}$, together with its maximizing dual variable $(\lambda^{'(1)}, \lambda^{'(2)})$, weak duality yields

$$\mathcal{R}(h^*_{\lambda^*}) = \mathcal{L}(h^*_{\lambda^*}, \lambda^{*(1)}, \lambda^{*(2)}) \leq \mathcal{L}(h', \lambda^{'(1)}, \lambda^{'(2)}) \leq \mathcal{R}(h').$$

Hence $h^*_{\lambda^*}$ minimizes the risk among all feasible classifiers, completing the proof. □

### B.4. Proof for Example 4.4

We have shown that the constraint matrix for DP is $\mathbf{D}^{a,(y,a')}$, where the $y$-th column elements of $\mathbf{D}^{a,(y,a')}$ are $\mathbb{I}[a = a'] - \omega_a$ with all other elements set to 0. Plugging it into the expression of $\beta^\lambda$ in (44), we obtain that

$$\beta^\lambda(x) = \sum_{a \in \mathcal{A}} p_a(x)[\mathbf{M}(a, \lambda)]^\top \eta(x, a)$$

$$= \sum_{a \in \mathcal{A}} p_a(x) \left[ \mathbf{I} - \frac{1}{\omega_a} \sum_{k=1}^K \lambda_k \mathbf{D}^{a,k} \right]^\top \eta(x, a)$$

$$= \eta(x) - \sum_{a \in \mathcal{A}} \frac{p_a(x)}{\omega_a} \sum_{y \in [m], a' \in \mathcal{A}} \lambda_{y,a'} \left[ \mathbf{D}^{a,(y,a')} \right]^\top \eta(x, a)$$

$$= \eta(x) - \sum_{a \in \mathcal{A}} \frac{p_a(x)}{\omega_a} \sum_{y \in [m], a' \in \mathcal{A}} \lambda_{y,a'} \mathbf{e}_y (\mathbb{I}[a = a'] - \omega_a) \sum_{j \in [m]} \eta_j(x, a)$$

$$= \eta(x) - \sum_{a \in \mathcal{A}} \frac{p_a(x)}{\omega_a} \sum_{y \in [m], a' \in \mathcal{A}} \lambda_{y,a'} \mathbf{e}_y \mathbb{I}[a = a'] + \sum_{a \in \mathcal{A}} p_a(x) \sum_{y \in [m], a' \in \mathcal{A}} \lambda_{y,a'} \mathbf{e}_y$$

$$= \eta(x) - \sum_{y \in [m]} \sum_{a \in \mathcal{A}} \frac{p_a(x)}{\omega_a} \lambda_{y,a} \mathbf{e}_y + \sum_{y \in [m]} \sum_{a \in \mathcal{A}} \lambda_{y,a} \mathbf{e}_y$$

$$= \eta(x) + \sum_{y \in [m]} \sum_{a \in \mathcal{A}} \lambda_{y,a} \left( 1 - \frac{p_a(x)}{\omega_a} \right) \mathbf{e}_y$$

Now if we only consider the $i$-th element of the vector $\beta^\lambda$, the expression comes to

$$\beta_i^\lambda(x) = \eta_i(x) + \sum_{a \in \mathcal{A}} \lambda_{i,a} \left( 1 - \frac{p_a(x)}{\omega_a} \right).$$

This finishes the proof. □

## C. Proofs and Discussions for Section 5

### C.1. Proof of Theorem 5.1

In this section, we prove that the representation in Eq. (5.1) is calibrated for inner optimization problem. We begin by presenting the following lemma.

**Lemma C.1.** *For any categorical distribution characterized by $\mathbf{p} \in \Delta_m$, the minimizer of the expected risk*

$$\mathbb{E}_{y \sim \mathbf{p}}\left[-\log\left(\mathbf{q}_y\right)\right] = -\sum_{i=1}^{m} \mathbf{p}_i \log\left(\mathbf{q}_i\right)$$

*over all $\mathbf{q} \in \Delta_m$ is unique and achieved at $\mathbf{p} = \mathbf{q}$.*

This lemma is commonly used in the design of multiclass loss functions (Williamson et al., 2016; Menon et al., 2021; Narasimhan & Menon, 2021; Zhou et al., 2024; Zeng et al., 2025a; Yang et al., 2026b).

*Proof of Theorem 5.1.* We aim to prove that for any fixed $x \in \mathcal{X}$, the optimal scoring function $f^* : \mathcal{X} \to \mathbb{R}^m$ that minimizes the expected loss $\mathbb{E}[\ell_{\text{cal}}(Y, f(X), A)]$ over the local data distribution $(X, Y, A)$ recovers the Bayes-optimal fair classifier $h_\beta^*$ given $\lambda$.

To this end, by leveraging the properties of conditional expectation, the cost-sensitive loss is reformulated as a function of the marginal distribution $(X, Y, A)$:

$$\mathbb{E}_{(x,y,a) \sim (X,Y,A)}[\ell_{\text{cal}}(y, f(x), a)] = -\mathbb{E}_{X,Y,A}\left[\sum_{i=1}^{m} \mathbf{M}'_{Y,i}(A, \lambda) \log \frac{\exp([f(X)]_i)}{\sum_{j=1}^{m} \exp([f(X)]_j)}\right]$$

$$= -\mathbb{E}_{X,A}\left[\mathbb{E}_{Y|X,A}\left[\sum_{i=1}^{m} \mathbf{M}'_{Y,i}(A, \lambda) \log \frac{\exp([f(X)]_i)}{\sum_{j=1}^{m} \exp([f(X)]_j)}\right]\right]$$

$$= -\mathbb{E}_{X,A}\left[\sum_{y \in [m]} \mathbb{P}(Y = y \mid X, A) \sum_{i=1}^{m} \mathbf{M}'_{y,i}(A, \lambda) \log \frac{\exp([f(X)]_i)}{\sum_{j=1}^{m} \exp([f(X)]_j)}\right]$$

$$= -\mathbb{E}_{X}\left[\mathbb{E}_{A|X}\left[\sum_{y \in [m]} \eta_y(X, A) \sum_{i=1}^{m} \mathbf{M}'_{y,i}(A, \lambda) \log \frac{\exp([f(X)]_i)}{\sum_{j=1}^{m} \exp([f(X)]_j)}\right]\right]$$

$$= \mathbb{E}_{X}\left[-\sum_{a \in \mathcal{A}} \mathbb{P}(A = a|X) \sum_{i=1}^{m} \left(\left[\mathbf{M}'(a, \lambda)\right]^\top \eta(X, a)\right)_i \log \frac{\exp([f(x)]_i)}{\sum_{j=1}^{m} \exp([f(x)]_j)}\right]$$

Denoting $\mathbf{v}_i(x) := \sum_{a \in \mathcal{A}} \mathbb{P}(A = a|X = x)\left(\left[\mathbf{M}'(a, \lambda)\right]^\top \eta(x, a)\right)_i$, we have

$$\mathbb{E}_{X,Y,A}[\ell_{\text{cal}}(y, f(x), a)] = \mathbb{E}_{X}\left[-c_X \sum_{i=1}^{m} \frac{\mathbf{v}_i(X)}{\sum_{j \in [m]} \mathbf{v}_j(X)} \log \frac{\exp([f(X)]_i)}{\sum_{j=1}^{m} \exp([f(X)]_j)}\right]$$

where $c_x = \sum_{j \in [m]} \mathbf{v}_j(x)$ can be treated as a constant for fixed $x$. According to Lemma C.1, given fixed $x$, an optimal classifier $f^*(x)$ minimizing the cost-sensitive loss point-wise satisfies

$$\frac{\mathbf{v}_i(x)}{\sum_{j \in [m]} \mathbf{v}_j(x, k)} = \frac{\exp([f^*(x)]_i)}{\sum_{j=1}^{m} \exp([f^*(x)]_j)}, \quad \forall i \in [m].$$

It presents that, for all $i \in [m]$, since $\sum_{i \in [m]} \eta_i(x, a) = 1$,

$$
\begin{aligned}
\exp([f^*(x)]_i) = \mathbf{v}_i(x) &= \sum_{a \in \mathcal{A}} \mathbb{P}(A = a | X = x) \left( \left[ \mathbf{M}'(a, \lambda) \right]^\top \eta(x, a) \right)_i \\
&= \sum_{a \in \mathcal{A}} \mathbb{P}(A = a | X = x) \left( \left[ \mathbf{M}(a, \lambda) + \alpha \mathbf{1}_{m \times m} \right]^\top \eta(x, a) \right)_i \\
&= \sum_{a \in \mathcal{A}} \mathbb{P}(A = a | X = x) \left( \left[ \mathbf{M}(a, \lambda) \right]^\top \eta(x, a) + \alpha \mathbf{1}_m \right)_i \\
&= \sum_{a \in \mathcal{A}} \mathbb{P}(A = a | X = x) \left( \left[ \mathbf{M}(a, \lambda) \right]^\top \eta(x, a) \right)_i + \alpha. \\
&= \beta_i^\lambda(x) + \alpha.
\end{aligned}
$$

- For the primal problem in Eq. (1), a straightforward solution to inner optimization is $h^*(x; \beta) := \arg\max_{y \in [m]} (\beta_y^\lambda(x))$. If there exist $i \neq j$, such that $\beta_i^\lambda(x) = \beta_j^\lambda(x) = \max_{y \in [m]} (\beta_y^\lambda(x))$, we determine $h^*(x; \beta) = \mathbf{e}_i$, for $i < j$, otherwise $h^*(x; \beta) = \mathbf{e}_j$. After fixing the order, we have that

$$
\arg\max_{y \in [m]} [f^*(x)]_y = \arg\max_{y \in [m]} \left( \beta^\lambda(x) + \alpha \right)_y = \arg\max_{y \in [m]} \left( \beta^\lambda(x) \right)_y,
$$

  Therefore, the optimal classifier $h^*(x; f) := \arg\min_{y \in [m]} [f^*(x)]_y$ recovers the optimal classifier $h^*(x; \beta)$ given $\lambda$ derived from the the primal problem (1).

- For the entropy-relaxed problem in Eq. (9), the classifier induced by $f^*(x)$ can be constructed as

$$
h_i^*(x; f) := \mathrm{softmax}\left( \exp([f^*(x)]_i)/\tau \right), \quad i \in [m].
$$

Then, it can be obtained that

$$
h_i^*(x; f) = \frac{\exp(\beta_i^\lambda(x)/\tau + \alpha/\tau)}{\sum_{y=1}^m \exp(\beta_y^\lambda(x)/\tau + \alpha/\tau)} = \frac{\exp(\beta_i^\lambda(x)/\tau)}{\sum_{y=1}^m \exp(\beta_y^\lambda(x)/\tau)},
$$

which cover the optimal classifier $h^*(x; \beta)$ in Eq. (10) given $\lambda$ derived from the the entropy-relaxed problem (9).

$\square$

### C.2. Proof of Theorem 5.2

C.2.1. NOTATIONS AND ASSUMPTIONS

With a little abuse of notation, we introduce several symbols that will be used in the proof. Note that the calculation of calibrated loss $\ell_{\mathrm{cal}}$ depends on the dual parameter $\lambda$, given iteration $t \in [T]$, we let

$$
\ell_{\mathrm{cal}}^t(s; \theta) := - \sum_{i=1}^m \left[ \mathbf{M}'(a, \lambda^t) \right]_{y,i} \log \frac{\exp([f(x; \theta)]_i)}{\sum_{j=1}^m \exp([f(x; \theta)]_j)},
$$

where $s := (x, y, a)$.

The dataset $\mathcal{S} = \{x_n, y_n, a_n\}_{n=1}^N$ is sampled from the joint data distribution $\mathcal{D} = (X, Y, A)$. The empirical Lagrangian function is

$$
\begin{aligned}
\widehat{\mathcal{L}}(h, \lambda) &:= \widehat{\mathcal{R}}(h) + \lambda^\top \widehat{\mathscr{D}}(h) - \xi \|\lambda\|_1 \\
&= 1 - \sum_{a \in \mathcal{A}} \left\langle \frac{N_a}{N} \mathbf{I} - \sum_{k \in K} \lambda_k \widehat{\mathbf{D}}^{a,k}, \widehat{\mathbf{C}}^a(h) \right\rangle - \xi \|\lambda\|_1.
\end{aligned}
$$

**Assumption C.2.** ($\nu$-optimal response.) Each loss function $\{\ell^t_{\text{cal}} : t \in [T]\}$ receives an approximate optimal parameter response $\theta^{t+1}$, such that $h^{t+1}(x) = \arg\max_{y \in [m]} f(x; \theta^{t+1})$ achieves the $\nu^t$-approximate optimal solution of empirical inner optimization $\min_{h \in \mathcal{H}} \widehat{\mathcal{L}}(h, \lambda^t)$, i.e.,

$$\widehat{\mathcal{L}}(h^{t+1}, \lambda^t) \le \min_{h \in \mathcal{H}} \widehat{\mathcal{L}}(h, \lambda^t) + \nu^t.$$

Assumption C.2 is also used in previous work (Agarwal et al., 2018; Cotter et al., 2019; Yang et al., 2026a; Lu & Yin, 2025; Cui et al., 2025) for the error analysis of the algorithm's accuracy-fairness trade-off.

Besides, we tackle the equilibrium problem in multi-class classification by designing a more refined, calibrated loss for the inner optimization. Theorem 2 in Tewari & Bartlett (2007) shows that as the sample size increases, the empirical solution minimizing the calibrated loss $\ell^t_{\text{cal}}$ converges to an optimal solution of the parametric inner optimization problem $\min_{\theta \in \theta} \mathcal{L}(h(\cdot; \theta), \lambda^t)$, where $h(\cdot; \theta) := \arg\max_{y \in [m]} f(\cdot; \theta)$, thus making Assumption C.2 mild.

### C.2.2. COMPLETE PROOF OF THEOREM 5.2

For the sake of completeness, we restate Theorem 5.2 below.

**Theorem C.3.** *Under Assumption C.2, denoting $\lambda \in \Lambda := \{\lambda \in \mathbb{R}^K : \|\lambda\|_1 \le B_\Lambda\}$ and $u := \max_{h \in \mathcal{H}} \|\widehat{\mathscr{D}}(h)\|_\infty + \xi$, for $\eta_\lambda = \frac{B_\Lambda}{u\sqrt{2KT}}$, the sequences of classifiers $h^1, \cdots, h^T$ and Lagrange multipliers $\lambda^1, \cdots, \lambda^T$ comprise an approximate mixed Nash equilibrium:*

$$\max_{\lambda \in \Lambda} \widehat{\mathcal{L}}(\bar{h}^T, \lambda) - \min_{h \in \mathcal{H}} \widehat{\mathcal{L}}(h, \bar{\lambda}^T) \le \rho^T := \bar{\nu}^T + uB_\Lambda \sqrt{\frac{2K}{T}}, \tag{19}$$

*where $\bar{h}^T := \sum_{t=1}^{T} h^t/T, \bar{\lambda}^T := \sum_{t=1}^{T} \lambda^t/T, \bar{\nu}^T := \sum_{t=1}^{T} \nu^t/T$.*

*Proof.* Considering that $\partial_\lambda \widehat{\mathcal{L}}(h, \lambda) = \widehat{\mathscr{D}}(h) + \partial\|\lambda\|_1 \xi$, the choices of $\lambda^t$ then depend on sub-gradient descent and lazy projection with the learning rate $\eta_\lambda$. By the setting of the theorem, we have $\|\lambda\|_2^2 \le B_\Lambda^2$ and $\widehat{\mathcal{L}}(h, \lambda)$ is $u\sqrt{K}$-Lipschitz with respect to $\lambda$. We introduce Corollary 3 and Corollary 4 in (Duchi et al., 2010) to bound the regret in our case. By plugging the Euclidean mirror $\frac{1}{2}\|x\|_2^2$ and non-smooth function $r(\lambda) = \xi\|\lambda\|_1$ into these two corollaries, the regret bound indicates that, for any $\lambda \in \Lambda$,

$$\sum_{t=1}^{T} \left( \lambda^\top \widehat{\mathscr{D}}(h^t) - \xi\|\lambda\|_1 \right) \le \sum_{t=1}^{T} \left( (\lambda^t)^\top \widehat{\mathscr{D}}(h^t) - \xi\|\lambda^t\|_1 \right) + \frac{\|\lambda - \lambda^0\|_2^2}{2\eta_\lambda} + \xi\|\lambda^0\|_1 + \frac{1}{2}\eta_\lambda TKu^2. \tag{20}$$

In particular, by setting $\lambda^0 = 0$ and $\eta_\lambda = \frac{B_\Lambda}{u\sqrt{KT}}$, it shows that

$$\sum_{t=1}^{T} \lambda^\top \widehat{\mathscr{D}}(h^t) - \xi\|\lambda\|_1 \le \sum_{t=1}^{T} \left( (\lambda^t)^\top \widehat{\mathscr{D}}(h^t) - \xi\|\lambda^t\|_1 \right) + uB_\Lambda \sqrt{KT} \tag{21}$$

Returning to the Lagrangian function, we can derive the sub-optimality of the problem $\max_{\lambda \in \Lambda} \widehat{\mathcal{L}}(h, \lambda)$. For any $\lambda \in \Lambda$,

$$\widehat{\mathcal{L}}(\bar{h}^T, \lambda) = \frac{1}{T}\sum_{t=1}^{T}(\widehat{\mathcal{R}}(h^t) + \lambda^\top \widehat{\mathscr{D}}(h^t) - \xi\|\lambda\|_1)$$

$$\le \frac{1}{T}\sum_{t=1}^{T}(\widehat{\mathcal{R}}(h^t) + (\lambda^t)^\top \widehat{\mathscr{D}}(h^t) - \xi\|\lambda^t\|_1) + uB_\Lambda \sqrt{\frac{K}{T}}$$

$$= \frac{1}{T}\sum_{t=1}^{T} \widehat{\mathcal{L}}(h^t, \lambda^t) + uB_\Lambda \sqrt{\frac{K}{T}}.$$

The second equation follows the regret bound (21). The result tells us that,

$$\max_{\lambda^* \in \Lambda} \widehat{\mathcal{L}}(\bar{h}^T, \lambda^*) - \frac{1}{T}\sum_{t=1}^{T} \widehat{\mathcal{L}}(h^t, \lambda^t) \le uB_\Lambda \sqrt{\frac{K}{T}}. \tag{22}$$

On the other side, considering the optimization with respect to classifier $h$, under Assumption C.2, it presents that, for any classifier $h \in \mathcal{H}$,

$$\widehat{\mathcal{L}}(h, \bar{\lambda}^T) = \frac{1}{T} \sum_{t=1}^{T} (\widehat{\mathcal{R}}(h) + (\lambda^t)^\top \widehat{\mathscr{D}}(h) - \xi \|\lambda^t\|_1)$$

$$\geq \frac{1}{T} \sum_{t=1}^{T} (\widehat{\mathcal{R}}(h^t) + (\lambda^t)^\top \widehat{\mathscr{D}}(h^t) - \xi \|\lambda^t\|_1 - \nu^t)$$

$$= \frac{1}{T} \sum_{t=1}^{T} \widehat{\mathcal{L}}(h^t, \lambda^t) - \bar{\nu}^T,$$

where $\bar{\nu}^T := \frac{1}{T} \sum_{t=1}^{T} \nu^t$. The second equation follows the Assumption C.2. The result shows that,

$$\frac{1}{T} \sum_{t=1}^{T} \widehat{\mathcal{L}}(h^t, \lambda^t) - \min_{h^* \in \mathcal{H}} \widehat{\mathcal{L}}(h^*, \bar{\lambda}^T) \leq \bar{\nu}^T. \tag{23}$$

Summing the equation (22) and (23), we derive that

$$\max_{\lambda \in \Lambda} \widehat{\mathcal{L}}(\bar{h}^T, \lambda) - \min_{h \in \mathcal{H}} \widehat{\mathcal{L}}(h, \bar{\lambda}^T) \leq \bar{\nu}^T + uB_\Lambda \sqrt{\frac{K}{T}} := \rho^T, \tag{24}$$

which presents the approximate mixed Nash equilibrium of the stochastic saddle-point problem. Hence, it is clear that

$$\widehat{\mathcal{L}}\left(\bar{h}^T, \bar{\lambda}^T\right) \leq \widehat{\mathcal{L}}\left(h, \bar{\lambda}^T\right) + \rho^T, \qquad \forall\, h \in \mathcal{H}, \tag{25}$$

$$\widehat{\mathcal{L}}\left(\bar{h}^T, \bar{\lambda}^T\right) \geq \widehat{\mathcal{L}}\left(\bar{h}^T, \lambda\right) - \rho^T, \qquad \forall\, \lambda \in \Lambda. \tag{26}$$

This completes the proof. □

## C.3. Proof of Theorem 5.3

### C.3.1. DEFINITIONS AND USEFUL LEMMA

The empirical group-specific confusion matrix can be written as

$$\widehat{\mathbf{C}}_{i,j}^a(h) := \mathbb{E}_{\mathcal{S}}[\mathbb{I}(Y = e_i, h(X) = e_j) \mid A = a] = \frac{1}{N_a} \sum_{n=1}^{N} \mathbb{I}(y_n = e_i \wedge a_n = a) h_j(x_n),$$

where $N_a$ denotes the sample size of group $a$, and the conditional expectation is evaluated over the empirical distribution.

Here, we recall the definition of pseudo-shattering and pseudo-dimension.

**Definition C.4.** (Pseudo-Shattering.) Let $\mathcal{H}$ be a set of real valued functions from input space $\mathcal{X}$. We say $C = (x_1, \ldots, x_m)$ is pseudo-shattered by $\mathcal{H}$ if there exists a vector $r = (r_1, \ldots, r_m)$ (called "witness") such that for all $b \in \{\pm 1\}^m = (b_1, \ldots, b_m)$ there exists $h_b \in \mathcal{H}$ such that $\mathrm{sign}\,(h_b(x_i) - r_i) = b$.

**Definition C.5.** (Pseudo-Dimension.) Let $\mathcal{H}$ be a set of real valued functions from input space $\mathcal{X}$. The pseudo-dimension $P \dim(\mathcal{H})$ is the cardinality of the largest set pseudo-shattered by $\mathcal{H}$.

**Lemma C.6.** *Let* $\mathcal{H} : \mathcal{X} \to \Delta_m$, $\mathcal{D}$ *is a distribution over* $\mathcal{X} \times [m]$, *of which* $\{x_i, y_i\}_{i=1}^N$ *are i.i.d. data sampling from* $\mathcal{D}$. *Denoting* $\mathcal{H}_y = \{h_y(\cdot) \in [0, 1] : h \in \mathcal{H}\}$ *and* $\max_{y \in [m]} P\dim(\mathcal{H}_y) \leq d$, *then with probability at least* $1 - \delta$, *for* $\forall i, j \in [m]$,

$$\sup_{h \in \mathcal{H}} \left| \mathbf{C}_{i,j}(h) - \widehat{\mathbf{C}}_{i,j}(h) \right| \leq 2\sqrt{\frac{2d \log(eN/d)}{N}} + \sqrt{\frac{2 \log(4/\delta)}{N}}.$$

*Proof.* Given $h \in \mathcal{H}$, letting $\ell_{i,j}(h(x), y) = \mathbb{I}(y = e_i)h_j(x)$, it can be shown that

$$\mathbb{E}_{\mathcal{D}}[\ell_{i,j}(h(x), y)] = \mathbb{E}_X[\mathbb{E}_{Y|X}[\mathbb{I}(y = e_i)h_j(x)]]$$

$$= \mathbb{E}_X\left[h_j(x) \sum_{y=1}^{m} \eta_y(x)\mathbb{I}(y = e_i)\right]$$

$$= \mathbb{E}_X[\eta_i(x)h_j(x)]$$

$$= \mathbf{C}_{i,j}(h).$$

Hence, the loss $\ell_{i,j}$ is an unbiased estimator for $\mathbf{C}_{i,j}(h)$. Moreover, we consider the function class $\mathcal{G} = \{g(x,y) = \ell_{i,j}(h(x), y) \in [0,1] : h \in \mathcal{H}\}$. Let $(x_1, y_1), \cdots, (x_{d'}, y_{d'}) \in \mathcal{X} \times [m]$ be distinct points. Suppose $\mathcal{G}$ pseudo-shatters this set, then $\exists r_1, \ldots, r_{d'}$ such that $\forall b_1, \ldots, b_{d'} \in \{0,1\}$ and for all $i \in [d']$, $\exists h \in \mathcal{H}$, $\mathbb{I}[\mathbb{I}[y = \mathbf{e}_i]h_j(x_i) \geq r_i] = b_i$. Then, if $y = \mathbf{e}_i$, we have

$$\exists h \in \mathcal{H}, \quad \begin{cases} h(x_i) \geq r_i & \text{if } b_i = 1 \\ h(x_i) < r_i & \text{if } b_i = 0 \end{cases}. \tag{27}$$

If $y \neq \mathbf{e}_i$, $g(x,y) = 0$, we can deduce that there is not function $h_j \in \mathcal{H}_j$ such that the above expression holds. It indicates that $y_1 = y_2 = \cdots = y_{d'} = \mathbf{e}_i$. Consequently, Eq. (27) holds for all $i \in [d']$, which tell us that $\mathcal{H}_j$ pseudo-shatters the set $x_1, \cdots, x_{d'}$. Hence, the pseudo-dimension of $\mathcal{G}$ is less than that of $\mathcal{H}_j$ by the definition of Pseudo-Shattering (Def. C.4), i.e., $P\dim(\mathcal{G}) \leq P\dim(\mathcal{H}_j) \leq d$.

Therefore, by the generalization error derived from Pseudo-Dimension (e.g., Theorem 11.8 of Mohri et al. (2018)), with probability at least $1 - \delta$,

$$\sup_{h \in \mathcal{H}} |\mathbf{C}_{i,j}(h) - \widehat{\mathbf{C}}_{i,j}(h)| \leq 2\sqrt{\frac{2d \log(eN/d)}{N}} + \sqrt{\frac{2 \log(4/\delta)}{N}}.$$

This completes the proof. $\qquad\qquad\qquad\qquad\qquad\qquad\qquad\qquad\qquad\qquad\qquad\qquad\qquad\qquad\qquad\qquad\qquad\square$

### C.3.2. COMPLETE PROOF OF THEOREM 5.3

In this subsection, we state the complete formulation of Theorem 5.3 and provide the proofs.

**Theorem C.7.** *If classifier $\bar{h}^T$ with dual parameters $\bar{\lambda}^T$ comprise a $\rho^T$-saddle point of the empirical Lagrangian $\widehat{\mathcal{L}}(h, \lambda)$, and an optimal solution $h^* \in \mathcal{H}$ satisfies fairness constraints, assuming that $\mathcal{H}_y = \{h_y(\cdot) \in [0,1] : h \in \mathcal{H}\}$ and $\max_{y \in [m]} P\dim(\mathcal{H}_y) \leq d$, denoting $u_k := \max_{a \in \mathcal{A}} \|\mathbf{D}^{a,k}\|_1$ and $\Omega_k^N := \max_{a \in \mathcal{A}} \|\mathbf{D}^{a,k} - \widehat{\mathbf{D}}^{a,k}\|_\infty$, $k \in K$, then with probability at least $1 - \delta$,*

$$\mathcal{R}(\bar{h}^T) \leq \mathcal{R}(h^*) + 2\rho^T + 2\sqrt{\frac{8 \log(eN/d) + 2 \log(4m^2/\delta)}{N}},$$

$$|\mathscr{D}_k(\bar{h}^T)| \leq \xi + \frac{1 + 2\rho^T}{B_\Lambda} + 4u_k \sum_{a \in \mathcal{A}} \left(\sqrt{\frac{8d \log(eN_a/d) + 2 \log(4m^2|\mathcal{A}|/\delta)}{N_a}}\right) + 2|\mathcal{A}|\Omega_k^N..$$

*Proof.* We analyze the generalization error of the accuracy-fairness trade-off obtained by Algorithm 1. Firstly, we derive the corresponding generalization errors for risk $\mathcal{R}$ and fairness constraints $\mathscr{D}_k, k \in K$. For fairness constraints, given $k \in K$ and $h \in \mathcal{H}$, by Hölder's inequality,

$$\left|\widehat{\mathscr{D}}_k(h) - \mathscr{D}_k(h)\right| = \left|\sum_{a \in \mathcal{A}} \langle \mathbf{D}^{a,k}, \mathbf{C}^a(h) \rangle - \langle \widehat{\mathbf{D}}^{a,k}, \widehat{\mathbf{C}}^a(h) \rangle\right|$$

$$= \left|\sum_{a \in \mathcal{A}} \langle \mathbf{D}^{a,k}, \mathbf{C}^a(h) - \widehat{\mathbf{C}}^a(h) \rangle + \langle \mathbf{D}^{a,k} - \widehat{\mathbf{D}}^{a,k}, \widehat{\mathbf{C}}^a(h) \rangle\right|$$

$$\leq \sum_{a \in \mathcal{A}} \|\mathbf{D}^{a,k}\|_1 \|\mathbf{C}^a(h) - \widehat{\mathbf{C}}^a(h)\|_\infty + \sum_{a \in \mathcal{A}} \|\mathbf{D}^{a,k} - \widehat{\mathbf{D}}^{a,k}\|_\infty \|\widehat{\mathbf{C}}^a(h)\|_1$$

$$\leq \sum_{a \in \mathcal{A}} \|\mathbf{D}^{a,k}\|_1 \|\mathbf{C}^a(h) - \widehat{\mathbf{C}}^a(h)\|_\infty + |\mathcal{A}|\left(\max_{a \in \mathcal{A}} \|\mathbf{D}^{a,k} - \widehat{\mathbf{D}}^{a,k}\|_\infty\right)$$

By introducing $\ell_{i,j}^{a'}(x, y, a; h) = \mathbb{I}(y = e_i \wedge a = a') h_j(x_i)$ and taking a union bound with respect to $i, j \in [m]$ in Lemma C.6, we obtain that, for $a \in \mathcal{A}$, with probability at least $1 - \delta$,

$$\sup_{h \in \mathcal{H}} \|\mathbf{C}^a(h) - \widehat{\mathbf{C}}^a(h)\|_\infty \leq 2\sqrt{\frac{2d \log(eN_a/d)}{N_a}} + \sqrt{\frac{2 \log(4m^2/\delta)}{N_a}}.$$

We have denoted $u_k := \max_{a \in \mathcal{A}} \|\mathbf{D}^{a,k}\|_1$ and $\Omega_k^N := \max_{a \in \mathcal{A}} \|\mathbf{D}^{a,k} - \widehat{\mathbf{D}}^{a,k}\|_\infty$. Here, $\Omega_k^N \to 0$ as $N \to \infty$ by the Law of Large Numbers, because $\mathbf{D}^{a,k}$ is composed of the statistics of samples $\mathcal{S}$. Taking a union bound with respect to $a \in \mathcal{A}$, the generalization bound presents that, with probability at least $1 - \delta$,

$$\left|\widehat{\mathscr{D}}_k(h) - \mathscr{D}_k(h)\right| \leq u_k \sum_{a \in \mathcal{A}} \left(2\sqrt{\frac{2d \log(eN_a/d)}{N_a}} + \sqrt{\frac{2 \log(4m^2|\mathcal{A}|/\delta)}{N_a}}\right) + |\mathcal{A}|\Omega_k^N.$$

For classification risk, given $h \in \mathcal{H}$,

$$\left|\widehat{\mathcal{R}}(h) - \mathcal{R}(h)\right| = \left|\langle \mathbf{I}, \widehat{\mathbf{C}}(h) - \mathbf{C}(h)\rangle\right| \leq \|\widehat{\mathbf{C}}(h) - \mathbf{C}(h)\|_\infty.$$

By taking a union bound in Lemma C.6, we obtain that, with probability at least $1 - \delta$,

$$\left|\widehat{\mathcal{R}}(h) - \mathcal{R}(h)\right| \leq \sup_{h \in \mathcal{H}} \|\widehat{\mathbf{C}}(h) - \mathbf{C}(h)\|_\infty \leq 2\sqrt{\frac{2d \log(eN/d)}{N}} + \sqrt{\frac{2 \log(4m^2/\delta)}{N}}.$$

Next, we turn to the generalization bound of the fairness constraints. The following proof largely follows the proof of Theorem 4 in Agarwal et al. (2018). With a little abuse of notation for clarity, given that $(\bar{h}^T, \bar{\lambda}^T)$ is an approximate saddle point of the empirical Lagrangian with relaxed bounds,

$$\widehat{\mathcal{L}}(h, \lambda) := \widehat{\mathcal{R}}(h) + \lambda^\top \widehat{\mathscr{D}}_k(h) - \sum_{k=1}^K \widehat{\xi}_k |\lambda_k|,$$

$$\widehat{\xi}_k := \xi + \epsilon_k, \quad \epsilon_k \geq u_k \sum_{a \in \mathcal{A}} \left(2\sqrt{\frac{2d \log(eN_a/d)}{N_a}} + \sqrt{\frac{2 \log(4m^2|\mathcal{A}|/\delta)}{N_a}}\right) + |\mathcal{A}|\Omega_k^N.$$

Then, since $h^*$ satisfies fairness constraints $|\mathscr{D}_k(h^*)| \leq \xi$, it shows that, with probability at least $1 - K\delta$,

$$\begin{aligned}
|\widehat{\mathscr{D}}_k(h^*)| &= |\widehat{\mathscr{D}}_k(h^*) - \mathscr{D}_k(h^*) + \mathscr{D}_k(h^*)| \\
&\leq |\widehat{\mathscr{D}}_k(h^*) - \mathscr{D}_k(h^*)| + |\mathscr{D}_k(h^*)| \\
&\leq \epsilon_k + \xi = \widehat{\xi}_k.
\end{aligned}$$

The items in empirical Lagrangian introduced by fairness constraints turn to

$$\begin{aligned}
k \in K, \quad \lambda_k \widehat{\mathscr{D}}_k(h^*) - \widehat{\xi}_k |\lambda_k| &= \lambda_k (\widehat{\mathscr{D}}_k(h^*) - \widehat{\xi}_k) \leq \lambda_k (|\widehat{\mathscr{D}}_k(h^*)| - \widehat{\xi}_k) \leq 0, \quad \text{if } \lambda_k \geq 0; \\
\lambda_k \widehat{\mathscr{D}}_k(h^*) - \widehat{\xi}_k |\lambda_k| &= \lambda_k (\widehat{\mathscr{D}}_k(h^*) + \widehat{\xi}_k) \leq \lambda_k (\widehat{\xi}_k - |\widehat{\mathscr{D}}_k(h^*)|) \leq 0, \quad \text{if } \lambda_k < 0.
\end{aligned}$$

Therefore, we obtain that

$$\widehat{\mathcal{L}}(h^*, \bar{\lambda}^T) = \widehat{\mathcal{R}}(h^*) + \sum_{k=1}^K \left(\bar{\lambda}_k^T \widehat{\mathscr{D}}_k(h) - \widehat{\xi}_k |\bar{\lambda}_k^T|\right) \leq \widehat{\mathcal{R}}(h^*).$$

The approximate saddle point's property 25 and 26 implies that

$$\begin{aligned}
\widehat{\mathcal{R}}(\bar{h}^T) - \widehat{\mathcal{R}}(h^*) &\leq \widehat{\mathcal{L}}(\bar{h}^T, \mathbf{0}_K) - \widehat{\mathcal{L}}(h^*, \bar{\lambda}^T) \\
&\leq \widehat{\mathcal{L}}(\bar{h}^T, \bar{\lambda}^T) + \rho^T - \widehat{\mathcal{L}}(h^*, \bar{\lambda}^T) \\
&\leq \widehat{\mathcal{L}}(\bar{h}^T, \bar{\lambda}^T) + \rho^T - \widehat{\mathcal{L}}(\bar{h}^T, \bar{\lambda}^T) + \rho^T \\
&= 2\rho^T
\end{aligned}$$

For the risk metric, it presents that

$$
\begin{aligned}
\mathcal{R}(\bar{h}^T) - \mathcal{R}(h^*) &= \mathcal{R}(\bar{h}^T) - \widehat{\mathcal{R}}(\bar{h}^T) + \widehat{\mathcal{R}}(\bar{h}^T) - \widehat{\mathcal{R}}(h^*) + \widehat{\mathcal{R}}(h^*) - \mathcal{R}(h^*) \\
&\leq \widehat{\mathcal{R}}(\bar{h}^T) - \widehat{\mathcal{R}}(h^*) + 4\sqrt{\frac{2d\log(eN/d)}{N}} + 2\sqrt{\frac{2\log(4m^2/\delta)}{N}} \\
&\leq 2\rho^T + 4\sqrt{\frac{2d\log(eN/d)}{N}} + 2\sqrt{\frac{2\log(4m^2/\delta)}{N}} \\
&\leq 2\rho^T + 4\sqrt{\frac{8d\log(eN/d) + 2\log(4m^2/\delta)}{N}}.
\end{aligned}
$$

When it comes to the fairness constraints, denoting $k^* := \arg\max_{k \in K} |\widehat{\mathscr{D}}_k(\bar{h}^T)|$ and $\widehat{B}_\Lambda := \text{sign}(\widehat{\mathscr{D}}_k(\bar{h}^T))B_\Lambda$, then we have

$$
B_\Lambda(|\widehat{\mathscr{D}}_{k^*}(\bar{h}^T)| - \widehat{\xi}_{k^*}) = \widehat{\mathcal{L}}(\bar{h}, \widehat{B}_\Lambda e_{k^*}) - \widehat{\mathcal{R}}(\bar{h}^T) \leq \widehat{\mathcal{L}}(\bar{h}^T, \overline{\lambda}^T) - \widehat{\mathcal{R}}(\bar{h}^T) + \rho^T, \tag{28}
$$

where $e_{k^*}$ is the $K$-dimensional basis vector with 1 at the $k^*$-th element and 0 otherwise. Let $h'$ satisfy the fairness constraints. With (25), we obtain

$$
\widehat{\mathcal{L}}(\bar{h}^T, \overline{\lambda}^T) - \widehat{\mathcal{R}}(\bar{h}^T) \leq \widehat{\mathcal{L}}(h', \overline{\lambda}^T) - \widehat{\mathcal{R}}(\bar{h}^T) + \rho^T \leq \widehat{\mathcal{R}}(h') - \widehat{\mathcal{R}}(\bar{h}^T) + \rho^T. \tag{29}
$$

Combining (28) and (29), it shows that

$$
\max_k \left( |\widehat{\mathscr{D}}_k(\bar{h}^T)| - \widehat{\xi}_k \right) \leq \frac{\widehat{\mathcal{R}}(h') - \widehat{\mathcal{R}}(\bar{h}^T) + 2\rho^T}{B_\Lambda} \leq \frac{1 + 2\rho^T}{B_\Lambda}. \tag{30}
$$

Therefore, the result shows that, for any $k \in K$,

$$
\begin{aligned}
|\mathscr{D}_k(\bar{h}^T)| &= |\mathscr{D}_k(\bar{h}^T) - \widehat{\mathscr{D}}_k(\bar{h}^T) + \widehat{\mathscr{D}}_k(\bar{h}^T)| \leq |\mathscr{D}_k(\bar{h}^T) - \widehat{\mathscr{D}}_k(\bar{h}^T)| + |\widehat{\mathscr{D}}_k(\bar{h}^T)| \\
&\leq \xi + \frac{1 + 2\rho^T}{B_\Lambda} + 2u_k \sum_{a \in \mathcal{A}} \left( 2\sqrt{\frac{2d\log(eN_a/d)}{N_a}} + \sqrt{\frac{2\log(4m^2|\mathcal{A}|/\delta)}{N_a}} \right) + 2|\mathcal{A}|\Omega_k^N \\
&\leq \xi + \frac{1 + 2\rho^T}{B_\Lambda} + 4u_k \sum_{a \in \mathcal{A}} \left( \sqrt{\frac{8d\log(eN_a/d) + 2\log(4m^2|\mathcal{A}|/\delta)}{N_a}} \right) + 2|\mathcal{A}|\Omega_k^N,
\end{aligned}
$$

which completes the proof. $\square$

### C.4. Proof of Theorem 5.6

To derive the sample complexity for the post-processing algorithm, we proceed by analyzing the complexity of the hypothesis class $h^\lambda, \lambda \in \Lambda$. To begin, we recall the definition of Rademacher complexity and introduce relevant uniform convergence bound.

**Definition C.8.** (Rademacher Complexity.) Let $\mathcal{G}$ be a family of functions mapping from $z$ to $[a, b]$ and $S = (z_1, \ldots, z_m)$ a fixed sample of size $m$ with elements in $z$. Then, the empirical Rademacher complexity of $\mathcal{G}$ with respect to the sample $S$ is defined as:

$$
\widehat{\mathfrak{R}}_S(\mathcal{G}) = \mathbb{E}_{\boldsymbol{\sigma}} \left[ \sup_{g \in \mathcal{G}} \frac{1}{m} \sum_{i=1}^m \sigma_i g(z_i) \right],
$$

where $\boldsymbol{\sigma} = (\sigma_1, \ldots, \sigma_m)^\top$, with $\sigma_i$ s independent uniform random variables taking values in $\{-1, +1\}$. The random variables $\sigma_i$ are called Rademacher variables.

**Lemma C.9.** *Let $\mathcal{G}$ be a family of functions mapping from $z$ to $[0, 1]$. Then, for any $\delta > 0$, with probability at least $1 - \delta$ over the draw of an i.i.d. sample $S$ of size $m$, the following inequality holds for all $g \in \mathcal{G}$ :*

$$
\left| \mathbb{E}[g(z)] - \frac{1}{m} \sum_{i=1}^m g(z_i) \right| \leq 2\widehat{\mathfrak{R}}_S(\mathcal{G}) + 3\sqrt{\frac{\log(4/\delta)}{2m}}.
$$

Lemma C.9 follows directly from Theorem 3.3 in Mohri et al. (2018). Here, we take a union bound so that the generalization bound holds for the absolute-value expression on the left-hand side.

**Lemma C.10.** *Let $\mathcal{Z}$ be any set, $(z_1, \ldots, z_N) \in \mathcal{Z}^N$, let $\mathcal{G}$ be a class of functions $g : \mathcal{Z} \to \ell_2(\mathbb{R}^m)$ and let $f_i : \ell_2(\mathbb{R}^m) \to \mathbb{R}$ have Lipschitz norm $L$. Then*

$$\mathbb{E}_{\boldsymbol{\sigma}}\left[\sup_{g \in \mathcal{G}} \sum_{n=1}^{N} \sigma_n f_n\left(g\left(z_n\right)\right)\right] \leq \sqrt{2} L \mathbb{E}_{\boldsymbol{\sigma}}\left[\sup_{g \in \mathcal{G}} \sum_{n=1}^{N} \sum_{i=1}^{m} \sigma_{ni} g_i\left(z_n\right)\right],$$

*where $\sigma_{ni}$ is an independent doubly indexed Rademacher sequence and $g_i\left(z_n\right)$ is the $i$-th component of $g\left(z_n\right)$.*

The structural generalization risk in Lemma C.10 is the main result of previous work (Lei et al., 2015; Maurer, 2016).

**Lemma C.11.** *Let $A$ be a discrete random variable supported on a finite set $\mathcal{A}$ and $\mathbb{P}(A = a) = p_a > 0$. Draw i.i.d. samples $\{z_i\}_{i=1}^{N}$, and define the empirical frequencies $\widehat{p}_a := \frac{1}{N} \sum_{i=1}^{N} \mathbb{I}(z_i = a)$, with $p_{\min} := \min_{a \in \mathcal{A}} p_a$.*

*1. For any $\delta \in (0, 1)$,*

$$\mathbb{P}\left(\max_{a \in \mathcal{A}} |\widehat{p}_a - p_a| \leq \sqrt{\frac{1}{2N} \log \frac{2|\mathcal{A}|}{\delta}}\right) \geq 1 - \delta \tag{31}$$

*2. For any $\varepsilon \in (0, 1)$ and any $\delta \in (0, 1)$, if $N \geq \frac{2}{\varepsilon^2 p_{\min}} \log \frac{|\mathcal{A}|}{\delta}$, it holds that*

$$\mathbb{P}\left(\min_{a \in \mathcal{A}} \widehat{p}_a \geq (1 - \varepsilon)p_{\min}\right) \geq 1 - \delta. \tag{32}$$

*In particular, taking $\varepsilon = \frac{1}{2}$, if $N \geq \frac{8}{p_{\min}} \log \frac{|\mathcal{A}|}{\delta}$, then*

$$\mathbb{P}\left(\min_{a \in \mathcal{A}} \widehat{p}_a \geq \frac{p_{\min}}{2}\right) \geq 1 - \delta. \tag{33}$$

*Proof of Lemma C.11.* Fix any $a \in \mathcal{A}$, define the count random variable $N_a := \sum_{i=1}^{N} \mathbb{I}(z_i = a)$. Since $z_i$ are i.i.d. and $\mathbb{P}(z_i = a) = p_a$, we have $N_a \sim \text{Binomial}(N, p_a)$ and $\widehat{p}_a = N_a/N$. The Hoeffding's inequality (e.g. Theorem D.2 in Mohri et al. (2018)) implies that

$$\mathbb{P}\left(|\widehat{p}_a - p_a| \geq t\right) \leq 2 \exp(-2Nt^2).$$

Applying the union bound over $a \in \mathcal{A}$, we obtain

$$\mathbb{P}\left(\exists a : |\widehat{p}_a - p_a| \geq \epsilon\right) \leq \sum_{a \in \mathcal{A}} 2 \exp(-2N\epsilon^2) = 2|\mathcal{A}| \exp(-2N\epsilon^2).$$

With $\epsilon^2 = \frac{1}{2N} \log \frac{4K}{\delta}$, we obtain

$$\mathbb{P}\left(\exists a : |\widehat{p}_a - p_a|\right) \leq 2|\mathcal{A}| \exp\left(-\log \frac{2|\mathcal{A}|}{\delta}\right) = \delta,$$

and hence we get Eq. (31)

$$\mathbb{P}\left(\max_{a \in \mathcal{A}} |\widehat{p}_a - p_a| \leq \sqrt{\frac{1}{2N} \log \frac{2|\mathcal{A}|}{\delta}}\right) \geq 1 - \delta.$$

By the standard multiplicative Chernoff bound (e.g. Theorem D.4 in Mohri et al. (2018)), for any $\varepsilon \in (0, 1)$,

$$\mathbb{P}\left(\widehat{p}_a \leq (1 - \varepsilon)p_a\right) \leq \exp\left(-\frac{\varepsilon^2 N p_a}{2}\right).$$

Consider the event $\mathcal{E} := \{\min_{a \in \mathcal{A}} \widehat{p}_a < (1 - \varepsilon)p_{\min}\}$. 5 If $\min_a \widehat{p}_a < (1 - \varepsilon)p_{\min}$, then there exists some $a \in \mathcal{A}$ such that $\widehat{p}_a < (1 - \varepsilon)p_{\min}$. Since $p_{\min} \le p_a$ for every $a$, we have $(1 - \varepsilon)p_{\min} \le (1 - \varepsilon)p_a$, hence $\{\widehat{p}_a < (1 - \varepsilon)p_{\min}\} \subseteq \{\widehat{p}_a \le (1 - \varepsilon)p_a\}$. Therefore, it presents that

$$\mathcal{E} \subseteq \bigcup_{a \in \mathcal{A}} \{\widehat{p}_a \le (1 - \varepsilon)p_a\}.$$

Applying the union bound and using the Chernoff bound, we obtain that

$$\mathbb{P}(\mathcal{E}) \le \sum_{a \in \mathcal{A}} \mathbb{P}\left(\widehat{p}_a \le (1 - \varepsilon)p_a\right) \le \sum_{a \in \mathcal{A}} \exp\left(-\frac{\varepsilon^2 N p_a}{2}\right) \le K \exp\left(-\frac{\varepsilon^2 N p_{\min}}{2}\right).$$

The last inequality uses $p_a \ge p_{\min}$ for all $a$.

Therefore, if $K \exp\left(-\frac{\varepsilon^2 N p_{\min}}{2}\right) \le \delta$, then $\mathbb{P}(\mathcal{E}) \le \delta$. Rearranging the inequality gives

$$-\frac{\varepsilon^2 N p_{\min}}{2} \le \log\frac{\delta}{K} \quad \Longleftrightarrow \quad N \ge \frac{2}{\varepsilon^2 p_{\min}} \log\frac{K}{\delta}.$$

Under this condition, $\mathbb{P}(\mathcal{E}) \le \delta$, i.e., $\mathbb{P}\left(\min_{a \in \mathcal{A}} \widehat{p}_a \ge (1 - \varepsilon)p_{\min}\right) \ge 1 - \delta$, which completes the proof. $\qquad \square$

### C.4.1. COMPLETE PROOF OF THEOREM 5.6

In this subsection, we state the complete formulation of Theorem 5.6 and provide its proof.

Denoting the minimal group probability as $\omega_{\min} := \min_a \omega_a$, the frequency vector $r = (r_1, \cdots, r_K)$, where $r_k := \max_{a \in \mathcal{A}} \|\mathbf{D}^{a,k}\|_\infty / \omega_{\min}$, $k \in K$, with the corresponding estimator $\widehat{r} = (\widehat{r}_1, \cdots, \widehat{r}_K)$ $\widehat{r}_k := \max_{a \in \mathcal{A}} \|\widehat{\mathbf{D}}^{a,k}\|_\infty / \widehat{\omega}_{\min}$, and $\Omega_k^N := \max_{a \in \mathcal{A}} \|\mathbf{D}^{a,k} - \widehat{\mathbf{D}}^{a,k}\|_\infty$, $k \in K$. Since the estimator $\widehat{\mathbf{D}}^{a,k}$ is composed by the frequencies as shown in Table 1, referring to Lemma C.11, we make the following mild assumption.

**Assumption C.12.** There exists $N_k = N_k(\{\mathbf{D}^{a,k}\}_{a \in \mathcal{A}, k \in K}, |\mathcal{A}|, \delta)$, such that for any $\delta \in (0, 1)$, if $N \ge N_k$, it holds that,

$$\mathbb{P}\left(\max_{a \in \mathcal{A}} \|\widehat{\mathbf{D}}^{a,k}\|_\infty \le 2\max_{a \in \mathcal{A}} \|\mathbf{D}^{a,k}\|_\infty\right) \le 1 - \delta.$$

The complete formulation of Theorem 5.6 is presented as follows.

**Theorem C.13.** *Under assumption C.12, denoting an optimal solution of the empirical objective $\max_{\lambda \in \Lambda} \widehat{H}(\lambda)$ as $\widehat{\lambda}^*$, if $N \ge \max\{\frac{8}{\omega_{\min}} \log \frac{16K|\mathcal{A}|}{\delta}, \max_{k \in [K]} N_k(\{\mathbf{D}^{a,k}\}_{a \in \mathcal{A}, k \in K}, |\mathcal{A}|, \delta/8K)\}$, the empirically optimal fair classifier $\widehat{h}^{\widehat{\lambda}^*}$ determined by $\widehat{\lambda}^*$ satisfies that:*

1. *Regrading fairness constraints, for any $\delta \in (0, 1)$, with probability at least $1 - \delta$,*

$$|\mathscr{D}_k(\widehat{h}^{\widehat{\lambda}^*})| \le \xi + r_k \left(|\mathcal{A}| \|\eta(x) - \widehat{\eta}(x)\|_1 + \sum_{a \in \mathcal{A}} \|\mathbf{q}_a - \widehat{\mathbf{q}}_a\|_1\right) + \frac{2|\mathcal{A}|}{\omega_{\min}} \Omega_k^N$$
$$+ \frac{16 r_k B_\Lambda}{\tau} \sqrt{\frac{m\|r\|_2^2}{N}} + \sqrt{\frac{(9 + 4r_k^2/\omega_{\min}^2) \log(8|\mathcal{A}|/\delta)}{N}}.$$

2. *Regrading predictive risk, the risk can be decomposed to (1) error introduced by auxiliary model, (2) error from statistics estimator, and (3) generalization error, denoting as*

$$\epsilon_1 := |\mathcal{A}| \|\eta - \widehat{\eta}\|_1 + \sum_{a \in \mathcal{A}} \|\mathbf{q}_a - \widehat{\mathbf{q}}_a\|_1, \quad \epsilon_2 := \sum_{k=1}^{K} \Omega_k^N, \quad \epsilon_3 := \sqrt{\frac{\log(16K|\mathcal{A}|/\delta)}{N}}, \tag{34}$$

$\exists c_1, c_2$ *then there exist hyperparameter $\tau$ and constant $c_1, c_2, c_3, c_4$, such that for any $\delta \in (0, 1)$, with probability at least $1 - \delta$,*

$$\mathcal{R}(\widehat{h}^{\widehat{\lambda}^*}) \le \mathcal{R}(h^{\lambda^*}) + c_1 \epsilon_1^{\frac{1}{2}} + B_\Lambda \|r\|_1 \epsilon_1 + c_2 |\mathcal{A}| \epsilon_2^{\frac{1}{2}} + \frac{2B_\Lambda |\mathcal{A}|}{\omega_{\min}} \epsilon_2 + c_2 \sqrt{\|r\|_1} \epsilon_3^{\frac{1}{2}} + c_3 \epsilon_3 + c_4 \left(\frac{m}{N}\right)^{\frac{1}{4}},$$

*where* $c_1 = (1 + B_\Lambda m \|r\|_1) \sqrt{2 \log m}$, $c_2 = 2 \sqrt{\frac{m \log m B_\Lambda (1 + B_\Lambda \|r\|_\infty)}{\omega_{\min}}}$, $c_3 = B_\Lambda \sqrt{9 K^2 + 4 \|r\|_2^2 / \omega_{\min}^2}$, $c_4 = 8 \|r\|_1 B_\Lambda \sqrt{\log m}$.

*Proof.* We begin with analyzing the fairness risk of post-processing method, then further considering the predictive accuracy. The proof begins by analyzing the optimality of the empirical problem, then studies the generalization bound via Rademacher complexity, and finally considers the plug-in estimation error. Subsequently, we turn to the risk of predictive accuracy, deducing the generalization bound using the property of saddle-point.

**(Optimality.)** Let $\widehat{\lambda}^* := \arg\min_{\lambda \in \Lambda} \widehat{H}(\lambda)$. Considering the subgradient of the empirical dual function w.r.t. $\lambda_k$, by the additivity of subgradient in the set sense (Def B.1),

$$
\begin{aligned}
\frac{\partial}{\partial \lambda_k} \widehat{H}(\lambda) &= \frac{\tau}{N} \sum_{n=1}^{N} \frac{\partial}{\partial \lambda_k} \log \left( \sum_{j=1}^{m} \exp \left( \widehat{\beta}_j^\lambda(x_n) / \tau \right) \right) + \xi \partial |\lambda_k| \\
&= \frac{\tau}{N} \sum_{n=1}^{N} \frac{\sum_{i=1}^{m} \frac{\partial}{\partial \lambda_k} \exp \left( \widehat{\beta}_i^\lambda(x_n) / \tau \right)}{\sum_{j=1}^{m} \exp \left( \widehat{\beta}_j^\lambda(x_n) / \tau \right)} + \xi \partial |\lambda_k| \\
&= \frac{\tau}{N} \sum_{n=1}^{N} \frac{1}{\tau} \sum_{i=1}^{m} \frac{\exp \left( \widehat{\beta}_i^\lambda(x_n) / \tau \right)}{\sum_{j=1}^{m} \exp \left( \widehat{\beta}_j^\lambda(x_n) / \tau \right)} \cdot \frac{\partial}{\partial \lambda_k} \widehat{\beta}_i^\lambda(x_n) + \xi \partial |\lambda_k| \\
&= -\frac{1}{N} \sum_{n=1}^{N} \sum_{i=1}^{m} \widehat{h}_i^\lambda(x_n) \left( \sum_{a \in \mathcal{A}} \frac{1}{\widehat{\omega}_a} \left[ \mathrm{Diag}(\widehat{\mathbf{q}}_a(x_n)) \widehat{\mathbf{D}}^{a,k} \right]^\top \widehat{\eta}(x_n) \right)_i + \xi \partial |\lambda_k| \\
&= -\frac{1}{N} \sum_{n=1}^{N} [\widehat{\eta}(x_n)]^\top \left[ \sum_{a \in \mathcal{A}} \frac{1}{\widehat{\omega}_a} \mathrm{Diag}(\widehat{\mathbf{q}}_a(x_n)) \widehat{\mathbf{D}}^{a,k} \right] \widehat{h}^\lambda(x_n) + \xi \partial |\lambda_k| \\
&:= -\widehat{\mathscr{D}}_k(\widehat{h}^\lambda) + \xi \partial |\lambda_k|.
\end{aligned}
$$

Therefore, denoting by $\widehat{\mathscr{D}}_{1:K} := \{\widehat{\mathscr{D}}_k\}_{k \in K}$ (or simply $\widehat{\mathscr{D}}$), we can deduce the sub-differential of $\widehat{H}(\lambda)$,

$$
\frac{\partial}{\partial \lambda} \widehat{H}(\lambda) = -\widehat{\mathscr{D}}_{1:K}(\widehat{h}^\lambda) + \xi \partial \|\lambda\|_1.
$$

By Lemma B.5, the optimality condition for the dual problem implies that $\mathbf{0}_K \in \partial_\lambda \widehat{H}(\lambda)$. Given optimal $\widehat{\lambda}^*$ and corresponding $\widehat{h}^{\widehat{\lambda}^*}$, if $\widehat{\lambda}_k^* > 0$, the $k$-th sub-differential reduces to $-\widehat{\mathscr{D}}_k(\widehat{h}^{\widehat{\lambda}^*}) + \xi$, which implies $\widehat{\mathscr{D}}_k(\widehat{h}^{\widehat{\lambda}^*}) = \xi$. Similarly, if $\widehat{\lambda}_k^* < 0$, the $k$-th sub-differential reduces to $-\widehat{\mathscr{D}}_k(\widehat{h}^{\widehat{\lambda}^*}) - \xi$, which implies $\widehat{\mathscr{D}}_k(\widehat{h}^{\widehat{\lambda}^*}) = -\xi$. Besides, if $\widehat{\lambda}_k^* = 0$, it follows that $0 \in -\widehat{\mathscr{D}}_k(\widehat{h}^{\widehat{\lambda}^*}) + [-\xi, \xi]$, namely $\widehat{\mathscr{D}}_k(\widehat{h}^{\widehat{\lambda}^*}) \in [-\xi, \xi]$. Overall, we have shown that for all $k \in K$,

$$
|\widehat{\mathscr{D}}_k(\widehat{h}^{\widehat{\lambda}^*})| \leq \xi \tag{35}
$$

**(Generalization Bound.)** To bridge the empirical estimator with the original formulation of fairness constraints in post-processing, we introduce the following functional to facilitate the transition,

$$
\widetilde{\mathscr{D}}_k(h) := \mathbb{E}_X \left[ [\widehat{\eta}(x)]^\top \left[ \sum_{a \in \mathcal{A}} \frac{1}{\widehat{\omega}_a} \mathrm{Diag}(\widehat{\mathbf{q}}_a(x)) \widehat{\mathbf{D}}^{a,k} \right] h(x) \right], k \in K.
$$

Given samples $S = (x_n, a_n, y_n)_{n=1}^N$, we have that

$$
\begin{aligned}
|\widehat{\mathscr{D}}_k(h)| - |\widetilde{\mathscr{D}}_k(h)| &\leq |\widehat{\mathscr{D}}_k(h) - \widetilde{\mathscr{D}}_k(h)| \\
&= \left| \mathbb{E}_X \left[ [\widehat{\eta}(x)]^\top \left[ \sum_{a \in \mathcal{A}} \frac{1}{\widehat{\omega}_a} \mathrm{Diag}(\widehat{\mathbf{q}}_a(x)) \widehat{\mathbf{D}}^{a,k} \right] h(x) \right] - \frac{1}{N} \sum_{n=1}^{N} [\widehat{\eta}(x_n)]^\top \left[ \sum_{a \in \mathcal{A}} \frac{1}{\widehat{\omega}_a} \mathrm{Diag}(\widehat{\mathbf{q}}_a(x_n)) \widehat{\mathbf{D}}^{a,k} \right] h(x_n) \right| \\
&:= \left| \mathbb{E}_{\mathcal{P}} \left[ [\widehat{\phi}(x, k)]^\top h(x) \right] - \frac{1}{N} \sum_{n=1}^{N} [\widehat{\phi}(x_n, k)]^\top h(x_n) \right|,
\end{aligned}
$$

where we define

$$\phi(x,k) := \left( \sum_{a \in \mathcal{A}} \frac{1}{\omega_a} \mathrm{Diag}(\mathbf{q}_a(x)) \mathbf{D}^{a,k} \right)^{\top} \eta(x),$$

$$\widehat{\phi}(x,k) := \left( \sum_{a \in \mathcal{A}} \frac{1}{\widehat{\omega}_a} \mathrm{Diag}(\widehat{\mathbf{q}}_a(x)) \widehat{\mathbf{D}}^{a,k} \right)^{\top} \widehat{\eta}(x).$$

We have denoted $r_k := \max_a \|\mathbf{D}^{a,k}\|_\infty / \omega_{\min}$, the vector function $\phi(x,k)$ is $\ell_1$-bounded by $r_k$, since

$$\|\phi(x,k)\|_2 \leq \|\phi(x,k)\|_1 \leq \max_{a \in \mathcal{A}} \|\mathbf{D}^{a,k}\|_\infty \left\| \left( \sum_{a \in \mathcal{A}} \frac{1}{\omega_a} \mathrm{Diag}(\mathbf{q}_a(x)) \right) \eta(x) \right\|_1$$

$$\leq r_k \left\| \sum_{a \in \mathcal{A}} \mathrm{Diag}(\mathbf{q}_a(x)) \eta(x) \right\|_1$$

$$= r_k \left\| \sum_{a \in \mathcal{A}} [\mathbb{P}(A = a, Y = y \mid X = x)]_{y \in [m]} \right\|_1$$

$$= r_k$$

Following the same derivation, it shows that $\|\widehat{\phi}(x,k)\|_1 \leq \widehat{r}_k := \max_a \|\widehat{\mathbf{D}}^{a,k}\|_\infty / \widehat{\omega}_{\min}$. Note that take a union bound for Assumption C.12 and Lemma C.11 to make these events hold, we can obtain that, for $N \geq \max\{\frac{8}{p_{\min}} \log \frac{2|\mathcal{A}|}{\delta}, N_k(\{\mathbf{D}^{a,k}\}_{a \in \mathcal{A}, k \in K}, |\mathcal{A}|, \delta/2)\}$, with the probability at least $1 - \delta$,

$$\widehat{r}_k \leq \frac{4 \max_a \|\mathbf{D}^{a,k}\|_\infty}{\omega_{\min}} = 4r_k. \tag{36}$$

Moreover, considering the structure of optimal fair classifiers $\widehat{h}^\lambda \in \mathcal{H}_\Lambda : \mathcal{X} \to \Delta_m$ determined by the dual parameter $\lambda$, it presents that

$$\widehat{h}^\lambda(x) = \mathrm{softmax}(\widehat{\beta}^\lambda(x)/\tau)$$

$$\widehat{\beta}^\lambda(x) = \left[ \sum_{a \in \mathcal{A}} \mathrm{Diag}(\widehat{\mathbf{q}}_a(x)) \left( \mathbf{I} - \frac{1}{\widehat{\omega}_a} \sum_{k=1}^{K} \lambda_k \widehat{\mathbf{D}}^{a,k} \right) \right]^{\top} \widehat{\eta}(x),$$

$$= \widehat{\eta}(x) - \sum_{k=1}^{K} \lambda_k \widehat{\phi}(x,k).$$

Combining the fact that the softmax mapping is $\frac{1}{2}$-Lipschitz and plugging in the optimal fair classifier class $\mathcal{H}_\Lambda$, the generalization error of the fairness constraints satisfies that, for any $\delta \in (0,1)$, with probability at least $1 - \delta$ over i.i.d. samples $S \sim \mathcal{P}^N$,

$$|\widehat{\mathscr{D}}_k(\widehat{h}^\lambda)| - |\widetilde{\mathscr{D}}_k(\widehat{h}^\lambda)| \leq \left| \mathbb{E}_{\mathcal{P}} \left[ [\widehat{\phi}(x,k)]^{\top} \widehat{h}^\lambda(x) \right] - \frac{1}{N} \sum_{n=1}^{N} [\widehat{\phi}(x_n,k)]^{\top} \widehat{h}^\lambda(x_n) \right|,$$

$$\leq 2\widehat{\mathfrak{R}}_S(\widehat{\phi} \circ \mathcal{H}_\Lambda) + 3\sqrt{\frac{\log(4/\delta)}{2N}}$$

$$= 2\mathbb{E}_{\boldsymbol{\sigma}} \left[ \sup_{\widehat{h}^\lambda \in \mathcal{H}_\Lambda} \frac{1}{N} \sum_{n=1}^{N} \sigma_i \left\langle \widehat{\phi}(x_n,k), \widehat{h}^\lambda(x_n) \right\rangle \right] + 3\sqrt{\frac{\log(4/\delta)}{2N}} \tag{37}$$

$$\leq \frac{\sqrt{2}}{\tau} \widehat{r}_k \mathbb{E}_{\boldsymbol{\sigma}} \left[ \sup_{\lambda \in \Lambda} \frac{1}{N} \sum_{n=1}^{N} \sum_{i=1}^{m} \sigma_{ni} \widehat{\beta}_i^\lambda(x_n) \right] + 3\sqrt{\frac{\log(4/\delta)}{2N}}$$

The second inequality (line 2) is by Lemma C.9, and the third inequality (line 4) is by Lemma C.10. To analyze the above expression, we further investigate the structure of $\widehat{\beta}^\lambda$, denoting $\widehat{\phi}_i(x_n, \cdot) := [\widehat{\phi}_i(x_n, k)]_{k=1}^K, i \in [m]$,

$$
\begin{aligned}
\mathbb{E}_{\boldsymbol{\sigma}}\left[\sup_{\lambda \in \Lambda} \frac{1}{N} \sum_{n=1}^N \sum_{i=1}^m \sigma_{ni} \widehat{\beta}_i^\lambda(x_n)\right] &= \mathbb{E}_{\boldsymbol{\sigma}}\left[\sup_{\lambda \in \Lambda} \frac{1}{N} \sum_{n=1}^N \sum_{i=1}^m \sigma_{ni}\left(\widehat{\eta}_i(x_n) - \sum_{k=1}^K \lambda_k \widehat{\phi}_i(x_n, k)\right)\right] \\
&= \mathbb{E}_{\boldsymbol{\sigma}}\left[\sup_{\lambda \in \Lambda} \frac{1}{N}\left\langle \lambda, \sum_{n=1}^N \sum_{i=1}^m \sigma_{ni} \widehat{\phi}_i(x_n, \cdot)\right\rangle\right] \\
&\leq \frac{B_\Lambda}{N} \mathbb{E}_{\boldsymbol{\sigma}}\left[\left\|\sum_{n=1}^N \sum_{i=1}^m \sigma_{ni} \widehat{\phi}_i(x_n, \cdot)\right\|_2\right] \\
&\leq \frac{B_\Lambda}{N}\left[\mathbb{E}_{\boldsymbol{\sigma}}\left[\left\|\sum_{n=1}^N \sum_{i=1}^m \sigma_{ni} \widehat{\phi}_i(x_n, \cdot)\right\|_2^2\right]\right]^{\frac{1}{2}} \\
&= \frac{B_\Lambda}{N}\left[\mathbb{E}_{\boldsymbol{\sigma}}\left[\sum_{n=1}^N \sum_{i=1}^m \left\|\widehat{\phi}_i(x_n, \cdot)\right\|_2^2\right]\right]^{\frac{1}{2}} \\
&\leq B_\Lambda \|\widehat{r}\|_2 \sqrt{\frac{m}{N}}
\end{aligned}
\tag{38}
$$

The second equality (line 2) follows from the symmetry of the Rademacher complexity. The first inequality (line 3) follows from the Cauchy-Schwarz inequality, and the second inequality (line 4) follows from the Jensen's inequality. The subsequent equality (line 5) is a consequence of $\mathbb{E}_{\boldsymbol{\sigma}}[\sigma_i \sigma_j] = 0$ for $i \neq j$ and $\mathbb{E}_{\boldsymbol{\sigma}}[\sigma_i^2] = 1$ for $\forall i$. The last inequality (line 6) follows from the upper bound of $\widehat{\phi}(x, k)$ shown above.

Therefore, Combining (37) and (38), it shows that, for any $\delta \in (0, 1)$, with probability at least $1 - \delta$

$$
|\widehat{\mathscr{D}}_k(\widehat{h}^\lambda) - \widetilde{\mathscr{D}}_k(\widehat{h}^\lambda)| \leq \frac{\widehat{r}_k B_\Lambda}{\tau}\sqrt{\frac{2m\|\widehat{r}\|_2^2}{N}} + 3\sqrt{\frac{\log(4/\delta)}{2N}}
\tag{39}
$$

**(Plug-in Error.)** Next, we consider the plug-in error between $\mathscr{D}_k(h)$ and $\widetilde{\mathscr{D}}_k(h)$. For any $h \in \mathcal{H}$, given $k \in K$, since

$\widehat{\eta}(x), \widehat{\mathbf{q}}_a(x), h(x) \in \Delta_m, \forall x \in \mathcal{X}$, we can obtain that

$$
\begin{aligned}
|\mathscr{D}_k(h) - \widetilde{\mathscr{D}}_k(h)| \leq & \left| \mathbb{E}_X \left[ [\widehat{\eta}(x)]^\top \left[ \sum_{a \in \mathcal{A}} \frac{1}{\widehat{\omega}_a} \mathrm{Diag}(\widehat{\mathbf{q}}_a(x)) \left( \widehat{\mathbf{D}}^{a,k} - \mathbf{D}^{a,k} \right) \right] h(x) \right] \right| \\
& + \left| \mathbb{E}_X \left[ [\widehat{\eta}(x)]^\top \left[ \sum_{a \in \mathcal{A}} \left( \frac{1}{\widehat{\omega}_a} - \frac{1}{\omega_a} \right) \mathrm{Diag}(\widehat{\mathbf{q}}_a(x)) \mathbf{D}^{a,k} \right] h(x) \right] \right| \\
& + \left| \mathbb{E}_X \left[ [\widehat{\eta}(x)]^\top \left[ \sum_{a \in \mathcal{A}} \frac{1}{\omega_a} \mathrm{Diag}(\widehat{\mathbf{q}}_a(x) - \mathbf{q}_a(x)) \mathbf{D}^{a,k} \right] h(x) \right] \right| \\
& + \left| \mathbb{E}_X \left[ [\eta(x) - \widehat{\eta}(x)]^\top \left[ \sum_{a \in \mathcal{A}} \frac{1}{\omega_a} \mathrm{Diag}(\mathbf{q}_a(x)) \mathbf{D}^{a,k} \right] h(x) \right] \right| \\
\leq & \left| \mathbb{E}_X \left[ |\mathcal{A}| \frac{\max_a \|\mathbf{D}^{a,k} - \widehat{\mathbf{D}}^{a,k}\|_\infty}{\min_a \widehat{\omega}_a} \right] \right| + \left| \mathbb{E}_X \left[ \sum_{a \in \mathcal{A}} \frac{|\omega_a - \widehat{\omega}_a| \|\mathbf{D}^{a,k}\|_\infty}{\omega_a \cdot \widehat{\omega}_a} \right] \right| \\
& + \left| \mathbb{E}_X \left[ \sum_{a \in \mathcal{A}} \left\| [\widehat{\eta}(x_n)]^\top (\mathbf{q}_a(x_n) - \widehat{\mathbf{q}}_a(x_n)) \right\|_1 \left\| \frac{\mathbf{D}^{a,k} h(x_n)}{\omega_a} \right\|_\infty \right] \right| \\
& + \left| \mathbb{E}_X \left[ \|\eta(x) - \widehat{\eta}(x)\|_1 \left\| \sum_{a \in \mathcal{A}} \frac{1}{\omega_a} \mathrm{Diag}(\mathbf{q}_a(x)) \mathbf{D}^{a,k} h(x) \right\|_\infty \right] \right| \\
\leq & \left| \mathbb{E}_X \left[ |\mathcal{A}| \frac{\max_a \|\mathbf{D}^{a,k} - \widehat{\mathbf{D}}^{a,k}\|_\infty}{\min_a \widehat{\omega}_a} \right] \right| + \left| \mathbb{E}_X \left[ \sum_{a \in \mathcal{A}} \frac{|\omega_a - \widehat{\omega}_a| \|\mathbf{D}^{a,k}\|_\infty}{\omega_a \cdot \widehat{\omega}_a} \right] \right| \\
& + \left| \mathbb{E}_X \left[ \sum_{a \in \mathcal{A}} \|\mathbf{q}_a(x_n) - \widehat{\mathbf{q}}_a(x_n))\|_1 \frac{\|\mathbf{D}^{a,k}\|_\infty}{\omega_a} \right] \right| \\
& + \left| \mathbb{E}_X \left[ \|\eta(x) - \widehat{\eta}(x)\|_1 \frac{|\mathcal{A}| \max_a \|\mathbf{D}^{a,k}\|_\infty}{\omega_{\min}} \right] \right|
\end{aligned}
$$

If $N \geq \max\{\frac{8}{p_{\min}} \log \frac{4|\mathcal{A}|}{\delta}, N_k(\{\mathbf{D}^{a,k}\}_{a \in \mathcal{A}, k \in K}, |\mathcal{A}|, \delta/4)\}$, applying Lemma C.11, we have that, with probability at least $1 - \frac{\delta}{4}$, $\min_{a \in \mathcal{A}} \widehat{\omega}_a \geq \frac{1}{2}\omega_{\min}$. Besides, Eq. (36) shows that for $N$ chosen in this way, with probability at least $1 - \frac{\delta}{4}$, $\widehat{r}_k \leq 4r_k, \forall k \in K$, therefore $\|\widehat{r}\|_2 \leq 4\|r\|_2$. Lemma C.11 also shows that $\max_{a \in \mathcal{A}} |\widehat{\omega}_a - \omega_a| \leq \sqrt{\frac{1}{2N} \log \frac{8|\mathcal{A}|}{\delta}}$ holds with probability at least $1 - \frac{\delta}{4}$. Note that we assume $N \geq \max\{\frac{8}{p_{\min}} \log \frac{16K|\mathcal{A}|}{\delta}, N_k(\{\mathbf{D}^{a,k}\}_{a \in \mathcal{A}, k \in K}, |\mathcal{A}|, \delta/8K)\}$, the expressions above certainly hold.

Under these conditions, with $\ell_p$ norm defined on $(\mathcal{X}, \mu_X)$, i.e., $\|f\|_p := \left( \int_\mathcal{X} |f|^p d\mu_X \right)^{\frac{1}{p}}$, we obtain that

$$
\begin{aligned}
|\mathscr{D}_k(h) - \widetilde{\mathscr{D}}_k(h)| \leq & 2|\mathcal{A}| \frac{\max_a \|\mathbf{D}^{a,k} - \widehat{\mathbf{D}}^{a,k}\|_\infty}{\omega_{\min}} + \frac{2r_k}{\omega_{\min}} \sqrt{\frac{1}{2N} \log \frac{8|\mathcal{A}|}{\delta}} \\
& + r_k \sum_{a \in \mathcal{A}} \mathbb{E}_X \left[ \|\mathbf{q}_a(x_n) - \widehat{\mathbf{q}}_a(x_n))\|_1 \right] + r_k |\mathcal{A}| \mathbb{E}_X \left[ \|\eta(x) - \widehat{\eta}(x)\|_1 \right] \\
= & \frac{2|\mathcal{A}|}{\omega_{\min}} \Omega_k^N + \frac{2r_k}{\omega_{\min}} \sqrt{\frac{1}{2N} \log \frac{8|\mathcal{A}|}{\delta}} + r_k \left( \sum_{a \in \mathcal{A}} \|\mathbf{q}_a - \widehat{\mathbf{q}}_a\|_1 + |\mathcal{A}| \|\eta(x) - \widehat{\eta}(x)\|_1 \right)
\end{aligned}
$$

Therefore, combining the result above and taking an union bound to make events introduced by Lemma C.11 and Eq. (36)

hold, we get the generalization risk of fairness constraints that, for any $\delta \in (0, 1)$, with probability at least $1 - \delta$,

$$
\begin{aligned}
|\mathscr{D}_k(\widehat{h}^{\widehat{\lambda}^*})| &\leq |\widehat{\mathscr{D}}_k(\widehat{h}^{\widehat{\lambda}^*})| + |\widehat{\mathscr{D}}_k(\widehat{h}^{\widehat{\lambda}^*}) - \widetilde{\mathscr{D}}_k(\widehat{h}^{\widehat{\lambda}^*})| + |\widetilde{\mathscr{D}}_k(\widehat{h}^{\widehat{\lambda}^*}) - \mathscr{D}_k(\widehat{h}^{\widehat{\lambda}^*})| \\
&\leq \xi + r_k \left( \sum_{a \in \mathcal{A}} \|\mathbf{q}_a - \widehat{\mathbf{q}}_a\|_1 + |\mathcal{A}|\|\eta - \widehat{\eta}\|_1 \right) + \frac{2|\mathcal{A}|}{\omega_{\min}} \Omega_k^N \\
&\quad + \frac{16 r_k B_\Lambda}{\tau} \sqrt{\frac{m\|r\|_2^2}{N}} + 3\sqrt{\frac{\log(16/\delta)}{2N}} + \frac{r_k}{\omega_{\min}} \sqrt{\frac{2\log(8|\mathcal{A}|/\delta)}{N}} \\
&\leq \xi + r_k \left( \sum_{a \in \mathcal{A}} \|\mathbf{q}_a - \widehat{\mathbf{q}}_a\|_1 + |\mathcal{A}|\|\eta - \widehat{\eta}\|_1 \right) + \frac{2|\mathcal{A}|}{\omega_{\min}} \Omega_k^N \\
&\quad + \frac{16 r_k B_\Lambda}{\tau} \sqrt{\frac{m\|r\|_2^2}{N}} + \sqrt{\frac{(9 + 4r_k^2/\omega_{\min}^2)\log(8|\mathcal{A}|/\delta)}{N}}.
\end{aligned}
\tag{40}
$$

This completes the generalization risk analysis for fairness constraints.

**(Risk of Accuracy.)** Next, we turn to the generalization risk of accuracy. Considering the gap between the empirical fair classifier and the grounded-true Bayes-optimal fair classifier. Suppose that $(h^{\lambda^*}, \lambda^*)$ achieves the optimal equilibrium of the entropically regularized mini-max problem $\max_\lambda \min_h \mathcal{L}(h, \lambda)$, while $(\widehat{h}^{\widehat{\lambda}^*}, \widehat{\lambda}^*)$ achieves the corresponding empirical optimum. Since $(\widehat{\lambda}^*)^\top \widehat{\mathscr{D}}(\widehat{h}^{\widehat{\lambda}^*}) - \xi\|\widehat{\lambda}^*\|_1 = 0$ (derived by plugging the empirical sample distribution into Theorem 4.3), we obtain that

$$
\begin{aligned}
\mathcal{R}(\widehat{h}^{\widehat{\lambda}^*}) - \tau\mathcal{E}(\widehat{h}^{\widehat{\lambda}^*}) &= \mathcal{R}(\widehat{h}^{\widehat{\lambda}^*}) - \tau\mathcal{E}(\widehat{h}^{\widehat{\lambda}^*}) + (\widehat{\lambda}^*)^\top \widehat{\mathscr{D}}(\widehat{h}^{\widehat{\lambda}^*}) - \xi\|\widehat{\lambda}^*\|_1 \\
&= \mathcal{R}(\widehat{h}^{\widehat{\lambda}^*}) - \tau\mathcal{E}(\widehat{h}^{\widehat{\lambda}^*}) + (\widehat{\lambda}^*)^\top \mathscr{D}(\widehat{h}^{\widehat{\lambda}^*}) - \xi\|\widehat{\lambda}^*\|_1 + (\widehat{\lambda}^*)^\top \left( \widehat{\mathscr{D}}(\widehat{h}^{\widehat{\lambda}^*}) - \mathscr{D}(\widehat{h}^{\widehat{\lambda}^*}) \right) \\
&= \mathcal{L}(\widehat{h}^{\widehat{\lambda}^*}, \widehat{\lambda}^*) + (\widehat{\lambda}^*)^\top \left( \widehat{\mathscr{D}}(\widehat{h}^{\widehat{\lambda}^*}) - \mathscr{D}(\widehat{h}^{\widehat{\lambda}^*}) \right) \\
&\leq \mathcal{L}(h^{\lambda^*}, \lambda^*) + \left( \mathcal{L}(\widehat{h}^{\widehat{\lambda}^*}, \widehat{\lambda}^*) - \mathcal{L}(h^{\lambda^*}, \lambda^*) \right) + \|\widehat{\lambda}^*\|_\infty \left\| \widehat{\mathscr{D}}(\widehat{h}^{\widehat{\lambda}^*}) - \mathscr{D}(\widehat{h}^{\widehat{\lambda}^*}) \right\|_1.
\end{aligned}
$$

The last line is by the Hölder's inequality. We proceed to analyze the second term in the above expression. To bridge the grounded-true $(h^{\lambda^*}, \lambda^*)$ and the corresponding empirical solution $(\widehat{h}^{\widehat{\lambda}^*}, \widehat{\lambda}^*)$ under the Lagrangian function $\mathcal{L}$, we introduce the classifier $h^{\widehat{\lambda}^*}$, such that,

$$
h_i^{\widehat{\lambda}^*}(x) = \frac{\exp\left( \beta_i^{\widehat{\lambda}^*}(x)/\tau \right)}{\sum_{j=1}^m \exp\left( \beta_j^{\widehat{\lambda}^*}(x)/\tau \right)}, \quad i \in [m].
$$

The proof in Theorem 4.3 indicates that $h^{\widehat{\lambda}^*}(x)$ characterizes a solution of $\min_{h \in \mathcal{H}} \mathcal{L}(h, \widehat{\lambda}^*)$ given $\widehat{\lambda}^*$. Hence, it satisfies that $\mathcal{L}(h^{\widehat{\lambda}^*}, \widehat{\lambda}^*) \leq \mathcal{L}(h^{\lambda^*}, \lambda^*)$, which shows that

$$
\begin{aligned}
\mathcal{L}(\widehat{h}^{\widehat{\lambda}^*}, \widehat{\lambda}^*) - \mathcal{L}(h^{\lambda^*}, \lambda^*) &\leq \mathcal{L}(\widehat{h}^{\widehat{\lambda}^*}, \widehat{\lambda}^*) - \mathcal{L}(h^{\widehat{\lambda}^*}, \widehat{\lambda}^*) \\
&= \mathbb{E}_X \left[ \left( \beta^{\widehat{\lambda}^*}(x) \right)^\top \left( \widehat{h}^{\widehat{\lambda}^*}(x) - h^{\widehat{\lambda}^*}(x) \right) \right] - \tau\mathcal{E}(\widehat{h}^{\widehat{\lambda}^*}) + \tau\mathcal{E}(h^{\widehat{\lambda}^*}) \\
&\leq \mathbb{E}_X \left[ \left\| \beta^{\widehat{\lambda}^*}(x) \right\|_\infty \left\| \widehat{h}^{\widehat{\lambda}^*}(x) - h^{\widehat{\lambda}^*}(x) \right\|_1 \right] - \tau\mathcal{E}(\widehat{h}^{\widehat{\lambda}^*}) + \tau\mathcal{E}(h^{\widehat{\lambda}^*}) \\
&\leq \mathbb{E}_X \left[ (1 + B_\Lambda\|r\|_\infty) \left\| \widehat{h}^{\widehat{\lambda}^*}(x) - h^{\widehat{\lambda}^*}(x) \right\|_1 \right] - \tau\mathcal{E}(\widehat{h}^{\widehat{\lambda}^*}) + \tau\mathcal{E}(h^{\widehat{\lambda}^*})
\end{aligned}
$$

In the second inequality (line 3), we again apply Hölder's inequality. By the Lipschitz property of the softmax function, we

decompose $\left\|\widehat{h}^{\widehat{\lambda}^*}(x) - h^{\widehat{\lambda}^*}(x)\right\|_1$ as follows,

$$
\left\|\widehat{h}^{\widehat{\lambda}^*}(x) - h^{\widehat{\lambda}^*}(x)\right\|_1 = \left\|\text{softmax}\left(\frac{1}{\tau}\left(\widehat{\eta}(x) - \sum_{k=1}^{K}\widehat{\lambda}_k^*\widehat{\phi}(x,k)\right)\right) - \text{softmax}\left(\frac{1}{\tau}\left(\eta(x) - \sum_{k=1}^{K}\widehat{\lambda}_k^*\phi(x,k)\right)\right)\right\|_1
$$

$$
\leq \frac{1}{2\tau}\left\|\widehat{\eta}(x) - \eta(x) + \sum_{k=1}^{K}\widehat{\lambda}_k^*\left[\widehat{\phi}(x,k) - \phi(x,k)\right]\right\|_1
$$

$$
\leq \frac{1}{2\tau}\|\widehat{\eta}(x) - \eta(x)\|_1 + \frac{B_\Lambda}{2\tau}\sum_{k=1}^{K}\left\|\widehat{\phi}(x,k) - \phi(x,k)\right\|_1,
$$

Similar to the derivation of plug-in error above, it shows that

$$
\mathbb{E}_X\left[\left\|\widehat{\phi}(x,k) - \phi(x,k)\right\|_1\right] \leq m|\mathcal{A}|\frac{\max_a\|\mathbf{D}^{a,k} - \widehat{\mathbf{D}}^{a,k}\|_\infty}{\omega_{\min}} + m\sum_{a\in\mathcal{A}}\frac{|\omega_a - \widehat{\omega}_a|\|\mathbf{D}^{a,k}\|_\infty}{\omega_a\cdot\widehat{\omega}_a}
$$

$$
+ \sum_{a\in\mathcal{A}}\frac{1}{\omega_a}\mathbb{E}_X\|(\mathbf{D}^{a,k})^\top\text{Diag}(\mathbf{q}_a(x) - \widehat{\mathbf{q}}_a(x))\widehat{\eta}(x)\|_1
$$

$$
+ \mathbb{E}_X\left[\|\eta(x) - \widehat{\eta}(x)\|_1\frac{m|\mathcal{A}|\max_a\|\mathbf{D}^{a,k}\|_\infty}{\omega_{\min}}\right]
$$

$$
\leq m|\mathcal{A}|\frac{\max_a\|\mathbf{D}^{a,k} - \widehat{\mathbf{D}}^{a,k}\|_\infty}{\omega_{\min}} + m\sum_{a\in\mathcal{A}}\frac{|\omega_a - \widehat{\omega}_a|\|\mathbf{D}^{a,k}\|_\infty}{\omega_a\cdot\widehat{\omega}_a}
$$

$$
+ mr_k\sum_{a\in\mathcal{A}}\|\mathbf{q}_a - \widehat{\mathbf{q}}_a\|_1 + mr_k|\mathcal{A}|\|\eta - \widehat{\eta}\|_1
$$

Note that we assume $N \geq \max\{\frac{8}{p_{\min}}\log\frac{16K|\mathcal{A}|}{\delta}, N_k(\{\mathbf{D}^{a,k}\}_{a\in\mathcal{A},k\in K}, |\mathcal{A}|, \delta/8K)\}$, applying Lemma C.11, we have with probability at least $1 - \frac{\delta}{8K}$, it holds that $\min_{a\in\mathcal{A}}\widehat{\omega}_a \geq \frac{1}{2}\omega_{\min}$. Besides, Eq. (36) shows that for $N$ chosen in this way, with probability at least $1 - \frac{\delta}{8K}$, $\widehat{r}_k \leq 4r_k, \forall k \in K$, therefore $\|\widehat{r}\|_2 \leq 4\|r\|_2$. Lemma C.11 also shows that $\max_{a\in\mathcal{A}}|\widehat{\omega}_a - \omega_a| \leq \sqrt{\frac{1}{2N}\log\frac{16K|\mathcal{A}|}{\delta}}$ holds with probability at least $1 - \frac{\delta}{8K}$.

Taking an union bound to make these events introduced by Lemma C.11 and Eq. (36) hold, we have that, for any $\delta \in (0,1)$, with probability at least $1 - \frac{\delta}{2K}$,

$$
\mathbb{E}_X\left[\left\|\widehat{\phi}(x,k) - \phi(x,k)\right\|_1\right] \leq \frac{2|\mathcal{A}|m}{\omega_{\min}}\Omega_k^N + \frac{2mr_k}{\omega_{\min}}\sqrt{\frac{1}{2N}\log\frac{16K|\mathcal{A}|}{\delta}} + mr_k\left(\sum_{a\in\mathcal{A}}\|\mathbf{q}_a - \widehat{\mathbf{q}}_a\|_1 + |\mathcal{A}|\|\eta(x) - \widehat{\eta}(x)\|_1\right).
$$

Again, taking an union bound for $k \in [K]$, it shows that, with probability at least $1 - \frac{\delta}{2}$,

$$
\mathcal{L}(\widehat{h}^{\widehat{\lambda}^*}, \widehat{\lambda}^*) - \mathcal{L}(h^{\lambda^*}, \lambda^*) \leq \frac{1 + B_\Lambda\|r\|_\infty}{2\tau}\mathbb{E}_X\left[\|\widehat{\eta}(x) - \eta(x)\|_1 + B_\Lambda\sum_{k=1}^{K}\left\|\widehat{\phi}(x,k) - \phi(x,k)\right\|_1\right] - \tau\mathcal{E}(\widehat{h}^{\widehat{\lambda}^*}) + \tau\mathcal{E}(h^{\widehat{\lambda}^*})
$$

$$
\leq \frac{1 + B_\Lambda\|r\|_\infty}{2\tau}\left(\left(1 + B_\Lambda m|\mathcal{A}|\sum_{k=1}^{K}r_k\right)\|\eta - \widehat{\eta}\|_1 + B_\Lambda m\sum_{k=1}^{K}r_k\sum_{a\in\mathcal{A}}\|\mathbf{q}_a - \widehat{\mathbf{q}}_a\|_1\right)
$$

$$
+ \frac{mB_\Lambda(1 + B_\Lambda\|r\|_\infty)}{\tau\omega_{\min}}\sum_{k=1}^{K}\left(|\mathcal{A}|\Omega_k^N + r_k\sqrt{\frac{\log(16K|\mathcal{A}|/\delta)}{2N}}\right) - \tau(\mathcal{E}(\widehat{h}^{\widehat{\lambda}^*}) - \mathcal{E}(h^{\widehat{\lambda}^*})).
$$

(41)

Denoting $\delta' = \frac{\delta}{2K}$ and plugging it into the generalization bound of fairness constraints (Eq. (40)), it presents that, with

probability at least $1 - \frac{\delta}{2}$,

$$\sum_{k=1}^{K} |\widehat{\mathscr{D}}_k(\widehat{h^{\widehat{\lambda}^*}}) - \mathscr{D}_k(\widehat{h^{\widehat{\lambda}^*}})| \leq \sum_{k=1}^{K} r_k \left( \sum_{a \in \mathcal{A}} \|\mathbf{q}_a - \widehat{\mathbf{q}}_a\|_1 + |\mathcal{A}| \|\eta - \widehat{\eta}\|_1 \right) + \frac{2|\mathcal{A}|}{\omega_{\min}} \sum_{k=1}^{K} \Omega_k^N$$
$$+ \frac{16 \sum_{k=1}^{K} r_k B_\Lambda}{\tau} \sqrt{\frac{m\|r\|_2^2}{N}} + \sqrt{\frac{(9K^2 + 4\|r\|_2^2/\omega_{\min}^2) \log(16K|\mathcal{A}|/\delta)}{N}}. \tag{42}$$

Given that the entropic regularizer $0 \leq \mathcal{E}(h) \leq \log m$, by $(\lambda^*)^\top \mathscr{D}(h^{\lambda^*}) - \xi\|\lambda^*\|_1 = 0$, taking the union bound again for making events introduced by Eq. (41) and Eq. (42) hold, it presents that, for any $\delta \in (0, 1)$, with probability at least $1 - \delta$,

$$\mathcal{R}(\widehat{h^{\widehat{\lambda}^*}}) \leq \mathcal{R}(h^{\lambda^*}) - \tau \left( \mathcal{E}(h^{\lambda^*}) - \mathcal{E}(\widehat{h^{\widehat{\lambda}^*}}) \right) + \left( \mathcal{L}(\widehat{h^{\widehat{\lambda}^*}}, \widehat{\lambda}^*) - \mathcal{L}(h^{\lambda^*}, \lambda^*) \right) + B_\Lambda \sum_{k=1}^{K} |\widehat{\mathscr{D}}_k(\widehat{h^{\widehat{\lambda}^*}}) - \mathscr{D}_k(\widehat{h^{\widehat{\lambda}^*}})|$$

$$\leq \mathcal{R}(h^{\lambda^*}) + \tau \left| \mathcal{E}(h^{\lambda^*}) - \mathcal{E}(h^{\widehat{\lambda}^*}) \right| + \frac{mB_\Lambda(1 + B_\Lambda\|r\|_\infty)}{\tau\omega_{\min}} \sum_{k=1}^{K} \left( |\mathcal{A}|\Omega_k^N + r_k \sqrt{\frac{\log(16K|\mathcal{A}|/\delta)}{2N}} \right)$$

$$+ \frac{1 + B_\Lambda\|r\|_\infty}{2\tau} \left( \left( 1 + B_\Lambda m|\mathcal{A}| \sum_{k=1}^{K} r_k \right) \|\eta - \widehat{\eta}\|_1 + B_\Lambda m \sum_{k=1}^{K} r_k \sum_{a \in \mathcal{A}} \|\mathbf{q}_a - \widehat{\mathbf{q}}_a\|_1 \right)$$

$$+ \frac{16\|r\|_1^2 B_\Lambda^2}{\tau} \sqrt{\frac{m}{N}} + B_\Lambda \sqrt{\frac{(9K^2 + 4\|r\|_2^2/\omega_{\min}^2) \log(16K|\mathcal{A}|/\delta)}{N}}$$

$$+ \sum_{k=1}^{K} B_\Lambda r_k \left( \sum_{a \in \mathcal{A}} \|\mathbf{q}_a - \widehat{\mathbf{q}}_a\|_1 + |\mathcal{A}| \|\eta - \widehat{\eta}\|_1 \right) + \frac{2B_\Lambda|\mathcal{A}|}{\omega_{\min}} \sum_{k=1}^{K} \Omega_k^N.$$

Since $\left| \mathcal{E}(h^{\lambda^*}) - \mathcal{E}(h^{\widehat{\lambda}^*}) \right| \leq \log m$, by adjusting $\tau$ and using inequality $\sqrt{\sum_{i=1}^{K} \alpha_i} \leq \sum_{i=1}^{K} \sqrt{\alpha_i}$, for $\alpha_i \geq 0, i \in [K]$, the minimum value of right-hand-side upper bound can be achieved as follows,

$$\mathcal{R}(\widehat{h^{\widehat{\lambda}^*}}) \leq \mathcal{R}(h^{\lambda^*}) + 2\sqrt{\frac{m \log m B_\Lambda(1 + B_\Lambda\|r\|_\infty)}{\omega_{\min}}} \left( \left( |\mathcal{A}| \sum_{k=1}^{K} \Omega_k^N \right)^{\frac{1}{2}} + \left( \sum_{k=1}^{K} r_k \right)^{\frac{1}{2}} \left( \frac{\log(16K|\mathcal{A}|/\delta)}{2N} \right)^{\frac{1}{4}} \right)$$

$$+ 8\|r\|_1 B_\Lambda \sqrt{\log m} \left( \frac{m}{N} \right)^{\frac{1}{4}} + (1 + B_\Lambda m\|r\|_1) \sqrt{2 \log m} \left( |\mathcal{A}| \|\eta - \widehat{\eta}\|_1 + \sum_{a \in \mathcal{A}} \|\mathbf{q}_a - \widehat{\mathbf{q}}_a\|_1 \right)^{\frac{1}{2}}$$

$$+ \frac{2B_\Lambda|\mathcal{A}|}{\omega_{\min}} \sum_{k=1}^{K} \Omega_k^N + B_\Lambda\|r\|_1 \left( |\mathcal{A}| \|\eta - \widehat{\eta}\|_1 + \sum_{a \in \mathcal{A}} \|\mathbf{q}_a - \widehat{\mathbf{q}}_a\|_1 \right)$$

$$+ B_\Lambda \sqrt{\frac{(9K^2 + 4\|r\|_2^2/\omega_{\min}^2) \log(16K|\mathcal{A}|/\delta)}{N}}.$$

This completes the proof. $\qquad\square$

## D. Discussion for Attribute-Aware Case

In this section, we derive the optimal fair classifier for the attribute-aware case. The proof technique closely parallels that of the attribute-blind setting, with only a reformulation of the classifier structure required. We provide the following theorem and example.

**Theorem D.1.** *Under the assumption 4.1, for any $\xi > 0$, there exists an optimal solution to the problem (1), which can be*

*realized through the following form,*

$$h^*(x, a) \in \text{conv} \left\{ e_y : y \in \underset{j \in [m]}{\arg \max} \, \beta_j^{\lambda^*}(x, a) \right\}, \tag{43}$$

$$\beta^{\lambda}(x, a) = \omega_a [\mathbf{M}(a, \lambda)]^{\top} \eta(x, a), \tag{44}$$

$$\mathbf{M}(a, \lambda) := \mathbf{I} - \frac{1}{\omega_a} \sum_{k=1}^{K} \lambda_k \mathbf{D}^{a,k} \tag{45}$$

*the dual parameter $\lambda \in \mathbb{R}^m$, and $\text{conv}\{\cdot\}$ denotes the convex hull. Denoting $\Lambda := \{\lambda \in \mathbb{R}^K : \|\lambda\|_1 \leq B_\Lambda\}$, the parameter $\lambda^*$ is the solution of the dual problem:*

$$\min_{\lambda \in \Lambda} H(\lambda) = \mathbb{E}_{X,A} \left[ \max_{j \in [m]} \beta_j^{\lambda}(X, A) \right] + \xi \|\lambda\|_1.$$

The proof of this theorem proceeds analogously to that of Theorem D.1, and we omit the details for brevity.

Next, we take the DP constraint as an example to illustrate how Theorem 1 can be applied to derive the optimal fair classifier under a specific fairness constraint.

**Example D.2.** (DP.) *Using the constraints for demographic parity as described in Table 1, denoting the dual parameter as $\lambda \in \mathbb{R}^{m \times |\mathcal{A}|}$, its optimal fair classifier is determined by the decision vector $\beta^{\lambda}(x, a) \in \mathbb{R}^m$, where*

$$\beta_i^{\lambda}(x) = \omega_a \eta_i(x, a) + \left( \omega_a \sum_{a' \in \mathcal{A}} \lambda_{i,a'} - \lambda_{i,a} \right), i \in [m]. \tag{46}$$

*Plugging (46) into Theorem D.1 can obtain the optimal fair classifier for the fairness-constrained problem.*

**Proof.** We have shown that the constraint matrix for DP is $\mathbf{D}^{a,(y,a')}$, where the $y$-th column elements of $\mathbf{D}^{a,(y,a')}$ are $\mathbb{I}[a = a'] - \omega_a$ with all other elements set to 0. Plugging it into the expression of $\beta^{\lambda}(x, a)$, we obtain that

$$\begin{aligned} \beta^{\lambda}(x, a) &= \left( \omega_a \mathbf{I} - \sum_{y \in [m], a' \in \mathcal{A}} \lambda_{y,a'} \mathbf{D}^{a,(y,a')} \right)^{\top} \eta(x, a) \\ &= \omega_a \eta(x, a) - \sum_{y \in [m], a' \in \mathcal{A}} \lambda_{y,a'} \left( \mathbb{I}(a = a') - \omega_a \right) \mathbf{e}_y \\ &= \omega_a \eta(x, a) - \sum_{y \in [m]} \lambda_{y,a} \mathbf{e}_y + \omega_a \sum_{y \in [m], a' \in \mathcal{A}} \lambda_{y,a'} \mathbf{e}_y \\ &= \omega_a \eta(x, a) + \left( \omega_a \sum_{y \in [m], a' \in \mathcal{A}} \lambda_{y,a'} \mathbf{e}_y - \sum_{y \in [m]} \lambda_{y,a} \mathbf{e}_y \right). \end{aligned}$$

Consider the $i$-th element in $\beta^{\lambda}(x, a)$, we get that

$$\beta_i^{\lambda}(x) = \omega_a \eta_i(x, a) + \left( \omega_a \sum_{a' \in \mathcal{A}} \lambda_{i,a'} - \lambda_{i,a} \right), i \in [m].$$

This completes the proof. □

Plugging the expression of $\beta^{\lambda}(x, a)$ into the classifier and dual problem in D.1, we obtain the Bayes-optimal classifier under DP constraint. Note that the structure of the optimal DP-fair classifier is very similar to that given in Theorem 5.1 in (Denis et al., 2024). In particular, the first component of the decision vector $\beta^{\lambda}(x, a)$ takes exactly the same form, while the second component differs slightly due to the fact that Denis adopts a group-wise DP constraint, whereas this paper considers a group-to-center DP constraint.

# E. Detailed Experimental Setting

## E.1. Datasets

- The **Adult** dataset (Asuncion et al., 2007) comprises more than 45000 samples based on 1994 U.S. census data, where the task is to predict whether the annual income of an individual is above $50,000. We consider the gender of each individual as the sensitive attribute and train the logistic regression as the classification model.

- The **ENEM** dataset (INEP, 2018) contains about 1.4 million samples from Brazilian college entrance exam scores along with student demographic information. We follow (Alghamdi et al., 2022) to quantized the exam score into 5 classes as label, and consider race as sensitive attribute. We train two-layer MLP as the classification model.

- The **ACSIncome** (Ding et al., 2021) extends the Adult dataset, containing much more examples ($1,664,500$ in total) from recent U.S. census data. We consider a multi-group multiclass setting with race as the sensitive attribute and five income buckets ($|\mathcal{Y}| = |\mathcal{A}| = 5$) following (Xian & Zhao, 2024), with the one-layer MLP (Multilayer Perceptron) as the model backbone.

- The **CelebA** dataset (Zhang et al., 2020) is a facial image dataset consists of about 200k instances with 40 binary attribute annotations. We identify 'Age' as the sensitive attribute, and employ the binary attributes "Smile" and "Big_Nose" to construct a multiclass task by mapping their joint values $\{0,1\} \times \{0,1\}$ onto a four-class label set $\{0,1,2,3\}$, thereby formulating a multiclass classification problem on the CelebA dataset. We then train Resnet-10 (He et al., 2016) on CelebA as the classification model.

The determination of sensitive attributes and labels on these datasets has been verified significant in previous research (Alghamdi et al., 2022; Han et al., 2024; Xian et al., 2023).

## E.2. Baselines

We compare the performance of `OptFair` with the the standard baseline **ERM** (Empirical Risk Minimization) and four SOTA methods tailored for attribute-blind fairness calibration. These baselines are selected from past literature based on two criteria: (1) applicability to multi-class fairness calibration and (2) ability to handle various fairness concepts in the attribute-blind setting.

*Standard baseline*:

- **ERM**. Empirical risk minimization is a standard machine learning method that minimizes the empirical risk over the training data. It serves as a common baseline for fairness methods.

*In-processing* algorithms:

- **AdvDebias** (Zhang et al., 2018). Adversarial debiasing maximizes a classifier's predictive ability while simultaneously minimizing an adversary's ability to predict sensitive attributes from the predictions. We use the implementation from the Fairlearn (Weerts et al., 2023) library for demographic parity and equal Odds, and use our own implementation for multiclass equal opportunity.

- **Weighted-ERM** (Yang et al., 2020). This algorithm attempts to extend the reduction algorithm (Agarwal et al., 2018) to the overlap fairness scenario, which is also adaptable to the multiclass setting and enables unfairness mitigation for arbitrary machine learning models with respect to confusion-matrix-based fairness constraints.

- **FairBatch** (Roh et al., 2021). By adjusting group sampling probabilities based on observed fairness violations, this method steers standard ERM optimization toward a better accuracy–fairness trade-off without changing the model architecture. The multi-class extension of FairBatch adapting its group-wise sampling/reweighting rule to multi-class fairness constraints is built following the approach described in Appendix B.8 of the original paper.

- **F-divergence** (Zhong & Tandon, 2023). The method achieves group-level distributional alignment by incorporating an $f$-divergence regularizer between group-conditioned predictive distributions into the task loss, which is similar to kernel estimation methods in binary fairness settings (Cho et al., 2020). It alternates between variational discriminator and classifier updates to enforce DP/EO constraints without retraining the model architecture.

*Post-processing* algorithms:

- **Fairprojection** (Alghamdi et al., 2022). This algorithm yeilds a fair classifier by multiplicatively tilting/calibrating the original class-probability outputs with group-dependent factors estimated from data, achieving an improved accuracy–fairness trade-off without retraining the backbone model. The KL divergence is used to quantify the divergence.

- **LinearPost** (Xian & Zhao, 2024). This post-processing method for multi-class classification, which can be applied to various fairness concepts, including DP, EOP, and EO. It formulates the fairness post-processing problem as a linear program and constructs a generalizable optimal fair classifier by solving the dual problem.

- **FRAPPÉ** (Tifrea et al., 2024). FRAPPÉ is a general post-processing framework that turns regularized in-processing fairness objectives (e.g., MinDiff-style penalties) into a standalone debiasing module trained on top of a fixed, pre-trained predictor. Concretely, it learns a lightweight transformation of the model's predictions to improve the accuracy-fairness trade-off while keeping the original training pipeline unchanged.

Meanwhile, we adapt `OptFair` to focus solely on in-processing or post-processing to calibrate group fairness, denoted as **OptFair-in** and **OptFair-post**. Here we do not perform model output distribution calibration for all post-processing methods to test their original debiasing ability.

### E.3. Parameter Settings

We provide hyperparameter selection ranges for each model in Table 2. For all other hyperparameters, we follow the codes provided by the authors and retain their default parameter settings.

*Table 2.* Hyperparameter Selection Ranges

| Model | Hyperparameter | Ranges |
|---|---|---|
| **General** | Learning rate | {0.0001, 0.001, 0.003, 0.005, 0.01, 0.03 ,0.05} |
| | Epoch | {20, 30, 50, 80, 100} |
| | Optimizer | {Adam, SGD} |
| **AdvDeBias** | Step size ($\alpha$) | {0.005, 0.01, 0.05, 0.3} |
| **FairBatch** | Fairness budget ($\lambda$) | {0.01, 0.05, 0.1, 0.3, 1} |
| **F-divergence** | Fairness regularizer | {0.01, 0.05, 0.1, 0.3, 0.8} |
| **Fairprojection** | Distance | KL |
| **FRAPPÉ** | Fairness regularizer | {0.01, 0.05, 0.1, 0.3, 0.8} |
| **OptFair-In** | Classifier number | 1 |
| | dual learning rate ($\eta_\lambda$) | {0.001, 0.005, 0.01, 0.05, 0.1} |
| | Dual parameter bound | 10 |
| **OptFair-Post** | Temperature $\tau$ | {0.05, 0.02} |
| | Dual parameter bound | 10 |

### E.4. Experiments Compute Resources

We conducted our experiments on a GPU server equipped with 8 CPUs and eight NVIDIA RTX 5090s (32GB).

# F. Additional Experiments

## F.1. Deterministic versus Randomized Classifiers

To assess the performance gap between deterministic and randomized classifiers, we evaluate the deterministic variants (described in Section 5.3) on four datasets under DP constraints $\epsilon = 0.001$, with results averaged over three runs.

*Table 3.* Performance comparison of randomized and deterministic classifiers under DP constraints.

| Method | Adult | | ENEM | | ACS | | CelebA | |
|---|---|---|---|---|---|---|---|---|
| | Acc | Bias | Acc | Bias | Acc | Bias | Acc | Bias |
| OptFair-in-rand | 82.82 | 0.0046 | 29.58 | 0.0187 | 46.99 | 0.0259 | 71.51 | 0.0160 |
| OptFair-in-det | 82.74 | 0.0042 | 29.45 | 0.0293 | 47.13 | 0.0332 | 72.41 | 0.0169 |
| OptFair-post-rand | 83.16 | 0.0054 | 27.52 | 0.0057 | 44.31 | 0.0553 | 73.83 | 0.0171 |
| OptFair-post-det | 83.25 | 0.0065 | 27.65 | 0.0074 | 44.42 | 0.0586 | 73.96 | 0.0183 |

Although deterministic classifiers induce a smaller hypothesis class than randomized classifiers, our results show that the deterministic variants remain very close to their randomized counterparts. This indicates that the performance gains of our methods over existing deterministic baselines do not simply stem from the use of randomized decision rules.

## F.2. Scalability over the Number of Classes and Sensitive Groups

Since existing fair benchmarks rarely contain very large numbers of sensitive groups or classes, we construct datasets $C_1, C_2, C_3$ from CelebA by intersecting the first $k \in \{1, 2, 3\}$ attributes from (Young, Male, Black Hair) and the first $k + 1$ attributes from (Smile, Big Nose, Attractive, Bag Under Eyes ), yielding $(2, 4), (4, 8), (8, 16)$ group×class settings. ResNet18 is used under EO constraints with $\xi = 0.05$.

*Table 4.* Performance scaling over different numbers of classes and groups.

| Method | $C_1$ | | $C_2$ | | $C_3$ | |
|---|---|---|---|---|---|---|
| | Acc | Bias | Acc | Bias | Acc | Bias |
| ERM | 75.77 | 0.1456 | 63.46 | 0.4372 | 56.26 | 0.7138 |
| OptFair-in | 75.24 | 0.0591 | 58.45 | 0.0914 | 48.89 | 0.3476 |
| OptFair-post | 75.17 | 0.0563 | 60.85 | 0.0831 | 46.35 | 0.2604 |

The results show that fairness becomes more challenging as groups and classes increase, but `OptFair` can still improve fairness, providing empirical evidence of scalability.

