# OpenReview forum: "Demystifying the Optimal Fair Classifier in Multi-Class Classification"
_ICML.cc/2026/Conference — ICML 2026 regular_

### Official Review · Reviewer_HRPM · 2026-02-25

**Soundness:** 3
**Presentation:** 4
**Significance:** 3
**Originality:** 3
**Overall Recommendation:** 5
**Confidence:** 3

**Summary:**

The paper attempts to characterize and achieve the optimal accuracy-fairness trade-off in multi-class classification settings. The authors propose an analytically tractable formulation of the Bayes-optimal fair classifier by introducing an entropic regularizer, which smooths the otherwise non-differentiable fairness constraints. Hence, the paper presents a unified framework, OptFair, which includes both an in-processing algorithm (via reduction to cost-sensitive learning) and a post-processing algorithm (via plug-in estimation). Theoretical analysis is provided to guarantee convergence to the Pareto frontier, and experiments are conducted on four datasets of different scales and modalities.

**Compliance With Llm Reviewing Policy:**

Affirmed.

**Final Justification:**

My concerns are fully resolved, so I maintain my **Accept** recommendation for the solid paper.

**Key Questions For Authors:**

1. Could you provide results using F1-score in addition to Accuracy?
2. How does the computational cost of the in-processing method compare to the post-processing method in practice, especially as the number of classes and sensitive attributes scales up?

**Limitations:**

Yes.

**Strengths And Weaknesses:**

### Strengths

**1. Comprehensive and Unified Framework:** The paper stands out by providing a systematic solution that covers both in-processing and post-processing paradigms. Deriving both algorithms from the same theoretical foundation (the entropy-regularized optimal classifier) is conceptually appealing and rigorous.

**2. Theoretical Solidness:** The characterization of the multi-class Pareto frontier is a significant contribution. The use of entropic regularization to handle the non-differentiability of fairness constraints is an elegant mathematical trick that makes the optimization tractable. The convergence guarantees add to the soundness of the proposed method.

**3. Clarity:** The paper is exceptionally well-written.

### Weaknesses

**1. Evaluation Metrics:** The experimental evaluation primarily relies on Accuracy as the performance metric. In multi-class classification tasks, class imbalance is a common issue. Accuracy can be misleading in such cases. It would be much more convincing if the authors included other utility metics like F1-score.

---

> ### Author Rebuttal · Authors · 2026-03-31
>
> ### For Weakness
>
> > **W1:** Evaluation Metrics.
>
> We appreciate your helpful suggestion. We fully agree that, in multiclass classification, accuracy alone may be insufficient, especially under class imbalance. To address this concern, we provide additional experimental results in **our response to Q1**.
>
> ### For Questions
>
> > **Q1:** Results of F1-score.
>
> We further report the **Macro-F1** score of our method, which provides a more balanced evaluation of utility across classes. The experiments are conducted on four datasets following the settings in the main paper. Due to space limitations, the results are presented only in table form, with $\xi = 0.001$ and DP constraints, averaged over three runs.
>
> |Dataset|Adult| |ENEM| |ACS| |CelebA| |
> |---|---|---|---|---|---|---|---|---|
> |Method|F1|Bias|F1|Bias|F1|Bias|F1|Bias|
> |AdvDeBias|0.7162|0.0096|0.1962|0.0217|0.3064|0.0876|0.6248|0.0314|
> |Weight-ERM|0.7037|0.0259|0.3026|0.1437|0.4642|0.1519|0.6480|0.0418|
> |FairBatch|0.7329|0.0106|0.2884|0.0857|0.4473|0.0916|0.6519|0.0494|
> |F-divergence|0.7330|0.0129|0.2515|0.0866|0.3840|0.0496|0.5982|0.0556|
> |FairProjection|0.6962|0.0070|0.2425|0.0451|0.3627|0.0580|0.5507|0.0364|
> |LinearPost|0.7341|0.0091|0.2389|0.0165|0.4635|0.0631|0.6211|0.0307|
> |FRAPPE|0.7264|0.0205|0.2894|0.0969|0.4398|0.1098|0.6076|0.0592|
> |OptFair-in|0.7338|0.0046|0.2708|0.0187|0.4472|0.0259|0.6613|0.0160|
> |OptFair-post|0.7375|0.0026|0.2784|0.0057|0.4583|0.0553|0.6332|0.0171|
>
> The results show that OptFair achieves a competitive utility-fairness trade-off under Macro-F1 in most cases. Notably, some methods may achieve higher accuracy than OptFair, but their fairness guarantees are substantially weaker. Overall, the conclusions remain consistent with those based on accuracy.
>
> > **Q2:** The computational cost.
>
> **Response:** Thank you for this valuable question regarding computational cost. We discuss the computational cost from both the theoretical and empirical perspectives.
>
> **Computational complexity.** Let $T$ denote the number of outer iterations, $R$ the number of inner training iterations for in-processing, $d$ the model parameter size, $n$ the sample size, and $K$ the number of fairness constraints.
>
> - **For in-processing**, the runtime is dominated by iterative updates of the classifier and dual variables. According to Algorithm 1, its computational complexity is $O(T(Rnd + nK))$. By following the reduction-based paradigm, this complexity is of the same order as prior binary in-processing methods [1,2].
>
> - **For post-processing**, given a pretrained model, the complexity is only $O(TnK)$. This is enabled by our entropy-regularized representation, which makes the fair-classification problem uniquely solvable and allows us to adopt an efficient proximal-gradient-based method. Compared with prior approaches based on solving linear programs [3,4], whose complexity can scale polynomially with $nK$, our method significantly reduces the computational cost.
>
> - **In-processing vs. post-processing.** Since in modern deep models the parameter dimension typically satisfies $d \gg K$, the above complexity analysis suggests that post-processing is substantially cheaper than in-processing. Moreover, the post-processing objective is convex, which also leads to fewer iterations in practice. Overall, post-processing is much faster than in-processing.
>
> - **Effect of the number of classes and sensitive attributes.** As the number of classes and sensitive attributes increases, the number of constraints $K$ may grow linearly or polynomially, depending on the fairness notion. For in-processing, this has limited impact because $d \gg K$ typically holds. For post-processing, the runtime does increase with $K$, but **only linearly**; since $K$ is still much smaller than the parameter size of modern models, this increase is usually moderate in practice.
>
> - **Empirical results.** We report the runtime comparison of the two methods on four datasets using a single RTX 5090 (32GB) GPU. In these experiments, both methods were run until convergence, with approximately $T \approx 40$ for in-processing and $T \approx 20$ for post-processing. The results are shown below.
>
> |Dataset|Adult|ENEM|ACS|CelebA|
> |---|---|---|---|---|
> |OptFair-in|205.5s|1531.9s|1605.4s|2590.7s|
> |OptFair-post|8.8s|125.5s|153.5s|103.9s|
>
> The empirical results confirm our analysis that post-processing is more efficient than in-processing in practice, and this advantage becomes even more pronounced as the model size increases.
>
> We sincerely appreciate your valuable questions, espically on evaluation metrics and computational cost. We hope this resolves your concerns.
>
> [1] A Reductions Approach to Fair Classification
>
> [2] Bayes-Optimal Fair Classification with Linear Disparity Constraints via Pre-, In-, and Post-processing
>
> [3] Fair and Optimal Classification via Post-Processing
>
> [4] A Unified Post-Processing Framework for Group Fairness in Classification

---

> > ### Author Rebuttal · Reviewer_HRPM · 2026-04-02
> >
> > The Macro-F1 experiments and computational cost analysis adequately address my concerns. **I maintain my Accept recommendation.**

---

### Official Review · Reviewer_d4mU · 2026-03-06

**Soundness:** 3
**Presentation:** 4
**Significance:** 3
**Originality:** 3
**Overall Recommendation:** 5
**Confidence:** 3

**Summary:**

This paper addresses fairness in multiclass classification, a scenario little explored in the literature compared to the binary case. The authors propose OptFair, a calibration framework based on group fairness, which aims to optimize accuracy subject to explicit fairness constraints.
The approach is based on representing fairness notions using probabilistic confusion matrices, which encode the quantities needed to calculate metrics such as DP, EOP and EO. From this formulation, the authors derive a probabilistic characterization of the optimal fair classifier and make explicit the Pareto frontier between accuracy and fairness. The framework is implemented using two strategies, in-processing and post-processing.

The experiments cover multiple datasets, architectures and compare against several SOTA methods for both in-processing and post-processing. The results show that OptFair achieves better control of the accuracy-fairness trade-off and empirically approximates the proposed frontier.

**Compliance With Llm Reviewing Policy:**

Affirmed.

**Final Justification:**

The authors addressed my questions in the rebuttal. I have no changes to my recommendation, as it was already positive.

**Key Questions For Authors:**

1. How does OptFair scale when the number of sensitive classes or groups is very large?
2. What happens in scenarios where sensitive attributes are partially observed or contain noise?
3. Does the method remain stable when classes are highly unbalanced?

**Limitations:**

First (and least important), the approach is restricted to the group fairness paradigm, leaving unexplored alternative notions such as individual fairness. On the other hand, the confusion matrix-based formulation introduces a number of variables and constraints that can grow rapidly with the number of classes and groups, which could affect scalability in more complex scenarios. Finally, the method also assumes explicit availability of sensitive attributes during training, a condition that is not always met in real-world applications. However, these limitations do not undermine the main contribution of the paper, which offers a clear and theoretical perspective on the trade-off between accuracy and fairness beyond the traditional binary case.

**Strengths And Weaknesses:**

Strengths:
- The extension of fairness analysis to the multi-class case is truly a valuable contribution.
- The use of confusion matrices allows different fairness metrics to be interpreted under the same mathematical formalism. This provides conceptual clarity.
- Unlike many more empirical fairness studies, there is a theoretical derivation here that directly guides the subsequent algorithmic design.
- Multiple datasets, architectures (LR, MLP, ResNet) and comparisons against various SOTA methods are considered.

Weaknesses:
- The theoretical presentation is difficult to follow in some passages.
- The method assumes explicit access to sensitive attributes, and this is not always guaranteed in real-world applications.
- The number of constraints grows with the number of sensitive classes and groups, which could hinder the scalability of the approach with many classes or multiple sensitive attributes.

---

> ### Author Rebuttal · Authors · 2026-03-31
>
> ### For Weaknesses
> > **W1:** The theoretical presentation is difficult to follow.
>
> **Response:** Thank you for this helpful comment. We find that some theoretical passages can be made clearer (e.g. convergence and generalization analysis). We will improve readability by adding more intuition and better explaining the roles of these theorems and proof steps.
>
> > **W2:** Sensitive attributes may be missed.
>
> **Response:** We fully agree that sensitive attributes are not always available in practice. Our work studies multiclass classification under group fairness constraints, a standard setting in the fairness literature that typically assumes access to sensitive attributes during training. Moreover, our method learns an attribute-blind classifier, so sensitive attributes are not needed at test time.
>
> > **W3:** The number of constraints grows with classes and sensitive groups, limiting scalability.
>
>
> **Response:** Thank you for this important point. We would like to clarify that the growth in constraints is due to the underlying definition of multiclass fairness notions (e.g. different DP in [1, 2]), rather than the confusion-matrix-based formulation itself, which only provides a unified representation (Appendix A). A future direction is to develop fairness notions that remain well-scaled as the numbers of classes and sensitive groups increase.
>
> ### For Questions
>
> > **Q1:** Scalability w.r.t. the number of sensitive classes or groups.
>
> **Response:** **Theoretically,** increasing the number of groups and classes increases the number of fairness constraints, which may slow down the convergence of OptFair-in (Theorem 5.2). By contrast, OptFair-post is less affected since its objective is convex. **Empirically**, since **existing fair benchmarks rarely contain very large numbers of sensitive groups or classes**, we construct datasets $C_1,C_2,C_3$ from CelebA by intersecting the first $k \in$ \{$1,2,3$\} attributes from {Young, Male, Black_Hair} and the first $k+1$ attributes from {Smile, Big_Nose, Attractive, Bag_Under_Eyes}, yielding $(2,4),(4,8),(8,16)$, group$\times$class settings. ResNet18 is used under EO constraints with $\xi=0.05$. All results below are averaged over 3 runs.
>
> |Dataset|C$_1$| |C$_2$| |C$_3$| |
> |---|---|---|---|---|---|---|
> |Method|Acc|Bias|Acc|Bias|Acc|Bias|
> |ERM|75.77|0.1456|63.46|0.4372|56.26|0.7138|
> |OptFair-in|75.24|0.0591|58.45|0.0914|48.89|0.3476|
> |OptFair-post|75.17|0.0563|60.85|0.0831|46.35|0.2604|
>
> The results show that fairness becomes more challenging as groups and classes increase, but OptFair still substantially improves fairness, providing empirical evidence of scalability.
>
> > **Q2:** Partially observed or noisy sensitive attributes.
>
> **Response:** We discuss this issue from two perspectives:
> For **partially observed sensitive attributes**, a post-processing-compatible solution is to train the base model on unlabeled samples and use the labeled subset for fairness calibration [3]. Methods like auxiliary prediction of missing attributes are beyond our scope.
> For **noisy sensitive attributes**, we corrupt sensitive attributes with symmetric noise at rates $r = 0.1, 0.2$, and evaluate DP on Adult and ENEM with $\xi = 0.01$.
>
> |Dataset|Adult| | | |ENEM| | | |
> |---|---|---|---|---|---|---|---|---|
> |Corruption|r=0.1| |r=0.2| |r=0.1| |r=0.2| |
> |Method|Acc|Bias|Acc|Bias|Acc|Bias|Acc|Bias|
> |OptFair-in|83.78|0.0335|83.85|0.0363|30.67|0.0771|32.99|0.1084|
> |OptFair-post|84.21|0.0273|84.67|0.0466|28.52|0.0305|28.85|0.0660|
>
> The results show that noise may prevent OptFair from exactly meeting the target fairness level, but it remains close to the Pareto-optimal region.
>
> > **Q3:** Stability when classes are highly unbalanced.
>
> Thank you for this important question. The datasets used in our paper are already relatively class-imbalanced (e.g., in CelebA, the smallest class accounts for less than 15%). To examine a more extreme regime, we generate class proportions using a Dirichlet distribution with $\gamma = 0.2,0.3$, and evaluate EO on Adult and ENEM with $\xi=0.01$.
>
> |Dataset|Adult| | | |ENEM| | | |
> |---|---|---|---|---|---|---|---|---|
> |Imbalance|$\gamma=0.2$| |$\gamma=0.3$| |$\gamma=0.2$| |$\gamma=0.3$| |
> |Method|Acc|Bias|Acc|Bias|Acc|Bias|Acc|Bias|
> |ERM|85.34|0.1829|83.48|0.1816|72.79|0.1439|47.42|0.2306|
> |OptFair-in|82.39|0.0145|80.82|0.0136|69.77|0.0222|44.99|0.0592|
> |OptFair-post|82.09|0.0122|79.42|0.0241|67.46|0.0392|41.23|0.0465|
>
> The results show that OptFair remains stable and continues to achieve a competitive fairness-utility trade-off under more severe class imbalance.
>
> We really appreciate your insightful feedback, especially on missing or corrupted sensitive attributes, scalability and stability. We hope the discussion above resolves your concerns.
>
> [1] Beyond Adult and COMPAS: Fair Multi-Class Prediction via Information Projection
>
> [2] Fair and Optimal Classification via Post-Processing
>
> [3] Leveraging Labeled and Unlabeled Data for Consistent Fair Binary Classification

---

> > ### Author Rebuttal · Reviewer_d4mU · 2026-03-31
> >
> > I am grateful to the authors for conducting additional experiments on scalability and robustness to noise. These results strengthen the validation of their proposed method.

---

### Official Review · Reviewer_n2z9 · 2026-03-10

**Soundness:** 4
**Presentation:** 3
**Significance:** 3
**Originality:** 2
**Overall Recommendation:** 5
**Confidence:** 4

**Summary:**

This paper considers group fairness (notions like SP, EO) in multi-class classification.

1. The authors studied the form of the probabilistic optimal fair classifier, and derived a representation result for it via first writing down the fair classification problem as a linear program then analyzing the dual form. More interestingly, they resolved a non-uniqueness issue with the representation (i.e., the result in eq. (2) is $h(x)\in \cdots$ as opposed to $h(x) = \cdots$) by adding entropic regularization to the loss objective (we then get eq. (10)), which borrows ideas from the optimal transport literature.
2. Based on representation result, the authors adapted the Reductions algorithm to propose (1) an in-processing algorithm via optimizing the potential function $\beta$ in the representation, and (2) a post-processing algorithm.
3. Empirical results show good performance in both in-processing and post-processing settings.

**Compliance With Llm Reviewing Policy:**

Affirmed.

**Final Justification:**

The rebuttal has address my concerns; overall a solid contribution!

**Key Questions For Authors:**

1. See weakness 3.1. In particular, it seems to the reviewer that Theorem 4.2 has already been established in [1].
2. Since the classifier learned under the proposed method is probabilistic, how is the experiment setup?  Sample a prediction according to the probabilities (which I suspect will break the fairness guarantees), or taking the argmax.
3. Could the authors elaborate on the choice of entropic regularization $\tau$?  It seems that taking $\tau\rightarrow 0$ is desirable in terms of classifier performance, but may be harder to optimize or not possible due to numerical precision.
4. Could the authors comment on the running time of the algorithm?

**Limitations:**

See weaknesses

**Strengths And Weaknesses:**

**Strengths.**

- This work is solid: it provides a concrete representation result for the optimal fair classifier, and operationalizes the result by proposing in-processing and post-processing algorithms.
    - The reviewer in particular likes the entropic regularization for making the solution unique.
- The theoretical results look sound, and the analyses are complete: including convergence results and sample complexity.

**Weaknesses.**

1. (More like limitation than weakness). The classifier is probabilistic, which may not be permitted in some applications. On the other hand, in-processing algorithms like Reductions and AdvDeBias learn deterministic classifier (which has a smaller hypothesis class, and may explain some of the performance gap in the experiments). It would be nice to clarify this aspect.
2. The reviewer would have liked to see a comparison with Reductions.
3. Most of the techniques that led to the main results have been explored in prior work. E.g.,
    1. [1,2] have studied the linear program formulation of the fair classification problem (eq. (1)), and derived similar representation results as Theorem 4.2. And given a representation result,
    2. [3] laid out ways to turn it into pre-, in-, and post-processing algorithms. In particular, the same cost-sensitive classification scheme is used in [3].

[1] Chen et al., Post-hoc bias scoring is optimal for fair classification, 2023.
[2] Xian and Zhao, A unified post-processing framework for group fairness in classification, 2024.
[3] Zeng et al., Bayes-Optimal Fair Classification with Linear Disparity Constraints via Pre-, In-, and Post-processing, 2024.

---

> ### Author Rebuttal · Authors · 2026-03-31
>
> ### For Weaknesses
>
> > **W1:** The method yields a probabilistic classifier.
>
> **Response:** We agree that a probabilistic classifier may be less suitable in deterministic applications. However, it can be converted into a deterministic one with only a slight change in the fairness-accuracy trade-off. For in-processing, we can use the learned dual variable $\bar{\lambda}$ to construct the corresponding cost-sensitive objective and train a deterministic classifier. For post-processing, we can use $\arg\max h(x)$ as the final prediction. Since $\tau$ is small in practice, this causes only negligible fairness violation. We evaluate deterministic variants on four datasets under DP with $\xi=0.001$, averaged over 3 runs:
>
> |Dataset|Adult| |ENEM| |ACS| |CelebA| |
> |---|---|---|---|---|---|---|---|---|
> |Method|Acc|Bias|Acc|Bias|Acc|Bias|Acc|Bias|
> |OptFair-in-rand|82.82|0.0046|29.58|0.0187|46.99|0.0259|71.91|0.0160|
> |OptFair-in-det|82.74|0.0042|29.45|0.0293|47.13|0.0332|72.40|0.0169|
> |OptFair-post-rand|83.12|0.0026|27.52|0.0057|44.31|0.0553|73.83|0.0171|
> |OptFair-post-det|83.20|0.0038|27.65|0.0074|44.42|0.0586|73.96|0.0183|
>
> Overall, the deterministic variants are very close to the randomized ones, partly suggesting that the gap to deterministic baselines is not mainly caused by larger hypothesis class.
>
> > **W2:** Comparison with Reductions [1].
>
> **Response:** Thank you for the suggestion. We compare OptFair with Reductions [1] (probabilistic vision) on the binary datasets Adult and COMPAS, using the AIF360 implementation, logistic models, and $\xi = 0.001$, averaged over 3 runs:
>
> |Dataset|Adult|DP|COMPAS|DP|Adult|EO|COMPAS|EO|
> |---|---|---|---|---|---|---|---|---|
> |Method|Acc|Bias|Acc|Bias|Acc|Bias|Acc|Bias|
> |Reductions|82.69|0.0035|62.23|0.0389|83.26|0.0195|63.72|0.0592|
> |OptFair-in|82.82|0.0046|62.45|0.0414|84.91|0.0246|63.21|0.0578|
> |OptFair-post|83.02|0.0026|58.85|0.0371|84.85|0.0204|60.22|0.0719|
>
> The results show that OptFair-in performs comparably to Reductions, supporting our discussion in Section 7. OptFair-post is slightly better on Adult but worse on COMPAS.
>
> > **W3:** Most of the techniques have been explored in prior work.
>
> **Response:** We agree that our work builds on prior studies, and we do not claim that every ingredient is new in isolation. Our work focuses on extending the characterization of the optimal fair classifier to the multiclass setting via an entropy-regularized formulation, together with the corresponding proximal-gradient-based algorithms. The theoretical distinction from [2,3] is clarified in **response to Q1** below. Algorithmically, our in-processing extends reduction-based methods [1] to the multiclass setting via designing cost-sensitive loss from saddle-point problem, while our post-processing is more distinct, avoiding prior grid-search [2,4] or LP-based procedures [3].
>
> ### For Questions
>
> > **Q1:**  Theorem 4.2 has already been established in [1].
>
> **Response:** Our Theorem 4.2 is related in spirit to [1], but it is not a restatement of the result (Theorem 1 or Lemma 1 in [1]). The key difference is that we study the more general multiclass setting and do not require the continuity assumption. This introduces potential non-uniqueness and changes the proof technique. Accordingly, our proof does not rely on LP strong duality, but instead uses variational optimization techniques. We further introduce entropy regularization to resolve non-uniqueness and obtain the explicit form in Theorem 4.3.
>
> > **Q2:**  How is the probabilistic classifier evaluated?
>
> **Response:** We sample predictions according to the learned classifier, following the implementation of the probabilistic classifier in [1]. This will not significantly affect fairness: theoretically, our in- and post-processing methods have the same $O(\sqrt{1/n})$ order of fairness generalization error as prior work [2,3]; empirically, the deterministic results above show a very similar fairness-accuracy trade-off.
>
> > **Q3:**  The choice of $\tau$ for entropic regularization.
>
> **Response:** Thank you for noticing our method details. As reported in the appendix (Table 2), we search over $\tau \in$ \{$0.02, 0.05$\}, and an empirical finding is that $\tau=0.05$ gives the best fairness-accuracy trade-off in most cases.
>
>
> > **Q4:**    The running time of the algorithm.
>
> **Response:** Due to the character limit, we kindly refer the reviewer to **our response to Q2 from Reviewer HRPM** for the detailed complexity analysis and runtime results.
>
> We greatly appreciate your insightful feedback, especially on deterministic classifiers, comparison with prior work and implementation details. We hope the discussion above resolves your concerns.
>
> [1] A Reductions Approach to Fair Classification
>
> [2] Post-hoc Bias Scoring Is Optimal For Fair Classification
>
> [3] A Unified Post-Processing Framework for Group Fairness in Classification
>
> [4] Bayes-Optimal Fair Classification with Linear Disparity Constraints via Pre-, In-, and Post-processing

---

> > ### Author Rebuttal · Reviewer_n2z9 · 2026-04-04
> >
> > The reviewer thanks the author for the response, and has raised the score.

---

### Official Review · Reviewer_YAeB · 2026-03-11

**Soundness:** 3
**Presentation:** 3
**Significance:** 3
**Originality:** 3
**Overall Recommendation:** 5
**Confidence:** 3

**Summary:**

This paper studies optimal fair classifier in multi-class classification. The paper derives the optimal accuracy-fairness Pareto frontier under group-fairness constraints. Then, the paper introduces OptFair and its two variants, an in-processing reduction-based method and a post-processing plug-in method, to learn fair classifiers from limited training data. Experiments on Adult, ENEM, ACSIncome, and CelebA compare these methods against several multi-class fairness baselines and show favorable accuracy–fairness trade-offs across multiple settings.

**Compliance With Llm Reviewing Policy:**

Affirmed.

**Final Justification:**

I will maintain my positive score.

**Key Questions For Authors:**

How critical is Assumption 4.1 to the main theoretical and algorithmic results? In particular, if the assumption fails for a given fairness criterion, do the characterization and the proposed methods still apply after replacing $\xi=0$ with the minimum achievable fairness violation?

**Limitations:**

Yes.
However, the impact statement could be strengthened by discussing the potential impacts of the proposed methods, including their possible positive societal benefits, especially given that this is a fairness paper.

**Strengths And Weaknesses:**

This paper presents a solid study of fair multi-class classification. It goes beyond heuristic debiasing by deriving an explicit characterization of the Bayes-optimal accuracy–fairness frontier. The theoretical development is substantial, including both a structural characterization of the optimal fair classifier and an entropy-regularized closed-form solution that supports practical algorithm design. The resulting OptFair framework is instantiated in two variants, one in-processing and one post-processing. Experiments on multiple datasets show favorable accuracy–fairness trade-offs compared with several prior multi-class fairness baselines.

The first concern is that the sentence following Assumption 4.1 is somewhat misleading. It may leave readers with the impression that any fairness criterion, especially one expressible through the paper’s confusion matrix notations, automatically satisfies this assumption. However, this is not generally the case. For example, for a fairness metric that explicitly requires equal accuracy across groups, the claim that a naive classifier can achieve $\xi=0$ may fail.

Another concern is that the distinction between this paper and prior work (Zeng et al., 2024 or Chen et al., 2024) is not sufficiently clear. While the extension from binary to multi-class classification is evident and meaningful, the methodological differences from these prior approaches are not articulated clearly enough.

Another minor issue is that the introduction may give the impression that OptFair jointly combines in-processing and post-processing within a single method, whereas the experimental section clarifies that OptFair-in and OptFair-post are evaluated as separate variants, and their sequential combination is considered only in an ablation. The paper would benefit from stating this distinction explicitly in the introduction.

---

> ### Author Rebuttal · Authors · 2026-03-31
>
> ### For concerns
>
> > **Concern 1:** The statement following Assumption 4.1 is overly broad.
>
> **Response:** Thank you for raising this important point regarding the scope of our assumption. Our intention was not to claim that every fairness criterion expressible through the confusion-matrix notation automatically satisfies this assumption. Rather, it holds for various group fairness measures studied in prior work, including those considered in this paper, such as DP, EOP, and EO. We also note that, although a naive classifier may fail for fairness metrics requiring equal accuracy across groups, a uniformly random classifier $h(x) = \left(\frac{1}{m}, \cdots, \frac{1}{m}\right) \in \Delta_m$ for all $x \in \mathcal{X}$ can still satisfy Assumption 4.1 for this class of metrics, as discussed in the Appendix B.1. By contrast, there remain fairness constraints for which a solution satisfying Assumption 4.1 may not exist in the hypothesis class, such as loss-based metric (e.g. Equalized Loss [1]). We will revise the text to make this scope explicit and avoid overgeneralization.
>
> > **Concern 2:** The methodological differences between this paper and prior work.
>
> **Response:** We appreciate the reviewer’s careful reading and valuable suggestions. We  clarify the theoretical and algorithmic differences from prior work.
>
> - On the theory side, our main difference is that we characterize the optimal fair classifier in the multiclass setting without a continuity assumption (Theorem 4.2), and further derive a unique entropy-regularized solution (Theorem 4.3), which differs from prior fair-classification analyses [2,3].
>
> - On the algorithmic side, (1) although our in-processing method shares the high-level optimization perspective of prior work in binary case [2,4], its multiclass extension is nontrivial, where we design a multi-class cost-sensitive loss from saddle-point problem and carry out proximal-gradient updates. (2) Enabled by our explicit characterization of the optimal fair classifier, we develop a proximal-gradient-based approach for post-processing that differs fundamentally from representative binary post-processing methods based on grid search [3] or multi-class method via linear programming [5], while offering lower computational complexity and a strong approximation to the Pareto frontier.
>
> > **Concern 3:** The introduction may give the impression that OptFair combines in-processing and post-processing into a single method.
>
> **Response:** Thank you for pointing out this potential confusion in the introduction. We agree that the introduction should more clearly distinguish **OptFair-in** and **OptFair-post** as two separate variants derived from the same framework, rather than presenting them as a single method. The reason we evaluate the two variants separately is that, as shown in the ablation study, their sequential combination mainly yields an intermediate trade-off that roughly averages the performance of the two methods, rather than producing a clearly improved Pareto frontier beyond either one alone. We will clarify this point in the introduction.
>
>
> ### For Key Question
>
> > The role of Assumption 4.1 in the main results, and whether the framework remains valid if it is replaced by the minimum achievable fairness violation.
>
> **Response:** We appreciate your careful attention to the technical assumptions and this important question. The main role of Assumption 4.1 is to **guarantee feasibility**, i.e., the existence of a classifier in the hypothesis class satisfying the fairness constraint for any prescribed fairness level $\epsilon \ge 0$. If the assumption fails but there exists a minimum achievable fairness violation $c_1$, then the same characterization and the resulting in-processing&post-processing methods still apply for all $\epsilon \ge c_1$​, since the feasible set is non-empty beyond that threshold. Under this modified condition, the corresponding algorithms and generalization guarantees remain valid for all $\epsilon \ge c_1$.
>
> ### For impact statement
>
> **Response:** Thank you for this helpful suggestion. We fully agree that the impact statement can be strengthened. In the revised version, we will expand it to more clearly discuss the potential positive societal benefits of our methods, as well as their possible risks and limitations in practical deployment.
>
>
> We sincerely thank the reviewer for the valuable comments, especially those related to theoretical assumption and the distinction from prior work. We hope the discussion above resolves your concerns.
>
> [1] Loss Balancing for Fair Supervised Learning
>
> [2] Bayes-Optimal Fair Classification with Linear Disparity Constraints via Pre-, In-, and Post-processing
>
> [3] Post-hoc Bias Scoring Is Optimal For Fair Classification
>
> [4] A Reductions Approach to Fair Classification
>
> [5] A Unified Post-Processing Framework for Group Fairness in Classification

---

> > ### Author Rebuttal · Reviewer_YAeB · 2026-04-03
> >
> > I thank the authors for their detailed responses. I will keep my Accept score.

---

### Decision · Program_Chairs · 2026-04-30

**Decision:**

Accept (regular)

**Comment:**

This paper considers the prolem of characterizing the accuracy-fairness frontier in multi-class classification and on designing algorithms for navigating this frontier. Specifically, the authors propose an in-processing algorithm (based on the reduction approach) and a post-processing algorithm based on plug-in estimation, both derived from the same probabilistic characterization of the optimal fair classifier. The work builds on multi-class fairness definitions introduces by Alghamdi et al.

The reviewers were positive and appreciated the contributions of the paper. Some reviewers raised questions about the relationship to prior work and on experiments and comparisons with additional baselines. These concerns were mostly addressed during the rebuttal, after which reviewers either maintained their positive recommendation or increased their score.

Overall, this is a nice contribution to the field of fairness in machine learning and an interesting new method for fair multi-class classification.